# Evaluating present-day and future impacts of agricultural ammonia emissions on atmospheric chemistry and climate

Maureen Beaudor[1,2], Didier Hauglustaine[1], Juliette Lathière[1], Martin Van Damme[3,4], Lieven Clarisse[3], and Nicolas Vuichard[1]

[1]Laboratoire des Sciences du Climat et de l'Environnement (LSCE) CEA-CNRS-UVSQ, Gif-sur-Yvette, France
[2]Now at : High Meadows Environmental Institute, Princeton University, Princeton, NJ 08544, USA
[3]Université libre de Bruxelles (ULB), Spectroscopy, Quantum Chemistry and Atmospheric Remote Sensing (SQUARES), Brussels, Belgium
[4]Royal Belgian Institute for Space Aeronomy, Brussels, Belgium

**Correspondence:** Maureen Beaudor (mb0142@princeton.edu)

**Abstract.** Agricultural practices are a major source of ammonia ($NH_3$) in the atmosphere which has implications for air quality, climate and ecosystems. Due to the rising demand for food and feed production, ammonia emissions are expected to increase significantly by 2100 and would therefore impact atmospheric composition such as nitrate ($NO_3^-$) or sulfate ($SO_4^{2-}$) particles and affect biodiversity from enhanced deposition. Chemistry-climate models which integrate the key atmospheric physico-chemical processes along with the ammonia cycle represent a useful tool to investigate present-day and also future reduced nitrogen pathways and their impact on the global scale. Ammonia sources are, however, challenging to quantify because of their dependencies on environmental variables and agricultural practices and represent a crucial input for chemistry-climate models. In this study, we use the chemistry-climate model LMDZ-INCA (Laboratoire de Météorologie Dynamique-INteraction with Chemistry and Aerosols) with agricultural and natural soil ammonia emissions from a global land surface model (ORCHIDEE; ORganizing Carbon and Hydrology In Dynamic Ecosystems, with the integrated module CAMEO; Calculation of AMmonia Emissions in ORCHIDEE) for the present-day and 2090-2100 period under two divergent Shared Socio-economic Pathways (SSP5-8.5 and SSP4-3.4). Future agricultural emissions under the most increased level (SSP4-3.4) have been further exploited to evaluate the impact of enhanced ammonia emissions combined with future contrasting aerosol precursor emissions (SSP1-2.6 ; low emissions and SSP3-7.0; regionally contrasted emissions). We demonstrate that CAMEO emission set enhances the spatial and temporal variability of atmospheric ammonia in regions such as Africa, Latin America, and the United States in comparison to the static reference inventory (Community Emissions Data System; CEDS) when assessed against satellite and surface network observations. The CAMEO simulation indicates higher ammonia emissions in Africa relative to other studies, which is corroborated by increased current levels of reduced nitrogen deposition ($NH_x$), a finding that aligns with observations in West Africa. Future CAMEO emissions lead to an overall increase of global $NH_3$ burden ranging from 59% to 235% while $NO_3^-$ burden increases by 57% - 114% depending on the scenario even when global $NO_x$ emissions decrease. When considering the most divergent scenarios (SSP5-8.5 and SSP4-3.4) for agricultural ammonia emissions the direct radiative forcing resulting from secondary inorganic aerosol changes ranges from -114 to -160 $mW.m^{-2}$. By combining a high level of $NH_3$ emissions with decreased or contrasted future sulfate and nitrate emissions, the nitrate radiative effect can either overcompen-

sate (net total sulfate and nitrate effect of -200 $mW.m^{-2}$) or be offset by the sulfate effect (net total sulfate and nitrate effect

of +180 $mW.m^{-2}$). We also show that future oxidation of $NH_3$ could lead to an increase in $N_2O$ atmospheric sources of 0.43 to 2.10 $Tg(N_2O)yr^{-1}$ compared to the present-day levels, representing 18% of the future $N_2O$ anthropogenic emissions. Our results suggest that accounting for nitrate aerosol precursor emission levels but also for the ammonia oxidation pathway in future studies is particularly important to understand how ammonia will affect climate, air quality, and nitrogen deposition.

## 1   Introduction

Ammonia ($NH_3$) is a key atmospheric species playing a crucial role in the alteration of air quality and climate through its implication in airborne particle matter formation (PM or aerosols) (Anderson et al., 2003; Bauer et al., 2007). The resulting aerosols, namely ammonium nitrates or ammonium sulfates, have important impacts on the Earth's radiative budget due to their ability to scatter the incoming radiation, act as cloud condensation nuclei, and indirectly increase cloud lifetime (Abbatt et al., 2006; Henze et al., 2012; Behera et al., 2013; Evangeliou et al., 2020). Through dry and wet deposition processes, $NH_3$

and $NH_4^+$ are also responsible for adverse damages to the ecosystems including biodiversity loss (Stevens et al., 2020; Guthrie et al., 2018; de Vries, 2021).

Agriculture and, more specifically, livestock manure management and land fertilization account for 85% of $NH_3$ sources (Behera et al., 2013). Because the volatilization of $NH_3$ is highly dependent on soil temperature and humidity, land surface models (LSM) are promising to estimate $NH_3$ emissions at the global scale. Recently, in Beaudor et al. (2023a), a specific

LSM module dedicated to agricultural ammonia emissions (CAMEO, Calculation of AMmonia Emissions in ORCHIDEE) has been presented and evaluated against satellite-derived emission fluxes. CAMEO is a process-based model in which emissions from livestock management, grazing, and N fertilization (as well as natural soil sources) are interactively calculated within the ORCHIDEE Land Surface Model (ORganizing Carbon and Hydrology In Dynamic Ecosystems, Vuichard et al., 2019). CAMEO-based seasonal variation of $NH_3$ emissions which depend on both meteorological and agricultural practices highlights

very satisfying correlation scores with satellite-based emissions as demonstrated in Beaudor et al. (2023a, 2024) . In addition, the ability of CAMEO to simulate natural soil emissions is useful since they have been up to now poorly quantified at the global scale and appear a non-negligible source in specific regions such as Africa. Livestock activities and synthetic fertilizer use are projected to intensify in the following decades leading to a potential crucial $NH_3$ emission increase (Bodirsky et al., 2012; Popp et al., 2017; Beaudor et al., 2024). Impacts of the $NH_3$ emissions on the future nitrate aerosol formations and climate

have already been assessed in Hauglustaine et al. (2014) under Radiative Concentration Pathways (RCPs) scenarios until 2100 by using the global climate-chemistry atmospheric model LMDZ-INCA (Laboratoire de Météorologie Dynamique-INteraction with Chemistry and Aerosols). They have illustrated the substantial impact of $NH_3$ emissions on the future formation of nitrate aerosols and on the direct radiative forcing (Hauglustaine et al., 2014) . By the year 2100, due to significant emissions from agriculture, the contribution of nitrate aerosol to anthropogenic aerosol optical depth could increase by as much as fivefold,

under the most impactful scenario considered. RCP scenarios have also been exploited to study the importance of future atmospheric $NH_3$ on chemistry and climate with a special focus on atmospheric $NH_3$ losses including oxidation processes

(Paulot et al., 2016; Pai et al., 2021). The ammonia oxidation pathway mentioned is a direct contributor to nitrous oxide ($N_2O$) emissions in the atmosphere, which is a potent greenhouse gas. Future losses of nitrous oxide could increase significantly due to intensified agricultural emissions and the emerging hydrogen fuel economy, which heavily relies on ammonia as an energy carrier Hauglustaine et al. (2014); Bertagni et al. (2023). Recently, in the the Phase 6 of the Coupled Model Intercomparison Project (CMIP6) framework, socioeconomical drivers have gained greater importance and have been incorporated into a new set of scenarios called SSPs (Shared Socioeconomic Pathways, SSPs) (O'Neill et al., 2016). The Sixth Assessment Report from the IPCC covers a broader range of greenhouse gas and air pollutant trajectories through the use of SSP scenarios (Intergovernmental Panel On Climate Change, 2023). However, future agricultural $NH_3$ emissions that have been prescribed for the Sixth Assessment Report have several limitations regarding the consideration of climate and livestock densities as described in Beaudor et al. (2024). Livestock distribution, which is considered an important driver of future $NH_3$ emissions, has been recently projected up to 2100 following a unique downscalling method (Beaudor et al., 2024). In this latter study, CAMEO has been exploited to calculate future $NH_3$ emissions accounting for the evolution of climate, livestock, and N fertilizers.

By demonstrating encouraging results for the present-day, especially when compared to reference inventories, CAMEO emissions open promising perspectives to represent ammonia and related aerosols within chemistry-climate models. Like most chemistry-climate models, LMDZ-INCA relies on the seasonally-forced inventory called CEDS (Community Emissions Data System, McDuffie et al., 2020). We hypothesize that prescribing CAMEO emissions instead of CEDS for agricultural sources could improve the simulated atmospheric species and aerosol concentrations as well as N deposition fluxes, especially over Africa, with important differences in the $NH_3$ emissions have been demonstrated previously (Beaudor et al., 2023a). In the present paper, we introduce the impact of the new present-day and future CAMEO emission datasets on atmospheric chemistry by using the global LMDZ-INCA model. As a first global and regional evaluation, the columns simulated by LMDZ-INCA with CAMEO and CEDS inventories for the present-day have been compared to the $NH_3$ columns observed by the IASI satellite instrument. Statistics involving ground-based measurements of surface concentrations (trace gases: $NH_3$, $NO_2$ and ionic species: $NH_4^+$, $NO_3^-$, $SO_4^{2-}$ ) have also been performed to ensure a more robust evaluation of the model.

For the first time, we propose to investigate how future agricultural $NH_3$ emissions, influenced by climate change, livestock management, and nitrogen fertilizer use, will impact atmospheric chemistry and climate (kept at present-day conditions). We assess the effects of future emissions under SSP4-3.4 and SSP5-8.5, which represent the scenarios with the most and least significant global increases by the year 2100. SSP4-3.4 and SSP5-8.5 describing respectively "A world of deepening inequality", and "Fossil-fueled Development – Taking the Highway" (Calvin et al., 2017; Kriegler et al., 2017), reflect divergent agricultural drivers. In the first place, SSP4-3.4 represents the scenario with the weakest evolution of livestock, while SSP5-8.5 shows the most significant increase among all Shared Socioeconomic Pathways (SSPs) according to Riahi et al. (2017). In line with these trends, the fossil fuel-intensive scenario SSP5-8.5 also experiences the highest demand and production of food and feed crops among the three scenarios considered, as noted by Beaudor et al. (2024). This increase occurs despite low population growth and is driven by the prevalence of diets high in animal products (Fricko et al., 2017; Kriegler et al., 2017). Despite the peak in food and feed crop production, projected fertilizer applications in SSP5-8.5 rise only slightly. This is attributed to the

minimal production of biofuel crops, a result of the lack of climate mitigation policies and rapid advancements in agricultural productivity. In contrast, SSP4-3.4 exhibits the highest use of fertilizer and reveals significant regional differences, with high consumption lifestyles among elite socioeconomic categories and low consumption levels for the rest of the population (Calvin
et al., 2017).

Knowing the importance of sulfate dioxide ($SO_2$) and nitrogen oxide ($NO_x$) emissions for nitrate and sulfate aerosol formation (Hauglustaine et al., 2014; Lachatre et al., 2019), scenarios have been designed to evaluate the impact of future $NH_3$ emissions under contrasted $SO_2$ and $NO_x$ conditions. In this paper, we present six simulations from LMDZ-INCA, including two present-day simulations, with CEDS and CAMEO inventories for $NH_3$ emissions and four future simulations over
2090-2100 with future $NH_3$ emissions from CAMEO and other sources at different future levels (i.e. globally decreased and regionally-contrasted level of emissions). The paper is organized as follows: in Sect. 2, we present the emission inventories prepared and considered in the global chemistry-climate model for both the present-day and future (2100) simulations. In Sect. 3, we describe the LMDZ-INCA chemistry-climate model used along with modelling setup. Sect. 4 presents the simulated $NH_3$ columns and the N deposition fluxes for the present-day including an evaluation of the model performance with
IASI and ground-based measurements. In Sect.5 the perturbations associated with future agricultural emissions on atmospheric chemistry and climate under different scenarios are illustrated. Finally, in Sect. 6, we draw the conclusions from this work.

## 2  Emissions datasets

In this work, we focus on the impact of agricultural $NH_3$ emissions calculated from CAMEO on atmospheric chemistry. Therefore, specific attention is given to the modelling of these emissions, which are further detailed in the two following
sub-sections. Please note that except for the agricultural and land-related $NH_3$ emissions, all the other anthropogenic sources used in this study are based on the same sets of data (i.e. derived from the CMIP6 exercise both for present-day (CEDS) and future scenarios (McDuffie et al., 2020; Gidden et al., 2018)). Emissions from biomass burning, including small fires from agricultural waste burning come from the Global Fire Emissions Database (GFED4s) inventory (van der Werf et al., 2017). $NH_3$ emissions from fire account for 4.2 $TgNyr^{-1}$ for the historical period (this source is excluded from the values in Table
1). The global anthropogenic $NH_3$, $NO_x$ and $SO_2$ emissions used in the study are presented in Table 1. As comparison the EDGARv8.1 inventory (https://edgar.jrc.ec.europa.eu/index.php/dataset_ap81) quantifies for all anthropogenic sectors a total of $NH_3$ emissions of 42 $TgNyr^{-1}$ in 2010 (including 36 $TgNyr^{-1}$ for the agricultural sector).

### 2.1  Present-day agricultural $NH_3$ emissions

Two present-day agricultural $NH_3$ emission datasets are tested. One simulation accounts for emissions from CEDS (McDuffie
et al., 2020) and another one uses the estimated emissions from the CAMEO module included in the LSM ORCHIDEE described in Beaudor et al. (2023a). CAMEO simulates manure production and agricultural $NH_3$ emissions from the manure management chain (including manure storage and grazing) and soil emissions after fertilizer or manure application. CAMEO simulates interactive $NH_3$ emissions into the global LSM ORCHIDEE (Krinner et al., 2005; Vuichard et al., 2019). In addition,

**Table 1.** Global anthropogenic ammonia (NH$_3$), nitrogen (NO$_x$) and sulfate (SO$_2$) emissions used for the present-day (2004-2014) and future (2090-2100) simulations. Agricultural NH$_3$ emissions are presented in parentheses. Please note that CAMEO emissions also include natural soil emissions. (TgNyr$^{-1}$ or TgSyr$^{-1}$).

| Simulation | NH$_3$ | NO$_x$ | SO$_2$ |
|---|---|---|---|
| Present-day (2004-2014) | | | |
| CEDS | 54 (38) | 39 | 64 |
| CAMEO | 64 (35) | 39 | 64 |
| Future (2090-2100) | | | |
| CAMEO[585] | 84 (50) | 39 | 64 |
| CAMEO[434] | 98 (68) | 39 | 63 |
| CAMEO[434-126] | 99 (68) | 9.2 | 11 |
| CAMEO[434-370] | 105 (68) | 39 | 47 |

natural soil NH$_3$ emissions are also accounted for in CAMEO. ORCHIDEE represents the C and N cycles and simulates the water and energy fluxes within the ecosystems. The vegetation is represented by 15 Plant Functional Type (PFTs) among which 2 crop types (C3 and C4) and 4 grass types (temperate, boreal and tropical C3 grasses and a single C4 grass). ORCHIDEE is constrained by land-use maps, meteorological fields, and N input such as synthetic fertilizers. Livestock densities represent one of the most critical inputs for CAMEO since it is the main driver of the feed need estimation and, thus, of manure management and, to a lesser extent, soil emissions.

Emissions from agriculture are slightly lower in CAMEO compared to CEDS (35 against 38 TgNyr$^{-1}$), but additional natural soil emissions account for 13 TgNyr$^{-1}$. As a result, global annual NH$_3$ emissions from CAMEO are 10 TgNyr$^{-1}$ higher than in CEDS (Table. 1). Please note that due to a different set of input data, the agricultural ammonia emissions from the present study are 9 TgNyr$^{-1}$ lower than the one reported in the reference study (Beaudor et al., 2023, GMD). This difference is mainly explained by the non-consideration of managed grasslands in the CMIP6 synthetic fertilizer forcing which led to a total fertilization input of 97 TgNyr$^{-1}$ against 118 TgNyr$^{-1}$ in the reference study. On another hand, the different climatic forcings may also impact the emissions. For self-consistency, CAMEO for the CMIP6 framework exploits the 3-hourly near-surface meteorological fields simulated by the Institut Pierre Simon Laplace (IPSL) Earth System Model : IPSL-CM6A-LR ESM (Boucher et al., 2020), in the context of CMIP6 for near-surface air temperature, specific humidity, wind speed, pressure, short- and longwave incoming radiation, rainfall, and snowfall. The reference paper is based on the Climatic Research Unit (CRU) and Japanese reanalysis (JRA) dataset (CRU-JRA V2.1) (Harris et al., 2014) (preprocessed and adapted by Vladislav Bastrikov, LSCE, July 2020), provided at 6 h time steps.

## 2.2 Future emission scenarios

In this study, future emissions for different SSPs are used for the 2090-2100 period. CAMEO emissions for SSP5-8.5 and SSP4-3.4 have been exploited for future agricultural and natural $NH_3$ emissions in the CAMEO[SSPi] (SSPi: 585, 434, 434-126, 434-370) simulations where agricultural sources account for 50 and 68 $TgNyr^{-1}$ (respectively for SSP5-8.5 and SSP4-3.4). SSP5-8.5 and SSP4-3.4 have been chosen primarily as they represent, respectively, the least and most important increase of $NH_3$ emissions estimated over 2090-2100 (Beaudor et al., 2024) The evolution of the global agricultural NH3 emissions from the different SSPs from CAMEO under future climate is shown in Fig. S1 in the Supplementary Material. These emission datasets have been recently constructed from a newly gridded livestock product and the use of the global process-based CAMEO before being evaluated against CMIP6 emissions developed by the Integrated Assessment Models (IAMs) in Beaudor et al. (2024). The future livestock distribution has been estimated until 2100, originally, for three divergent SSPs (SSP2-4.5, SSP4-3.4 and SSP5-8.5) through a downscaling method based on regional livestock trends and future grassland areas (the detailed methodology can be found in Beaudor et al., 2024).

In addition to future CAMEO $NH_3$ emissions for SSP4-3.4 and SSP5-8.5, future CMIP6 emissions have been used for SSP1-2.6 and SSP3-7.0 (Gidden et al., 2018) for other emitted species but also for the anthropogenic sectors - other than agriculture - for $NH_3$ (waste, industry, etc). These two SSPs were selected because they represent divergent scenarios for global $NO_x$ and $SO_2$ emissions. SSP1-2.6 represents a "low" scenario with air pollution and climate change being strongly mitigated. Emission regulations are implemented almost worldwide, in various economic sectors such as energy generation, industrial processes, and transportation. In contrast, no climate change mitigation and only weak air pollution control are considered in SSP3-7.0. This scenario features contrasting emission trends, with strong regulations in the Northern Hemisphere and increasing emissions in the Southern Hemisphere. A slight difference in the $NH_3$ emissions is observable from CAMEO[434-126] and CAMEO[434-370] (Table 1). This difference is explained by the differences in the emissions from other anthropogenic sectors between SSP1.2-6 and SSP3-7.0. It is worth noticing that even though future total $NO_x$ emissions are similar between the present-day level and under SSP3-7.0 at the global scale, different regional patterns are observable (see Figure S2 in the Supplementary Material).

Beaudor et al. (2024) demonstrate a global agreement between agricultural ammonia emissions developed by the IAMs and simulated with CAMEO. The global estimates from the IAMs inventories are, respectively, 50 and 66 $TgN.yr^{-1}$ under SSP5-8.5 and SSP4-3.4, compared to 50 $TgN.yr^{-1}$ and 68 $TgN.yr^{-1}$ for CAMEO. In this previous work, three interesting advantages are highlighted in favor of the use of CAMEO emissions:

- The consideration of environmental conditions (i.e. soil temperature and humidity, $CO_2$ increase, vegetation changes).

- The consistent consideration of the key ammonia emissions drivers (i.e. N input, meteorology, livestock, and land use) among all future SSPs which is the result of the use of a single process-based model.

- The spatial heterogeneity is driven by environmental conditions and not kept constant over time within predefined regions using the information from the historical period.

– Incorporating CAMEO into the land component of the IPSL ESM ensures better consistency throughout the various components, including LMDZ-INCA, paving the way for advancements in our understanding.

    Considering the constraints of IAMs in precisely reflecting the primary factors influencing ammonia emissions, exploring their effects on atmospheric chemistry and climate beyond a global level appears unconvincing. We propose a hypothetical comparison based on the regional differences observed in the IPCC emissions and the CAMEO emissions projected for 2100.

Figure S3 (Supplementary Material) highlights the major regional differences between CMIP6 and CAMEO emissions in 2100 for the two considered SSPs (SSP4-3.4 and SSP5-8.5). The most distinguishable region is Africa, specifically North Africa's savanna combined with equatorial Africa, where the CMIP6 emissions for both SSPs are more than twice as high as those for CAMEO (>15 $TgN.yr^{-1}$). The primary explanation for this pattern lies in the simplified downscaling strategy adopted by the IAM method for projection. The approach applies a constant factor across the entire African continent over time, neglecting to

account for regional influences such as livestock food expansion and changes in fertilizer application. Specifically, the northern Maghreb region is expected to play a significant role in the future, particularly under SSP4-3.4, as projections indicate an expansion in cultivated lands and fertilizer application, likely driven by the cultivation of bioenergy crops. As a consequence, one of the most expected differences between CMIP6 and CAMEO emissions impact would be a more enhanced production of aerosol formation and NOy and NHx deposition under [434-370] where $NO_x$ and $SO_2$ emissions are projected to increase

compared to the present-day in Africa. In contrast, in China, the smaller emission fluxes predicted by the IAMs under both SSPs compared to CAMEO indicate that we can expect a limitation / decrease in the formation of ammonium-related aerosols and therefore the resulting deposition, which would be stronger under [434-126].

## 3   The LMDZ-INCA model

    The LMDZ-INCA global chemistry–aerosol–climate model couples the LMDZ (Laboratoire de Météorologie Dynamique,

version 6) general circulation model (GCM; Hourdin et al., 2020) and the INCA (INteraction with Chemistry and Aerosols, version 5) model (Hauglustaine et al., 2004, 2014). The interaction between the atmosphere and the continental surface is ensured through the coupling of LMDZ with the ORCHIDEE (version 1.9) dynamical vegetation model (Krinner et al., 2005). The present configuration is parameterized with the "Standard Physics" of the GCM (Boucher et al., 2020). The model incorporates 39 hybrid vertical levels extending up to 70 km with a horizontal resolution of $1.3°$ in latitude and $2.5°$ in longitude.

The GCM primitive equations are solved with a 3 min time step, large-scale transport of tracers is carried out every 15 min, and physical and chemical processes are calculated at a 30 min time interval.

    The INCA model represents a state-of-art $CH_4$-$NO_x$-CO-NMHC-$O_3$ tropospheric photochemistry (Hauglustaine et al., 2004; Folberth et al., 2006). The tropospheric photochemistry and aerosol scheme includes a total of 123 tracers including 22 tracers representing aerosols. The model represents 234 homogeneous chemical reactions, 43 photolytic, and 30 hetero-

geneous reactions. The tropospheric chemistry reactions are listed in Hauglustaine et al. (2004) and Folberth et al. (2006). Comparisons with observations have been extensively carried out to evaluate the gas-phase version of the model in the lower stratosphere and upper troposphere. The distribution of aerosols is represented by considering anthropogenic sources such as

sulfates, nitrates, black carbon (BC), organic carbon (OC), as well as natural aerosols such as sea salt and dust. Reactions in the heterogeneous phase on both natural and anthropogenic tropospheric aerosols are also included (Bauer et al., 2004; Hauglustaine et al., 2004, 2014). A modal approach for the size distribution is used to track the number and mass of aerosols which is described by a superposition of five log-normal modes (Schulz, 2007). The particle modes are represented for three ranges: sub-micronic (diameter <1 $\mu$m) corresponding to the accumulation mode, micronic (diameter between 1 and 10 $\mu$m) corresponding to coarse particles, and super-micronic or super coarse particles (diameter >10 $\mu$m). The diversity in chemical composition, hygroscopicity and mixing-state is ensured by distinguishing soluble and insoluble modes. Specifically, soluble and insoluble aerosols are treated separately in both sub-micron and micron modes. Sea salt, $SO_4$, $NO_3$, and methane sulfonic acid (MSA) are treated as soluble components of the aerosol, dust is treated as insoluble, whereas BC and OC appear in both the soluble and insoluble fractions. Ammonia and nitrate aerosols are represented through an extended chemical scheme that includes the ammonia cycle as described by Hauglustaine et al. (2014). The formation of the ammonium sulfate aerosols depends on the relative ammonia and sulfate concentrations and is characterized by three chemical domains (ammonium-rich, sulfate-rich and sulfate-very rich conditions) as in Metzger et al. (2002). Extensive evaluations of the aerosol component of the LMDZ-INCA model have been carried out during the various phases of Aerosol Comparisons between Observations and Models (i.e. AeroCom Gliß et al., 2021; Bian et al., 2017). Simulated surface $NH_3$, $HNO_3$, $NH_4^+$, $SO_4^{2-}$, $NO_3^-$ concentrations indicate satisfying performances when evaluated against observation network from the US, Europe and Asia Bian et al. (2017) The dry and wet deposition processes of ammonia, ammonium nitrate and ammonium sulfate are described in Hauglustaine et al. (2004) with updated Henry's law coefficients taken from Sander (2023). Coarse nitrates on dust and sea salt are deposited as the corresponding dust and sea-salt components. In addition to the concentrations of ammonia-related aerosols and gases exploited in this study, the all-sky direct radiative fluxes at the top of the atmosphere and the Aerosol Optical Depth (AOD) of the various aerosol components. Multiple radiative forcings (RFs) and aerosol optical depths (AODs) related to changes in atmospheric composition due to agricultural emissions are calculated online during the LMDZ-INCA simulations. As mentioned by Terrenoire et al. (2022), the radiative calculations in the general circulation model (GCM) utilize an enhanced version of the ECMWF scheme established by Fouquart (1980) for the solar spectrum and by Morcrette (1991) for the thermal infrared spectrum. The short-wave spectrum is segmented into two ranges: 0.25–0.68 and 0.68–4.00 $\mu$m. The model incorporates the diurnal variation of solar radiation and permits fractional cloud cover within a grid cell. These RFs are computed as instantaneous, clear-sky, and all-sky forcings at both the surface and the top of the atmosphere. To evaluate the future effects of ammonia emissions on aerosol concentration and climate, the all-sky direct radiative forcings are determined by subtracting the historical CAMEO radiative fluxes from the future simulation being analyzed. In Section 5.3, the all-sky forcings at the top of the atmosphere and AOD will be discussed for aerosols, similar to what was done by Hauglustaine et al. (2014).

Ammonia losses occur as a result of both wet and dry deposition, ammonium formation, and the oxidation processes in the gas phase, although the latter only contributes a small amount to its overall loss. However, the loss through this oxidation pathway generates a non-negligible amount of nitrous oxide ($N_2O$). The production of $N_2O$ results from the following reaction:

$$NH_2 + NO_2 \rightarrow N_2O + H_2O \tag{1}$$

The overall production rate exploited in the study is calculated as:

$$\text{R}_{\text{NH2}\rightarrow\text{N2O}} = A \times \exp\left(\frac{-Ea/R}{T}\right)[\text{NH}_2][\text{NO}_2] \qquad (2)$$

With the Arrhenius factor $A = 2.1e^{-12}$, $E_a$ the molar activation energy for the reaction and R is the universal gas constant such as Ea/R = -650.

## 3.1 Model setup

The model was run with meteorological data from the European Centre for Medium-Range Weather Forecasts (ECMWF) ERA5 reanalysis. The GCM wind components are adjusted using the ECMWF meteorology and applying a correction term to the GCM u and v wind components at each time step with a relaxation time of 2.5 h (Hauglustaine et al., 2004). The ECMWF fields are provided every 6 hours and interpolated onto the LMDZ grid. We focus this work on the impact of agricultural emissions calculated from CAMEO on atmospheric composition and its future evolution. The CAMEO emissions are, first, carefully regridded onto the model grid through a preprocessor program and provided at a monthly time resolution to the chemistry-transport model. In order to isolate the impact of CAMEO $NH_3$ emissions, all snapshot simulations are performed under present-day climate conditions and run for 11 years after a 2-year spin-up. Therefore, ECMWF meteorological data for 2004–2014 are used. The combined impact of climate change and future agricultural emissions $NH_3$ on atmospheric chemistry and climate is an interesting topic to further investigate in the future.

Natural emissions are aggregated to anthropogenic sources in the INCA model. Biogenic surface fluxes of isoprene, terpenes, acetone, methanol are calculated offline within the ORCHIDEE vegetation model as described by Messina et al. (2016). $NH_3$ emissions from ocean are taken from Bouwman et al. (1997) and reach 8.2 $\text{TgNyr}^{-1}$ for the present-day, which is higher than the estimate from Paulot et al. (2015) (2-5 $\text{TgNyr}^{-1}$). Natural emissions of dust and sea salt are computed using the 10 m wind components from the ECMWF reanalysis. For the future simulations (2090-2100), the SSP1-2.6 and SSP3-7.0 anthropogenic emissions (except agricultural sources) provided by Gidden et al. (2018) are used. Natural emissions (except natural soil $NH_3$ emissions) and biomass burning for both gaseous species and particles are kept to their present-day level even in future simulations in order to isolate the impact of CAMEO emissions. In total, we performed six simulations, including two present-day simulations, respectively, with CEDS (1) and CAMEO (2) inventories for $NH_3$ emissions and four future simulations over 2090-2100 with $NH_3$ emissions from CAMEO under SSP5.8-5 and SSP4-3.4 by keeping other sources from the present-day levels (3 and 4), by taking the SSP1-2.6 (low levels; 5) and SSP3-7.0 (regionally-contrasted conditions; 6) for other sources. Table 2 summarizes the simulations performed and analyzed in this study.

## 4 Present-day atmospheric ammonia, aerosol concentrations and nitrogen deposition fluxes

### 4.1 Global, regional and seasonal evaluation with IASI

For over a decade, the IASI instrument has provided measurements of $NH_3$ at a satisfying spatial resolution and large-scale coverage, which is convenient for modelling evaluation (Van Damme et al., 2014). The simulated monthly distributions of

**Table 2.** Simulation set-up, scope and corresponding emission datasets. The emission period used is given in parentheses. 'Other anth.' accounts for all the species for all the anthropogenic sectors except $NH_3$ emitted from the agricultural sector.

| Simulation name | Agri. $NH_3$ emissions | Other anth. emissions | Scope of the simulations |
|---|---|---|---|
| Present-day (2004-2014) | | | |
| (1) CEDS | CEDS (2004-2014) *McDuffie et al. (2020)* | CEDS (2004-2014) *McDuffie et al. (2020)* | Historical reference CEDS simulation |
| (2) CAMEO | CAMEO (2004-2014) *Beaudor et al. (2023a)* | CEDS (2004-2014) *McDuffie et al. (2020)* | Historical reference CAMEO simulation |
| Future (2090-2100) | | | |
| (3) CAMEO[585] | CAMEO SSP5-8.5 (2090-2100) *Beaudor et al. (2024)* | CEDS (2004-2014) *McDuffie et al. (2020)* | Effects of low rise in agri. $NH_3$ emissions (high livestock pressure but efficient agriculture) |
| (4) CAMEO[434] | CAMEO SSP4-3.4 (2090-2100) *Beaudor et al. (2024)* | CEDS (2004-2014) *McDuffie et al. (2020)* | Effects of high rise in agri. $NH_3$ emissions (intensive use of fertilizer) |
| (5) CAMEO[434-126] | CAMEO SSP4-3.4 (2090-2100) *Beaudor et al. (2024)* | CEDS SSP1-2.6 (2090-2100) *Gidden et al. (2018)* | Effects of high rise in agri. $NH_3$ emissions under strict global regulations of all other anth. sectors |
| (6) CAMEO[434-370] | CAMEO SSP4-3.4 (2090-2100) *Beaudor et al. (2024)* | CEDS SSP3-7.0 (2090-2100) *Gidden et al. (2018)* | Effects of high rise in agri. $NH_3$ emissions under regionally-contrasted regulations of all other anth. sectors |

$NH_3$ are evaluated against observations over 2011-2014 from IASI at the global and regional scale. The IASI data used in
this study originates from the IASI instruments onboard Metop-A and B, which were launched in 2006 and 2012 respectively. Each instrument overpasses the Earth two times per day with a footprint of 12 km at the nadir. The instruments cross the equator in the morning at 9:30 am and evening at 9:30 pm. Here we used the IASI $NH_3$ columns retrieved with version 3 of the "Artificial Neural Network for IASI (ANNI)" algorithm. An extended description of the retrieval methods and validation works can be found in various publications (Whitburn et al., 2016; Van Damme et al., 2017, 2021; Guo et al., 2021). In the
present study, only the morning overpasses have been used as infrared instruments are more sensitive to the lowest layers of the atmosphere at this time of the day (Clarisse et al., 2010). Considering the daily cycle of $NH_3$ and to be consistent with the satellite observations, the model was run at a 30-min time-step, and sampled at the corresponding satellite overpass time. Regarding the spatial resolution, the IASI dataset has been gridded on the LMDZ-INCA grid (i.e. resolution of $1.3°$ in latitude and $2.5°$ in longitude). The evaluation consists of comparisons of the spatial total $NH_3$ columns for the two present-day
runs (CEDS and CAMEO) along with a seasonal cycle analysis over the hot-spot regions. Taylor plots and Mean Bias Errors

scores (MBE; regional mean of the difference between the observation and model) are also presented to assess the spatial and temporal variability of the simulated concentrations compared to the IASI observations considered as the reference. While IASI observations have already been used for CTM evaluations (Heald et al., 2012; Ge et al., 2020; Wang et al., 2021; Vira et al., 2022; Ren et al., 2023), this is the first time that simulated $NH_3$ columns from LMDZ-INCA are evaluated against spaceborne observations. The Taylor plot approach aims at representing multiple statistical metrics including the normalized standard deviation, the Pearson's R correlation and a skill function which help at discriminating the best simulation. The default skill function implemented is defined in Taylor (2001) and decreases toward zero as the correlation becomes more and more negative or as the standard deviation approaches either zero or infinity.

The distributions of the $NH_3$ columns observed by IASI and computed by LMDZ-INCA with the CEDS or CAMEO $NH_3$ emissions over 2011-2014 are shown in Figure 1. The CAMEO simulation captures the $NH_3$ hotspots over India, Equatorial Africa, Latin America, and the US where the columns are in the range 1 - 6 molecules $\times 10^{16} cm^{-2}$. When the CEDS inventory is replaced by CAMEO in LMDZ-INCA, the global simulated columns are 50% higher (of around 0.04 molecules $\times 10^{16} cm^{-2}$) but closer to the IASI-measured global average (0.15 molecules $\times 10^{16} cm^{-2}$). The biggest absolute differences are located in Northwestern India where the CAMEO columns are higher by about 2 molecules $\times 10^{16} cm^{-2}$. CAMEO emissions also enhance $NH_3$ columns in China, Latin America, the US, and Africa, especially in the Equatorial region when compared to the CEDS simulation. Using CAMEO emissions improved the agreement of LMDZ-INCA with IASI observations in these regions. In particular, the Mean Bias Error (MBE) of the model is reduced from at least 49 % of the observed IASI columns in Equatorial Africa and South America when using CAMEO emissions instead of CEDS inventory (Table 3). The Taylor plots in Figure 2 represent statistical metrics for both temporal and spatial analyses. The temporal analysis is shown for monthly time steps, using triangle markers with T labels, and involves averaging over the corresponding regions. On the other hand, the spatial analysis is derived by averaging over the monthly time-series from 2011-2014, indicated by plain circle markers with S labels. These plots include metrics such as normalized standard deviation (plotted on the x-y axis, where the observation is normalized to 1), Pearson's R correlation, and a skill function, represented by grey isolines. The Taylor plots highlight the better performance of the simulated spatial representation of the $NH_3$ columns in these two regions (Equatorial Africa and South America ) when CAMEO emissions are prescribed. However, it is important to note potential compensating errors within the regions, particularly in the selected African region (shown in the black box in Figure 1). For instance, in the Saharan area, CAMEO emissions cause an overestimation of column values by 0.3 molecules $\times 10^{16} cm^{-2}$. In contrast, in the tropical Sub-Saharan zone, these emissions lead to an underestimation of column values by -0.4 molecules $\times 10^{16} cm^{-2}$ (-45%). This discrepancy might arise from inaccuracies in CAMEO emissions, considering the particular environmental conditions. $NH_3$ emissions from biomass burning can also be uncertain, and surface-atmosphere exchanges involving fire interactions present in this area are often difficult to consider accurately. Using CAMEO also significantly reduced the modeled bias over the US, with an MBE close to 0 (Table 3). It is partially explained by a closer standard deviation to the observations (normalized standard deviation around 1); however, CEDS simulation seems to be slightly more correlated to IASI (Figure 2).

On the Western coast of Africa, the CAMEO emissions also lead to an improvement where the resulting columns over the Atlantic Ocean depict the same pattern as IASI. It is explained by higher agricultural emissions and the addition of natural soil

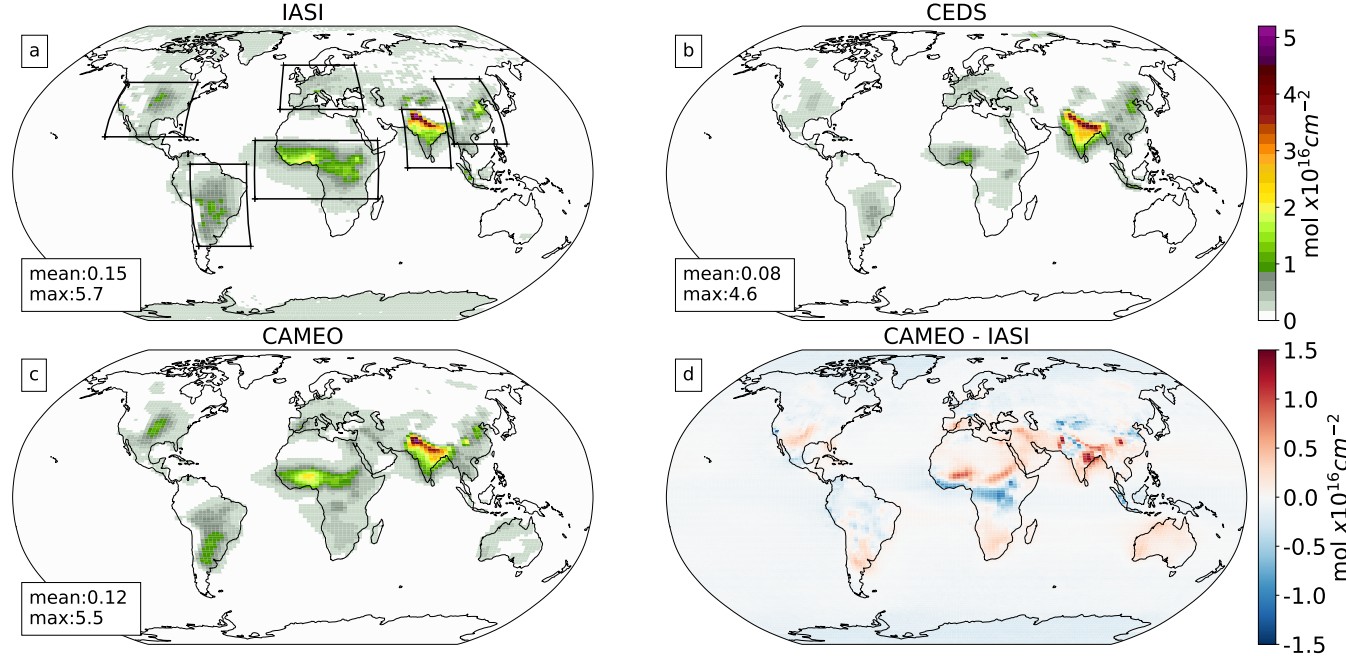

**Figure 1.** Mean annual $NH_3$ atmospheric columns observed by the IASI instrument (a) and calculated in the CEDS (b) and CAMEO (c) simulations (2011-2014). The absolute anomalies between the CAMEO and IASI columns are shown in (d). The black boxes in (a) delimit the regional bounds used in the statistical analysis in the Taylor plots (Figure 2), the time-series analysis (Figure 3) and in the Mean Bias Error calculation in Table 3. (molecules $\times 10^{16} cm^{-2}$).

emissions calculated by CAMEO, which are missing in CEDS (see Figure S2 in the Supplementary Material). In India, both model simulations result in a normalized standard deviation close to 1 for the spatial distributions. The correlations are high (|r|> 0.8), but the spatial patterns correlate better with IASI when using CAMEO. Over India, even though the bias is slightly reduced in CAMEO, the model overestimates the columns with a remaining high bias (~-0.18 molecules $\times 10^{16} cm^{-2}$).

The mean seasonal cycle over 2011-2014 is also analyzed for several regions (Figure 3). The seasonal cycle of the two simulated $NH_3$ columns correlates with the emission's temporal evolution (not shown). The seasonal variations of $NH_3$ columns in the CEDS simulation highlight two peaks in April and September for most regions reflecting the artificial seasonal profile used in the inventory. In CAMEO, the seasonality varies according to the region. In the US and Europe, the CAMEO columns show a unique peak (0.7 molecules $\times 10^{16} cm^{-2}$) during summer, while the IASI observations inform about a lower maximum

value (0.5 and 0.4 molecules $\times 10^{16} cm^{-2}$, respectively) reached over March-September. In Europe, CEDS surpasses CAMEO when it comes to the magnitude of seasonal variations. In Equatorial Africa, South America, India and China, CAMEO shows good agreement with the IASI columns, and the seasonal cycles are very close, with values of the same ranges. CAMEO emissions improve the representation of the atmospheric columns, especially in South America and Equatorial Africa, where the

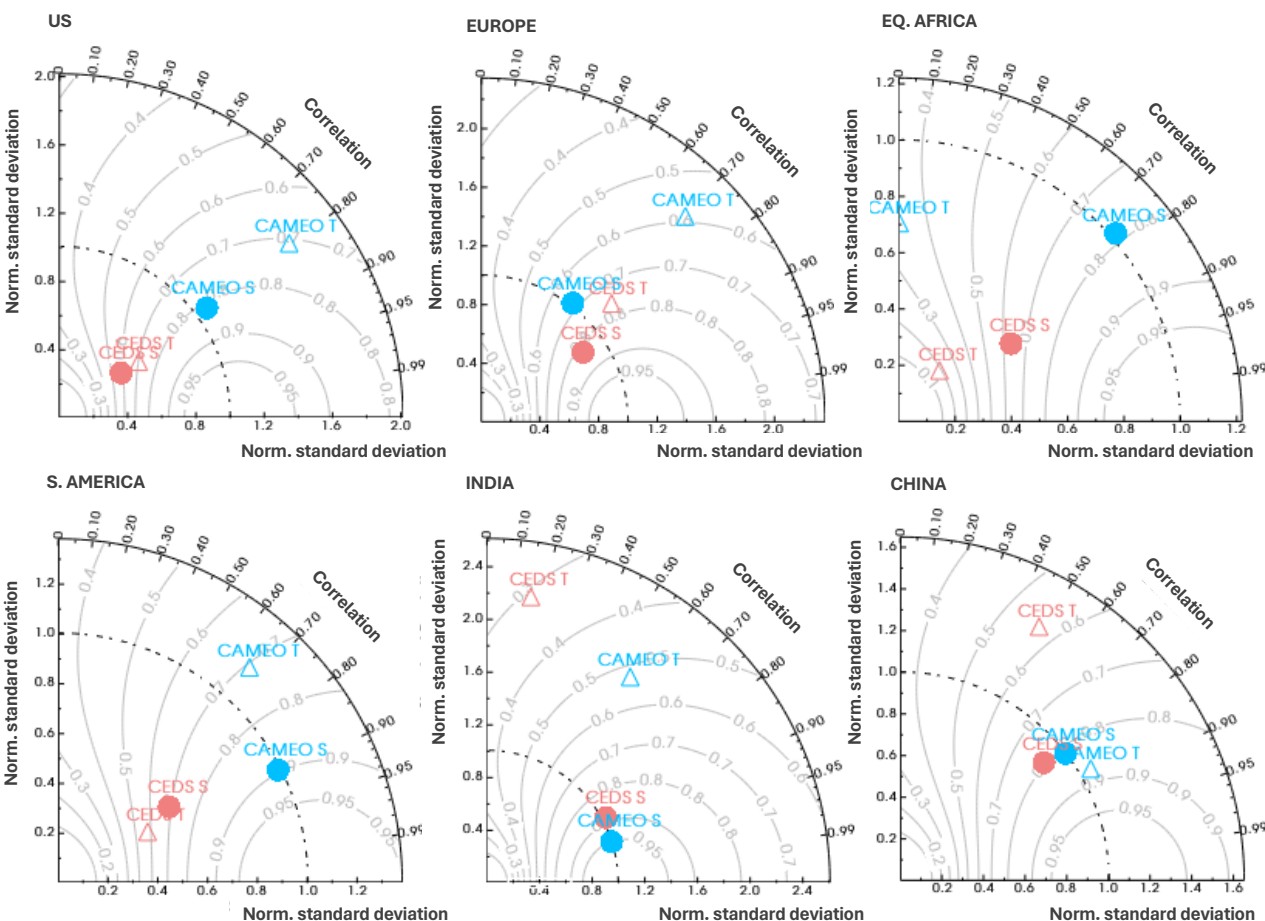

**Figure 2.** Regional Taylor plots for the simulated atmospheric NH$_3$ columns from the CAMEO and CEDS simulations evaluated with IASI observations. The plots include temporal at the monthly time step (first averaged in space over the corresponding regions, triangle markers, and T labels) and spatial (first averaged in time over monthly time-series of the 2011-2014 period, plain circles markers, and S labels) statistic metrics, including the normalized standard deviation (presented on the x-y axis, observation = 1), the Pearson's R correlation and a skill function (grey isolines). It is important to note that each region has been chosen carefully with a sufficient number of pixels as given in Table 3. The plots have been performed by using the CDAT library in Python according to Taylor (2001).

columns in CEDS are at least 2 times lower compared to IASI. In Africa, the temporal variability is more accurately simulated
with CAMEO with a higher skill function in the Taylor plot, even though the correlation is reduced (Figure 2). Over South America, the improvement is even stronger where the skill function gained 2 units. In India, both CAMEO and CEDS simulate a peak value occurring 2 months earlier than that measured by IASI, but the value is 1.5 times higher with CEDS than with CAMEO. CAMEO depicts a better seasonal amplitude with a 2 month peak starting in May and lasting until June, closer to the observations leading to a better model performance (Figure 2).

**Table 3.** Regional spatial Mean Bias Error (MBE) NH$_3$ columns from CEDS and CAMEO simulation (molecules $\times 10^{16}$cm$^{-2}$). The biases are computed by using IASI observations over the 2011-2014 period. The numbers of pixels within the regions over which the average has been computed are given in parenthesis for each region

| Region (# pixels) | Mean Obs. | MBE CEDS | MBE CAMEO |
| --- | --- | --- | --- |
| Eq. Africa (775) | 0.51 | 0.30 | 0.05 |
| China (360) | 0.31 | -0.06 | -0.01 |
| Europe (418) | 0.21 | 0.01 | 0.003 |
| India (286) | 0.83 | -0.23 | -0.18 |
| S. America (504) | 0.37 | 0.21 | -0.0007 |
| US (460) | 0.28 | 0.13 | -0.001 |

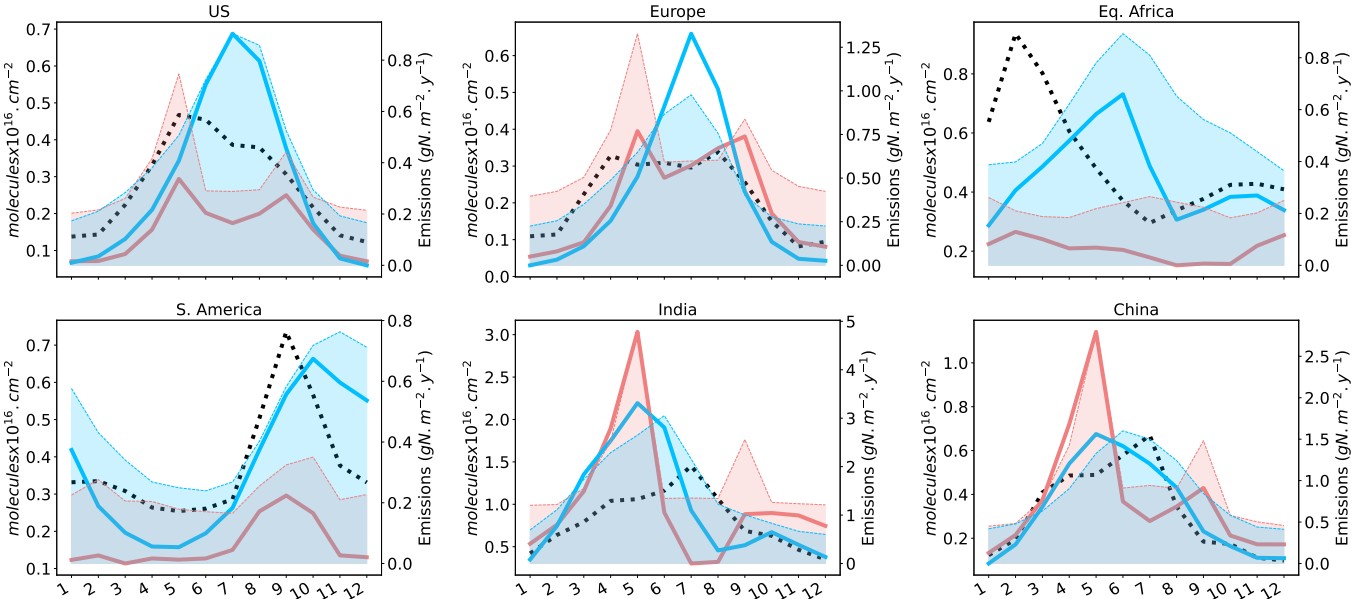

**Figure 3.** Regional mean seasonal cycle (2011-2014) of NH$_3$ atmospheric columns observed by the IASI satellite (black dotted lines) and calculated in the CEDS (pink lines) and CAMEO (blue lines) simulations (molecules $\times 10^{16}$cm$^{-2}$). Regional total NH$_3$ emissions from CEDS and CAMEO are shown with the shaded areas. Emissions include also biomass burning from GFED4s and other anthropogenic emissions from CEDS for consistency with the simulated columns (gNm$^{-2}$yr$^{-1}$)

## 4.2 Regional comparison with worldwide ground-based networks

Ten monitoring networks of surface NH$_3$, NH$_4^+$, NO$_2$, NO$_3^-$ and SO$_4^{2-}$ concentrations from East Asia, North America and Europe have been exploited to extend our evaluation beyond the NH$_3$ columns. The simulated surface concentrations for 2015 have been compared yearly with the data observed from 2015 by extracting the closest pixel from the LMDZ-INCA

simulation for each site. As recommended in Ge et al. (2021), we only consider measurements where 75% of the year was captured to avoid bias in our analysis and we perform yearly averages on the model data. In this study, we utilize data from the Chinese National Nitrogen Deposition Monitoring Network (NNDMN from Xu et al., 2019), the acid deposition monitoring network in East Asia and Southeast Asia (EANET, 13 countries, https://www.eanet.asia/), the UK Acid Gases and Aerosol Monitoring Network (AGANet, 30 sites, https://uk-air.defra.gov.uk/networks/network-info?view=aganet), the European Monitoring and Evaluation Programme/Chemical Coordinating Centre (EMEP/CCC, https://ebas-data.nilu.no/default.aspx), the United States Environmental Protection Agency (EPA, https://www.epa.gov/outdoor-air-quality-data), the Ammonia Monitoring Network (AMoN, https://nadp.slh.wisc.edu/sites/amon-ab35/) and the National Air Pollution Surveillance (NAPS, https://www.canada.ca/en/services/environment/weather/airquality.html) program. Main statistic scores are given in Table 4 comparing observations with both CAMEO and CEDS runs. Scatter plots of the annual mean modeled in CAMEO and measured surface concentrations along with Pearson's coefficients are given for each regional network in Figure S4, S5 and S6 from the Supplementary Material. Analog plots for the CEDS simulation are given in the Supplementary Material (Fig. S7, S8 and S9). An evaluation for 2010 has also been conducted to enhance the robustness of our findings, and similar regional signals are found as for 2015. Owing to the fewer observations available globally in 2010 compared to 2015, these results are presented in the Supplementary Material (Fig. S10, S11, and S12). Overall, surface $NH_3$, $NH_4^+$, $NO_2$, $NO_3^-$ and $SO_4^{2-}$ concentrations simulated by LMDZ-INCA are well correlated to the observations worldwide ($R_T > 0.5$). However, simulated concentrations are underestimated for most species, especially in China, where observed concentrations are by far the highest, with, for example, an estimated MBE for $NH_3$ concentrations at 6.0 $\mu$g.m$^{-3}$ (annual observed average at 10.4 $\mu$g.m$^{-3}$, Table 4). This positive bias seems to be due to an underestimation in the hotspot region of Beijing but also in more remote areas where differences can reach 15.5 $\mu$g.m$^{-3}$ (Figure S4, subplot F). The IASI instrument does not necessarily detect the highest columns in these regions (Figure 1). For most networks, prescribing CAMEO highlights reductions in bias compared to CEDS (-15% for US AMoN and around -4.5 % for NNDMN and NAPS). In North America, CAMEO reflects a realistic spatial pattern against measurements with high concentrations of $NH_3$ located in the Mid-West region of the US ($> 4$ $\mu$g.m$^{-3}$) and rather low concentrations on the Mid-Atlantic side. An underestimation of CAMEO is still observable in the North-East region of the Mid-West ($<2$ $\mu$g.m$^{-3}$; Figure S6, subplot F). Even though the spatial gradient is fairly represented in the model, it is crucial to note that only a few observations are available, especially in the Mid-US region. This intensive agricultural area would benefit from further observation data for more accurate evaluation. CAMEO emissions do not improve the $NH_3$ concentration representations measured in the EANET and European networks.

It is worth pointing out that the model-observation comparison highlights an underestimation of the simulated ammonium-nitrate concentrations at the surface (Figures S4-S12, subplots B and D). A combination of factors explains the low simulated nitrate concentrations at the surface. This version of the model has always shown a strong vertical transport combined with low scavenging in the upper troposphere (Bian et al., 2017). To some extend, this strong transport of nitrates to the upper troposphere is a robust signal and has been observed in the Asian Tropopause Aerosol Layer region during the monsoon season (June-July-August) (Höpfner et al., 2019; Yu et al., 2022). However, the CAMEO $NH_3$ emissions are significantly increased compared to CEDS during this period over India; more nitrates are produced and subsequently transported to the

upper-troposphere (UT) in that region and then spread all over the globe due to the high residence time of aerosols in the UT. This feature of the scavenging is currently investigated in a newer version (79 levels, CMIP6 physics) of the model.

The main takeaway from the evaluation of $NH_3$ columns and surface concentrations is that using CAMEO emissions results in a significant improvement in the spatial and temporal patterns, particularly in the seasonal cycle, compared to CEDS, except in the US and Europe. It is still important to note that, CAMEO improves the ground spatial variability of $NH_3$ in the US as highlighted by measurement comparison. The skill functions shown in the Taylor plots indicate that CAMEO emissions can more accurately capture the temporal variability of emissions in hotspot regions when compared to IASI observations. It is important to focus on matching seasonal cycles rather than only comparing annual averages for multiple reasons. Seasonal cycles provide insights into the variations in emissions and atmospheric pathways throughout the year, which can be linked to meteorological conditions (air temperature and precipitation), seasonal activities (like fertilizer application or manure handling) and specific events (like biomass burning). Understanding these patterns allows for more accurate predictions of air pollution and climate impacts. The effort to improve emission estimates, particularly in regions where discrepancies exist, such as Europe and the US, highlights the importance of utilizing process-based approaches that lets room for considering the bi-directionality property of ammonia.

## 4.3 Surface nitrogen deposition intercomparison

In this section, we present an analysis of the total (dry plus wet) annual deposition of $NH_x$ (= $NH_3$ + $NH_4^+$), and $NO_y$ (= NO + $NO_2$ + $NO_3$ + $HNO_2$ + $HNO_3$ + $HNO_4$ + 2 $N_2O_5$ + PAN + organic nitrates + particulate $NO_3^-$ ).

The simulated deposition fluxes (CEDS and CAMEO) are also compared against two model-based estimates, one used in the most recent CMIP exercise (IGAC/SPARC Chemistry–Climate Model Initiative (CCMI; Eyring et al., 2013 hereafter)) and the other using EMEP MSC-W (European Monitoring and Evaluation Programme Meteorological Synthesizing Centre –West) from Ge et al. (2022). N depositions fluxes from CCMI are commonly used as forcing files in LSM, as in the ORCHIDEE model. CCMI deposition fields are available globally at a resolution of 0.5 × 0.5 degrees from 1860 to 2014. In the CCMI models, nitrogen emissions from natural biogenic sources, lightning, anthropogenic sources, and biomass burning are taken from CMIP5 exercise (Lamarque et al., 2010). Regarding N deposition from Ge et al. (2022), the CTM EMEP MSC-W has been used to simulate dry and wet deposition fluxes of $N_r$ species for 2015. In their configuration, meteorology comes from the Weather Research and Forecast model (WRF, Simpson et al., 2012). The N anthropogenic emissions used were derived from the V6 ECLIPSE inventory (https://iiasa.ac.at/web/home/research/researchPrograms/air/ECLIPSEv6.html) for 2015 with monthly profiles deduced from the EDGAR time series (Crippa et al., 2020) according to Ge et al. (2021). $NO_x$ and VOC emissions from the forest, vegetation fires, lightning, and soil were also included.

As CCMI fluxes are only available until 2014 and files from Ge et al. (2022) are provided for 2015 only, a 2010-2014 climatology has been calculated for CCMI, CAMEO and CEDs simulated N depositions. Ge et al. (2022) do not provide monthly fields; thus, only CCMI, CAMEO and CEDS time series for the same 2010-2014 climatology have been further explored for the seasonality analysis.

**Table 4.** Summary statistics of model comparison (CAMEO and CEDS runs) with measurements for 2015 in East Asia and Southeast Asia (NNDMN and EANET networks), Europe and UK (EMEP/CCC, UK networks), North America (US EPA, AMoN and NAPS). $N$ represents the number of measuring sites. Annual average concentrations and Mean Bias Error (MBE) are given in $\mu g.m^{-3}$.

| Species | Region-Network | $N$ | Mean Obs. | Mean CAMEO | Mean CEDS | MBE CAMEO | MBE CEDS |
|---------|---------------|-----|-----------|------------|-----------|-----------|----------|
| **NH$_3$** | NNDMN | 25 | 10.4 | 4.00 | 3.52 | 6.39 | 6.86 |
| | EANET | 27 | 1.60 | 0.96 | 1.41 | 0.64 | 0.19 |
| | EMEP/CCC | 38 | 0.92 | 0.54 | 1.10 | 0.37 | -0.19 |
| | UK networks | 22 | 1.52 | 0.17 | 0.76 | 1.34 | 0.76 |
| | US AMoN | 31 | 1.22 | 0.77 | 0.59 | 0.45 | 0.63 |
| | NAPS | 7 | 1.41 | 0.49 | 0.43 | 0.92 | 0.98 |
| **NH$_4^+$** | NNDMN | 24 | 8.09 | 1.33 | 1.89 | 6.76 | 6.20 |
| | EANET | 28 | 0.76 | 0.20 | 0.25 | 0.57 | 0.51 |
| | EMEP/CCC | 49 | 0.60 | 0.15 | 0.23 | 0.45 | 0.37 |
| | UK networks | 16 | 0.40 | 0.06 | 0.22 | 0.34 | 0.18 |
| | US EPA | 79 | 0.50 | 0.20 | 0.20 | 0.30 | 0.30 |
| | NAPS | 13 | 0.31 | 0.17 | 0.14 | 0.14 | 0.16 |
| **NO$_2$** | NNDMN | 25 | 24.1 | 21.20 | 18.66 | 2.86 | 7.91 |
| | EANET | 7 | 15.6 | 16.08 | 12.33 | -0.51 | 1.15 |
| | EMEP/CCC | 72 | 4.7 | 5.40 | 4.92 | -0.69 | -0.21 |
| | UK networks | - | - | - | - | - | - |
| | US EPA | 124 | 13.12 | 4.70 | 4.05 | 8.42 | 9.07 |
| | NAPS | 58 | 10.06 | 2.87 | 2.67 | 7.20 | 7.40 |
| **NO$_3^-$** | NNDMN | 25 | 10.20 | 2.29 | 4.27 | 5.54 | 5.93 |
| | EANET | 29 | 1.26 | 0.11 | 0.28 | 1.11 | 0.98 |
| | EMEP/CCC | 50 | 1.12 | 0.21 | 0.38 | 0.91 | 0.74 |
| | UK networks | 15 | 0.91 | 0.002 | 0.39 | 0.90 | 0.52 |
| | US EPA | 155 | 0.60 | 0.26 | 0.22 | 0.62 | 0.38 |
| | NAPS | 13 | 0.38 | 0.32 | 0.18 | 0.48 | 0.20 |
| **SO$_4^{2-}$** | NNDMN | - | - | - | - | - | - |
| | EANET | 29 | 3.27 | 0.81 | 0.77 | 2.46 | 2.50 |
| | EMEP/CCC | 48 | 1.26 | 0.37 | 0.40 | 0.89 | 0.86 |
| | UK networks | 17 | 0.45 | 0.24 | 0.36 | 0.21 | 0.10 |
| | US EPA | 155 | 1.00 | 0.38 | 0.39 | 0.56 | 0.61 |
| | NAPS | 13 | 0.82 | 0.35 | 0.44 | 0.27 | 0.38 |

Global Nr deposition was estimated at 108 and 127 $TgN.yr^{-1}$ over 2010-2014 in the CEDS and CAMEO simulations (land, ~80 %; ocean, ~20 %). CEDS compares well with the 102 and 114 $TgN.yr^{-1}$ estimated from CCMI and Ge et al. (2022) but CAMEO is closer to the 119 $TgN.yr^{-1}$ quantified for 2010 from the recent study from Liu et al. (2022). The ratio of $NH_x$ to total $N_r$ depositions between CCMI, CEDS, and CAMEO show a good agreement, however, EMEP MSC-W depicts a much less important contribution of $NH_x$ to the total $N_r$ depositions all over the world.

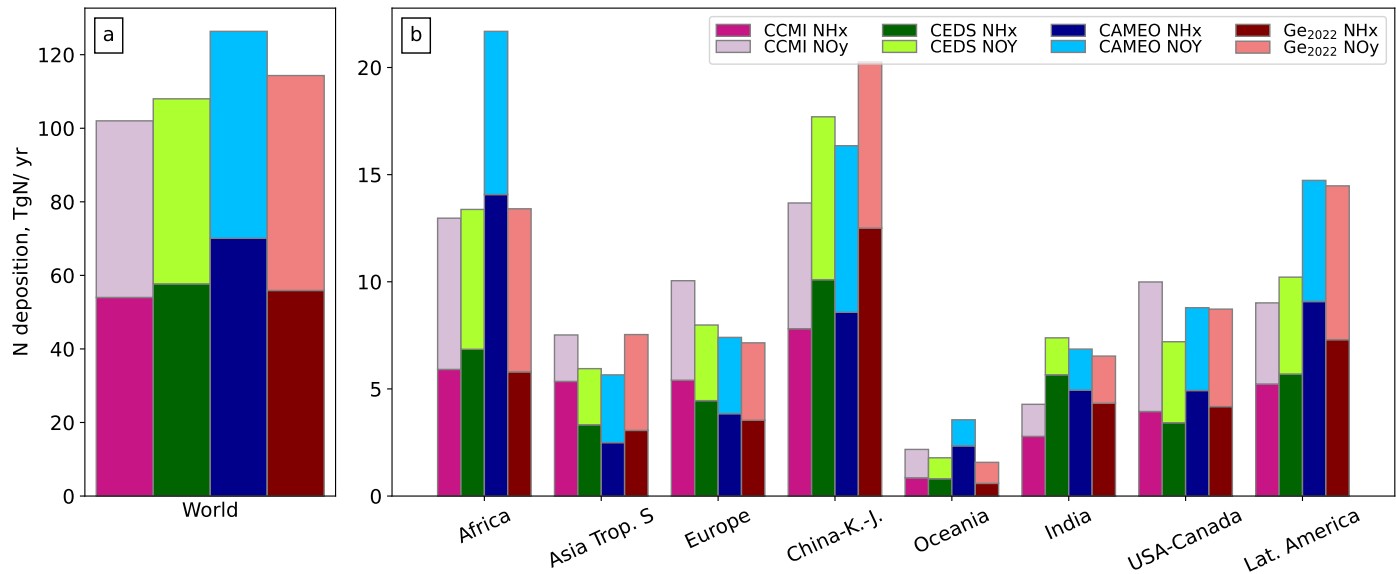

**Figure 4.** Global and regional annual $NH_x$ and $NO_y$ deposition in $TgN.yr^{-1}$ from CCMI for 2010-2014, CEDS and CAMEO simulations for 2010-2014 and by EMEP MSC-W (Ge et al., 2022) for 2015. Note that the global budget account for continents and oceans. China-K-J account for China-Korea-Japan

### 4.3.1 Reduced Nitrogen Deposition

Global surface $NH_x$ depositions reach 65 $TgN.yr^{-1}$ with CAMEO showing a good agreement between CCMI, CEDS and EMEP MSC-W and CAMEO appearing as the highest estimate (Figure 4). This difference between CAMEO and the other estimates is partly explained by the high values over Africa (Figure 5) with a total bugdet of 14 $TgN.yr^{-1}$ which is twice the one estimated in CCMI, CEDS and EMEP MSC-W but also to a smaller extent by higher budgets in Oceania and Latin America. Higher $[NH_3]$ due to enhanced $NH_3$ emissions in CAMEO explain these regional patterns together with no enhanced aerosol ($NH_4^+$, $NO_3^-$, $SO_4^{2-}$) formation because of low $NO_x$ conditions (Figure 6 and 7). It means that even though there is more $NH_3$, it remains in its gaseous phase and the deposition pathway is favored in these regions when CAMEO emissions are used. As mentioned previously, Vira et al. (2019) estimated high agricultural $NH_3$ emissions over Africa with the FAN v2 model when compared to the literature. In a recent evaluation work using observations from the INDAAF network, they show an overestimation of their $NH_x$ wet deposition flux of around 10 % (Vira et al., 2022). We also compared our simulated $NH_x$ wet deposition fluxes from two grid cells corresponding to stations from the INDAAF network situated in western Africa (see Figure S13 in the Supplementary Material for the exact locations). CAMEO simulation compares much better than CEDS to the observed $NH_4^+$ wet deposition, especially at the Katibougou station where a clear seasonal cycle with a similar peak in summer is represented (see Figure S14 in the Supplementary Material).

Regarding the other regions, $NH_x$ deposition from LMDZ-INCA (both CEDS and CAMEO) and EMEP MSC-W reach values up to 3000 $mgN.m^{-2}.yr^{-1}$ in India and China while CCMI fluxes do not exceed 1900 $mgN.m^{-2}.yr^{-1}$ (Figure 5). Same patterns are observable over central Africa, Latin America, and the US where CCMI $NH_x$ deposition (maximum between 500 and 1000 $mgN.m^{-2}.yr^{-1}$) are lower than LMDZ-INCA and EMEP MSC-W deposition rates (maximum between 800 and 1900 $mgN.m^{-2}.yr^{-1}$). Over these regions, CAMEO simulation depicts much higher deposition fluxes which is explained by higher emissions prescribed in this run than in CEDS (see Figure S2 in the Supplementary Material). However, in south-eastern Asia CCMI deposition reach 7000 $mgN.m^{-2}.yr^{-1}$ while in LMDZ-INCA and EMEP MSC-W maximum value is around 1400 $mgN.m^{-2}.yr^{-1}$.

There are important disagreements in the $NH_x$ deposition seasonal cycle between LMDZ-INCA simulations and CCMI in almost all the regions (see Figure S15 in the Supplementary Material). CEDS $NH_x$ depositions variations are well correlated with the $NH_3$ variations of the CEDS emission inventory used as forcing file for the flux calculation in the model. $NH_3$ emissions from the CEDS inventory describe two peaks: an important one in May and another smaller in September which are clearly observable in the CEDS depositions. CAMEO $NH_x$ depositions describe a pattern that differs from one region to another but with a peak in summer for most regions. The summer peak is also reflected in the emission seasonality as analyzed in Beaudor et al. (2023a). Both dry and wet $NH_x$ depositions from LMDZ-INCA have the same seasonal cycle except in East Africa, India, and Latin America, wherein these regions, wet deposition is largely dominant. Aside from these regions, wet and dry depositions have similar contributions to the total depositions. In their study, Ge et al. (2022) found a higher contribution of dry deposition in almost all the continental regions. In the CCMI depositions, except for South-East Asia, variations over the year are weak, with no clear seasonal pattern.

It is worth pointing out that in LMDZ-INCA model no pH adjustment is considered for the $NH_3$ Henry's law constant, while it appears to be important in controlling wet $NH_3$ deposition. Bian et al. (2017) investigated the impact of pH-dependent $NH_3$ wet deposition on atmospheric $NH_3$ and associated nitrogen species with the Global modelling Initiative (GMI) and found that without pH correction, $NH_3$ wet deposition decreases significantly (from 17.5 to 1.1 $TgNyr^{-1}$). Because $NH_3$ deposition has an impact on its atmospheric lifetime and, therefore, is an important factor in the ammonium-nitrate system, it would be interesting also to evaluate the sensitivity of the effective $NH_3$ Henry's law constant and the consideration of the pH correction in LMDZ-INCA.

### 4.3.2 Nitrogen Oxide Deposition

$NO_y$ deposition patterns over Africa, India and China are consistent between the four estimates, especially for CEDS and CAMEO simulations and EMEP MSC-W (Figure 5). There are no major differences between CEDS and CAMEO simulated $NO_y$ deposition fluxes since $NH_3$ emissions have only a small impact on nitrate deposition. However, CCMI fluxes in the US and Europe (1300 and 900 $mgN.m^{-2}.yr^{-1}$) are more important than LMDZ-INCA (600 and 500 $mgN.m^{-2}.yr^{-1}$ ) and EMEP MSC-W (900 and 700 $mgN.m^{-2}.yr^{-1}$) depositions. On the opposite, in Latin America, CCMI depositions are the lowest. Global $NO_y$ deposition budgets from CCMI and LMDZ-INCA vary between 39 and 43 $TgN.yr^{-1}$ while EMEP MSC-W estimate is 47 $TgN.yr^{-1}$ (Figure 4). Similarly as for $NH_x$, China, Africa and Latin America are the most important contributors

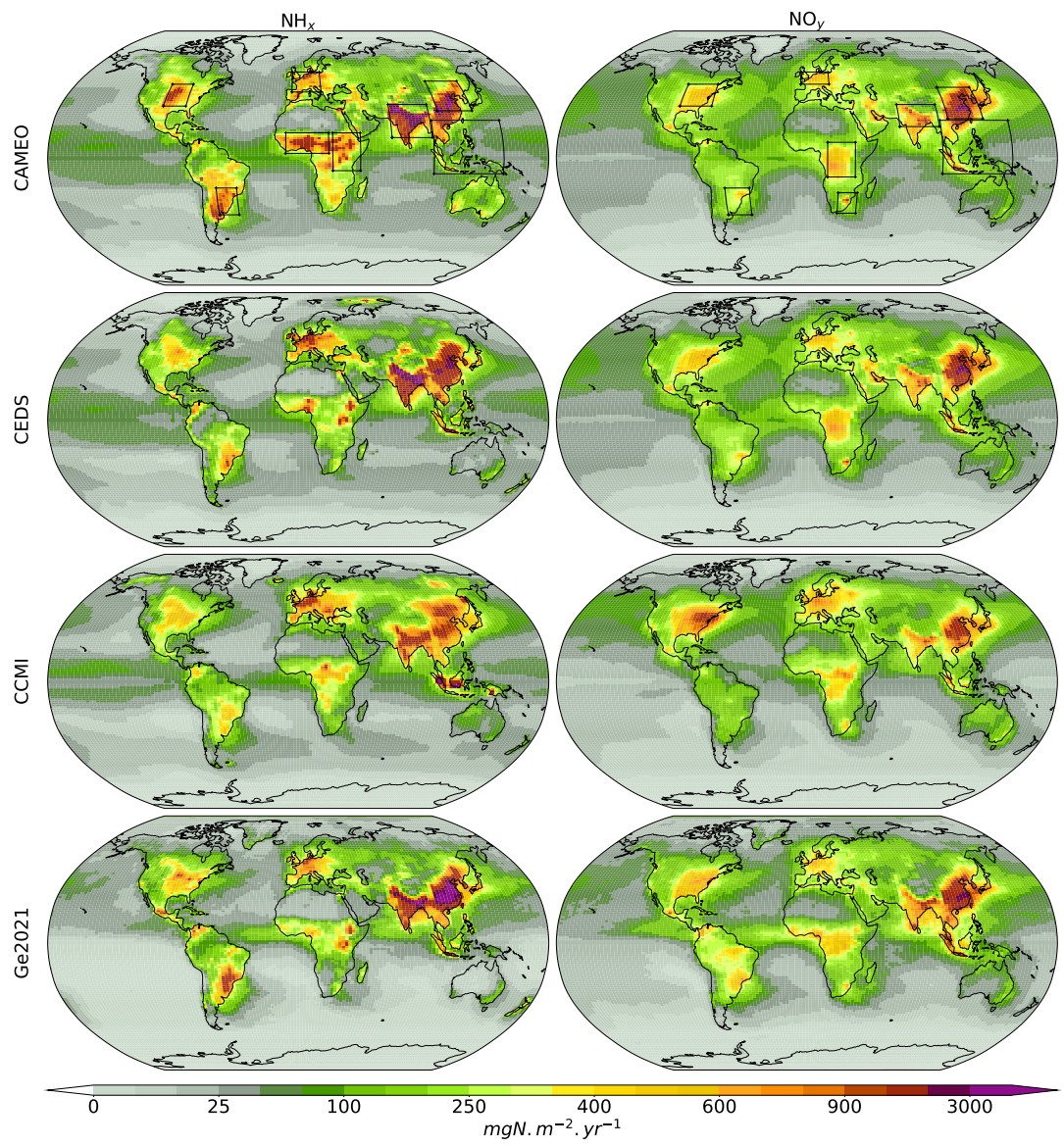

**Figure 5.** Annual mean total (dry + wet) $NH_x$ (first column) and $NO_y$ (second column) deposition for present-day conditions. The first row shows the N deposition fluxes calculated from the most recent CMIP exercise (IGAC/SPARC Chemistry–Climate Model Initiative (CCMI; Eyring et al., 2013, 2010-2014 climatology), second and third rows correspond to LMDZ-INCA simulations where CEDS and CAMEO emissions are prescribed respectively (2010-2014 climatology), last row display recent modelling results from Ge et al. (2022) using the EMEP model (2015). The black boxes in (a) and (b) delimit the regional bounds for the seasonal variability analysis. $(mgNm^{-2}yr^{-1})$.

to the global $NO_y$ depositions budget in EMEP MSC-W and LMDZ-INCA estimates (about 47 %). CCMI estimates higher

$NO_y$ depositions in North America than in Latin America. The three regions Africa, North America and China account for half of the CCMI budget.

CCMI and LMDZ-INCA seasonal cycles of $NO_y$ deposition are very well correlated together (see Figure S16 in the Supplementary Material). On the contrary of $NH_x$ which are primarily driven by only a few sources of emissions (mainly agricultural $NH_3$), $NO_y$ are the results of $NO_x$ sources and reactions involving several nitrate species. However, $NO_x$ emissions mainly come from the energy, transportation, and industrial sectors (Hoesly et al., 2018; McDuffie et al., 2020) whose seasonal cycles are better-known than the agricultural one. Similarly as $NH_x$, $NO_y$ wet fluxes are contributing the most to the total depositions in most regions except in South Africa, Europe, and India where dry deposition dominates during several months. The main differences between CEDS and CAMEO are observed in the wet deposition in winter in Latin America and South Africa but also in India in summer and the whole year in southern-eastern Asia. It indicates that $NH_3$ emissions rather impact wet $NO_y$ deposition fluxes mostly when a direct loss through scavenging occurs such as during the monsoon season in India.

## 5  Impact of future emissions

### 5.1  Impact on atmospheric composition

Considering future CAMEO emissions under SSP5-8.5 and SSP4-3.4 in LMDZ-INCA highlights the range of possible impact of future $NH_3$ emissions on N species and aerosol. Both scenarios of emissions lead to a global increase of the N species and aerosol burdens which also vary according to the $NO_x$ and $SO_2$ emission trends (Table 5).

Relative to the present-day level with CAMEO, $NH_3$ burdens are increased by 59% in CAMEO[585], 111% in both CAMEO[434] and CAMEO[434-370], and by 235% in CAMEO[434-126] which is considered as the 'higher' scenario regarding $NO_x$ and $SO_2$ emissions. In CAMEO[434-126], burden of $NH_4^+$ (0.55 $TgNyr^{-1}$) is similar to the value of $NH_3$ (0.58 $TgNyr^{-1}$) while in both CAMEO[434] and CAMEO[434-370], $NH_4^+$ burden (~0.72 $TgNyr^{-1}$) is about twice the one of $NH_3$. Regarding the $HNO_3$ burden, it is similar to present-day value in CAMEO[434] and CAMEO[434-370] (~0.74 $TgNyr^{-1}$), but much smaller in CAMEO[434-126] (Table 5). It is explained by the lower $NO_x$ emissions used in the later simulation compared to the other simulations (9.2 against 39 $TgNyr^{-1}$) (see Table 1). However, the $NO_3^-$ burden is within the same range of values for the three future simulations (0.34-0.45 $TgNyr^{-1}$) which can be twice as high as in the historical CAMEO run.

The impact of future CAMEO emissions under SSP4-3.4 on the distributions of $NH_3$, $NO_2$ and $HNO_3$ surface concentrations are presented in Figure 6. Compared to the historical CAMEO simulation, all CAMEO[SSP4-3.4-i] depicts large increases in [$NH_3$] of about 5-10 $\mu g.m^{-3}$ (>100%) over northern Africa, northern India and eastern China (subplots C-E) corresponding to the regions experiencing the most important increases in the agricultural $NH_3$ emissions ($> 4$ $gN.m^{-2}.yr^{-1}$, see Figure S2 in the Supplementary Material). As only negligible differences in the other future anthropogenic $NH_3$ emissions are notable, the impact of the CAMEO[SSP4-3.4] emissions on [$NH_3$] is similar for the three simulations. The impact on [$NO_2$] and [$HNO_3$] is much more contrasted between the simulations. In CAMEO[434], as the $NO_x$ emissions are kept at their present-day level, no impact is observable. However, in CAMEO[434-126] and CAMEO[434-370] the $NO_x$ emissions vary from the historical levels: in CAMEO[434-126], the emissions are much lower all over the globe while in CAMEO[434-370],

500 emissions are largely reduced in the most developed countries (Europe, China, and the US) and increased in the Southern

Hemisphere along with India and the Gulf States. It leads to a decrease of around 60 to 80% (5 to 12 $\mu$g.m$^{-3}$) in $[NO_2]$ and (1

to 3 $\mu$g.m$^{-3}$) in $[HNO_3]$ over China, Europe and the US in CAMEO[434-126] (subplots I and N). In CAMEO[434-370], the

impact of the future emissions on both $[NO_2]$ and $[HNO_3]$ also follows $NO_x$ emission trends with most important increases

located in India (15 and 8 $\mu$g.m$^{-3}$, respectively) and smaller decrease situated in Europe, China, and the US (subplots J and

505 O).

  As a result of these changes in nitrate precursor surface concentrations, nitrate and sulfate particles are expected to vary

significantly in the future. In order to understand future patterns in the nitrate and sulfate aerosol formations, the state of

ammonia neutralization of the sulfuric and nitric acids is shown for different pressure levels in Figure 8.

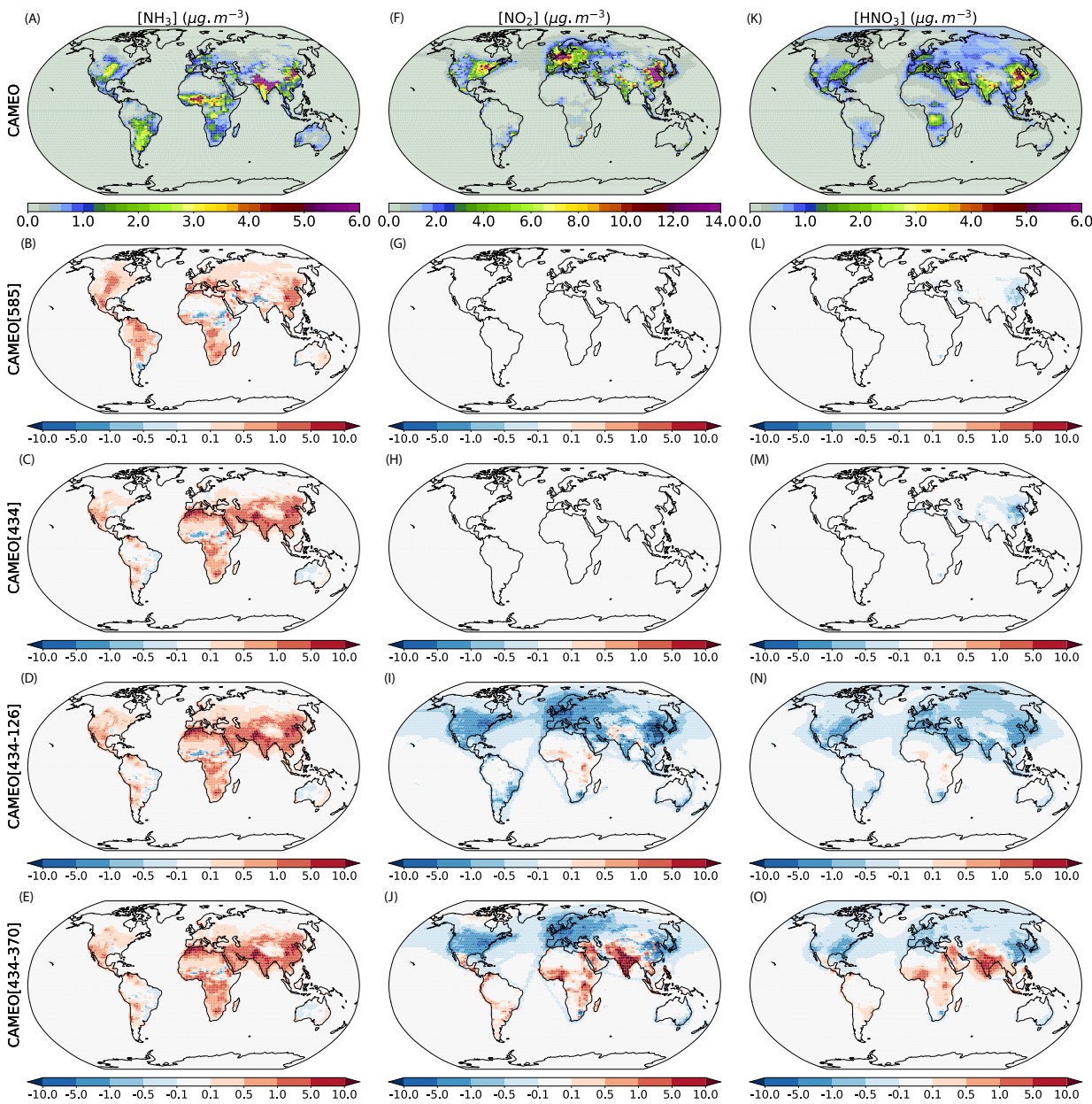

**Figure 6.** Mean annual surface concentrations of $NH_3$, $NO_2$ and $HNO_3$ simulated in the CAMEO simulation (1st row; over 2004-2014) and the anomalies between the CAMEO[SSPi] and CAMEO simulations ([SSPi]:585, 434, 434-126 and 434-370 in rows 2-5; over 2090-2100) ($\mu$g.m$^{-3}$).

**Table 5.** Tropospheric burden and deposition losses (TgNyr$^{-1}$) of ammonia (NH$_3$), ammonium particles ( NH$_4^+$), nitric acid (HNO$_3$) and fine nitrate particles (NO$_3^-$) for the present-day (2004-2014) and future (2090-2100) simulations. N$_2$O production through NH$_3$ gas phase loss (TgNyr$^{-1}$) is also included. Please note that total emissions include biomass burning (4.2 TgNyr$^{-1}$).

| Simulation | Bugdet (TgNyr$^{-1}$) | NH$_3$ | NH$_4^+$ | HNO$_3$ | NO$_3^-$ |
|---|---|---|---|---|---|
| **Present-day (2004-2014)** | | | | | |
| CEDS | Burden | 0.10 | 0.33 | 0.79 | 0.08 |
| | Sources (emissions) | 58.2 | | | |
| | Wet deposition | 17.9 | 16.6 | 29.2 | 9.6 |
| | Dry deposition | 20.1 | 1.53 | 61.4 | 0.87 |
| | N$_2$O prod. | 0.75 | | | |
| | **NH$_4^+$ formation** | 18.0 | | | |
| CAMEO | Burden | 0.17 | 0.47 | 0.79 | 0.22 |
| | Sources (emissions) | 68.8 | | | |
| | Wet deposition | 22.4 | 17.8 | 28.7 | 10.0 |
| | Dry deposition | 24.8 | 1.48 | 62.3 | 0.78 |
| | N$_2$O prod. | 1.01 | | | |
| | **NH$_4^+$ formation** | 18.8 | | | |
| **Future (2090-2100)** | | | | | |
| CAMEO[585] | Burden | 0.28 | 0.60 | 0.77 | 0.34 |
| | Sources (emissions) | 88.1 | | | |
| | Wet deposition | 30.6 | 20.4 | 27.5 | 11.3 |
| | Dry deposition | 33.0 | 1.63 | 61.0 | 0.86 |
| | N$_2$O prod. | 1.29 | | | |
| | **NH$_4^+$ formation** | 21.3 | | | |
| CAMEO[434] | Burden | 0.36 | 0.65 | 0.75 | 0.38 |
| | Sources (emissions) | 102 | | | |
| | Wet deposition | 36.0 | 21.5 | 27.0 | 11.8 |
| | Dry deposition | 39.4 | 1.74 | 60.0 | 0.91 |
| | N$_2$O prod. | 1.87 | | | |
| | **NH$_4^+$ formation** | 22.4 | | | |
| CAMEO[434-126] | Burden | 0.58 | 0.55 | 0.45 | 0.42 |
| | Sources (emissions) | 103 | | | |
| | Wet deposition | 44.5 | 11.3 | 13.6 | 7.5 |
| | Dry deposition | 43.5 | 0.43 | 22.2 | 0.32 |
| | N$_2$O prod. | 1.59 | | | |
| | **NH$_4^+$ formation** | 11.7 | | | |
| CAMEO[434-370] | Burden | 0.35 | 0.72 | 0.74 | 0.46 |
| | Sources (emissions) | 109 | | | |
| | Wet deposition | 39.1 | 21.8 | 24.8 | 13.3 |
| | Dry deposition | 42.7 | 1.88 | 61.7 | 1.16 |
| | N$_2$O prod. | 2.36 | | | |
| | **NH$_4^+$ formation** | 23.6 | | | |

Four chemical domains can be derived from the simulated relative abundances of $NH_3$, $NH_4^+$, $NO_3^-$, $HNO_3$ and $SO_4^{2-}$ (Metzger et al., 2002; Xu and Penner, 2012; Hauglustaine et al., 2014; Paulot et al., 2016; Ge et al., 2022) . To gain a better understanding of the behavior of ammonia and its persistence in the atmosphere under future scenarios, we have selected different pressure levels, including surface level, 900 hPa, and 500 hPa. First, we define the total molar concentrations of sulfate ($T_S$, including all forms of $SO_4^{2-}$ as $H_2SO_4$, $NH_4HSO_4$, $(NH_4)_3H(SO_4)_2$, and $(NH_4)2SO_4$)), nitrate ( $T_N$), ammonia ($T_A$) and ammonia needed to fully neutralize the sulfate ($T_{A-free}$) :

$$T_S = [SO_4^{2-}], \tag{3}$$

$$T_N = [NO_3^-] + [HNO_3], \tag{4}$$

$$T_A = [NH_3] + [NH_4^+], \tag{5}$$

$$T_{A-free} = T_A - 2 \times T_s \tag{6}$$

The four chemical domains are defined as : sulfate very rich ($T_A$ / $T_S$ < 1), sulfate rich (1 < $T_A$ / $T_S$ < 2), nitrate rich (0 < $T_{A-free}$/$T_N$ < 1) and ammonia rich ($T_{A-free}$/$T_N$ > 1). When $T_{A-free}$ / $T_N$ > 1, sufficient ammonia remains to react with nitrate to form $NH_4NO_3$. The resulting calculated domains are illustrated in Figure 8. In order to focus on the most important anthropogenic sources, we imposed a threshold on the secondary inorganic aerosol (SIA) concentration which is set as ($NH_4^+$ + $NO_3^-$ + $SO_4^{2-}$) $\geq$ 0.5 $\mu$g.m$^{-3}$. This threshold has been arbitrarily chosen similarly as in Ge et al. (2022). In the rich and very rich domains $SO_4^{2-}$ (yellow and blue areas in Figure 8), not all sulfuric acid is neutralized ($SO_4^{2-}$ not only exists as in $NH_4SO_4^2$). This is the case, for instance, at the surface, the regions where high $SO_2$ sources are collocated with low $NH_3$ sources. In the CAMEO simulation, these areas are located in the Sahara, northern Russia, and along the coastlines of Asia, the western US, and the Arabian Sea. These regions expand across the continents as we move away from the surface (at 900 hPa). The decrease in $NH_3$ can be attributed to its rapid transformation into $NH_4^+$ at pressures of 900 hPa and 500 hPa. In the green and red areas, all sulfuric acid has been neutralized and excess $NH_3$ is available to react with $HNO_3$ to form $NH_4NO_3$. Most continental regions characterized by important anthropogenic activities are under these regimes at the surface. Considered nitrate-rich, these regions are rather continental and remote from the main $NH_3$ hotspot as in the Middle East for example. They are generally characterized by high $NO_x$ emissions or large transport of $NO_x$ and relatively rapid deposition of $NH_x$. Finally, red areas correspond to regions where ammonia prevails and the availability of nitrate limits the formation of $NH_4NO_3$. It is the most dominant regime on the surface, covering most continents and especially places with the most intensive agricultural activities (Asia, Europe, southern and northeastern Africa, and the US).

By analyzing the change in the ammonia neutralization state of sulfuric and nitric acids between the different simulations in Figure 8, we investigate the impact of future emissions on the other surface aerosol concentrations shown in Figure 7.

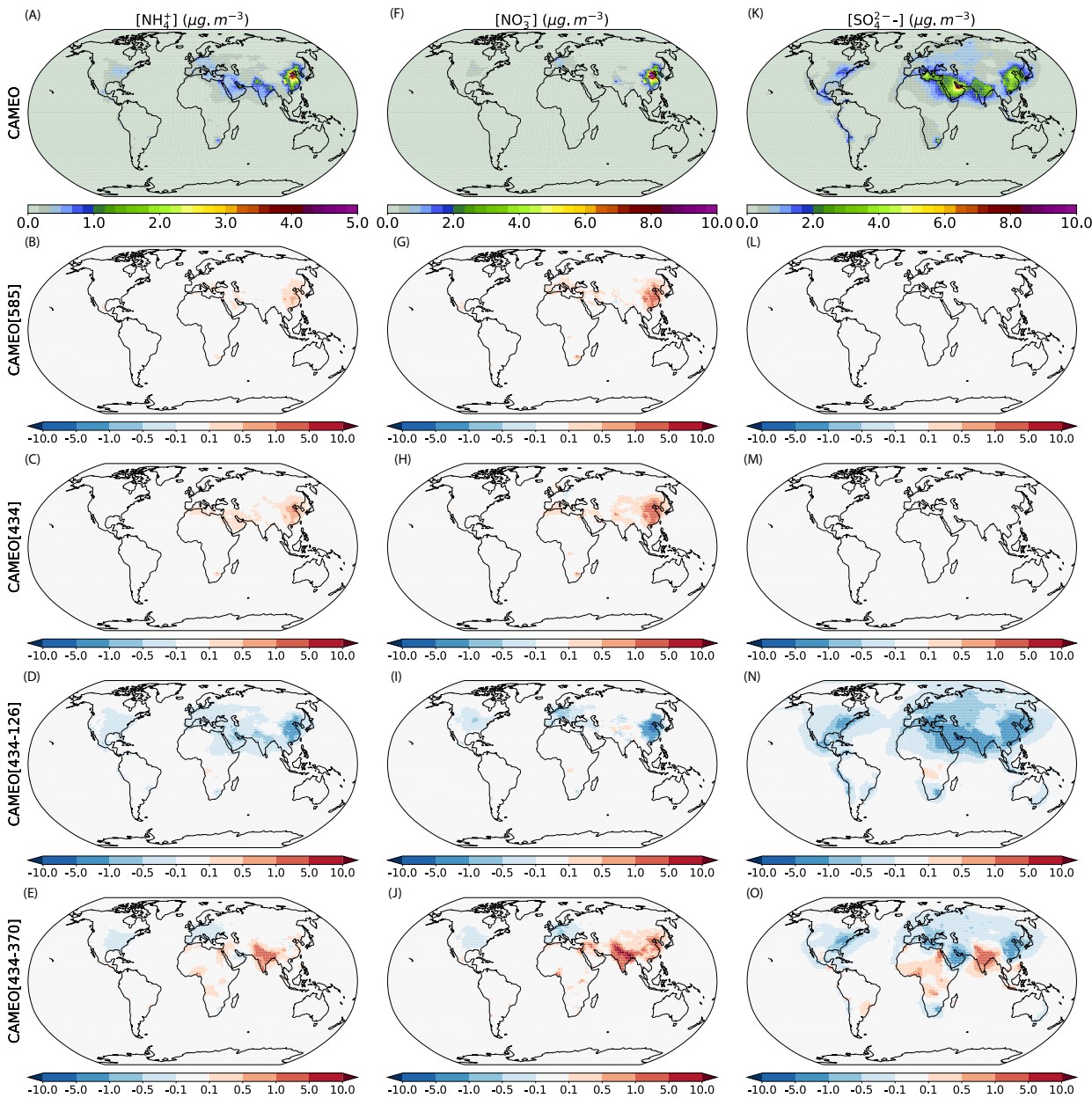

**Figure 7.** Mean annual surface concentrations of $NH_4^+$, $NO_3^-$ and $SO_4^{2-}$ simulated in the CAMEO simulation (1st row; over 2004-2014) and the anomalies between the CAMEO[SSPi] and CAMEO simulations ([SSPi]:585, 434, 434-126 and 434-370 in rows 2-5; over 2090-2100) ($\mu$g.m$^{-3}$).

Figure 7 highlights only small positive changes in China in the $[NO_3^-]$ ($< 2$ $\mu$g.m$^{-3}$) are observable in CAMEO[434] and CAMEO[585] compared to the CAMEO simulation. In this region, compared to the CAMEO simulation, there is a noticeable

expansion of the nitrate-rich and ammonia-rich domains at 900 hPa which is explained by relatively higher $[NH_3]$ and a stronger limitation by $HNO_3$ availability (Figure 6). It is a result of much higher $NH_3$ emissions and no change in other emissions in this

scenario. On another hand, CAMEO[434-126] depicts important negative anomalies of $[NH_4^+]$, $[NO_3^-]$ and $[SO_4^{2-}]$ especially in China (> 4 $\mu g.m^{-3}$, equivalent to 60-80% Figure 7, subplots D, I and N). In China, the ammonia-rich conditions observed in CAMEO are expanded (less fine PM are formed) as we reach 900 hPa, highlighting the abundance of gaseous ammonia (Figure 8). In CAMEO[434-126], even though more $NH_3$ is emitted, important reductions in $NO_x$ and $SO_2$ emissions are notable (see Figure S2 in the Supplementary Material). It means that almost no acids are available to react with ammonia and therefore it

is not converted into ammonium and its gaseous-form concentration is enhanced. A similar situation arose attention in China in the last decades, where an unexpected increase in the $[NH_3]$ has been observed after strong regulations in $NO_x$ and $SO_2$ emissions and no change in the $NH_3$ emissions (Lachatre et al., 2019). In line with the later study, the effect of the simultaneous reductions in $NO_x$ and $SO_2$ emissions in CAMEO[434-126] is even stronger on $[NH_3]$ due to the combined increase in $NH_3$ emissions mainly explained by the significant increase in the use of synthetic fertilizers in China (+30 $TgNyr^{-1}$ compared

to historical application). This is also confirmed by comparing $[NH_3]$ from CAMEO[434-126] and CAMEO[434] where $NH_3$ emissions are identical but a slightly stronger impact on the concentrations is highlighted for instance in China and India (Figure 6, subplots C and D). It is notable that other combined factors have been shown to significantly contribute to the increased $NH_3$ levels in China. For instance, in Warner et al., 2017, the authors suggest that the rise in ammonia levels in China between 2003 and 2015 can be attributed to sulfur controls, greater fertilizer application, and rising local temperatures. The present study does

not explore the impact of meteorological factors, as it focuses on the isolated impact of human-related ammonia emissions. $[SO_4^{2-}]$ in CAMEO[434-126] also decreases considerably over the Arabian Peninsula, India, and the western US (of about 2-4 $\mu g.m^{-3}$, equivalent to 80%) which is a direct consequence of the $SO_2$ regulations in scenario SSP1-2.6 (Figure 7). The shift in the emissions in CAMEO[434-370] compared to CAMEO for the present-day highlights positive anomalies in the N inorganic aerosols concentrations over northern India (around + 3 $\mu g.m^{-3}$ in $[NH_4^+]$ and + 5 $\mu g.m^{-3}$ in $[NO_3^-]$, Figure 7, subplots E

and J). The enhanced aerosol formation in this region is due to the important increase in $NH_3$ emissions along with the highest $NO_x$ and $SO_2$ emissions. The formation of the secondary inorganic aerosol is very sensitive to the $NO_x$ and $SO_2$ emissions, as demonstrated by the distinct responses between CAMEO[434], CAMEO[434-126], CAMEO[434-370] while $NH_3$ levels are similar in the three simulations (Figure 7). Interesting patterns also arise in CAMEO[434-370] in regions situated in Africa which are characterized by a very rich $NH_3$ domain not observable in the other simulations (Figure 8). Contrary to India where

both $NO_3^-$ and $SO_4^{2-}$ formations are favored, significant increases in $[SO_4^{2-}]$ only are observed in Africa. It is likely that in Africa, $HNO_3$ availability is still limited to react with the excess of ammonia despite the small increases in $NO_x$ emissions under SSP3-7.0. Over Europe and the US, a notable decrease of $[SO_4^{2-}]$ (around - 1 $\mu g.m^{-3}$) is observed. It is a direct consequence of lower levels of $NO_x$ and $SO_2$ emissions along with constant levels of $NH_3$ leading to less ammonium-related aerosol formation as shown in Figure 7 (subplots E,J and O). Finally, the evolution of the neutralization state by ammonia

is also notable throughout the vertical profile and is particularly distinctly influenced by $NO_x$ and $SO_2$ emission levels. In CAMEO[434-370], the ammonia-rich state remains predominant not only at the surface but also at 900 hPa likely enhanced by convection that transports the excess of ground ammonia to more elevated layers. Additionally, at this altitude, we note

the emergence of coastal nitrate-rich regions in West Africa, India and East Asia. By moving further from the surface to the
upper troposphere, nitrate-rich regions expand across Africa, the Middle East and Asia indicating non negligible impacts on
tropospheric chemistry (Figure 8).

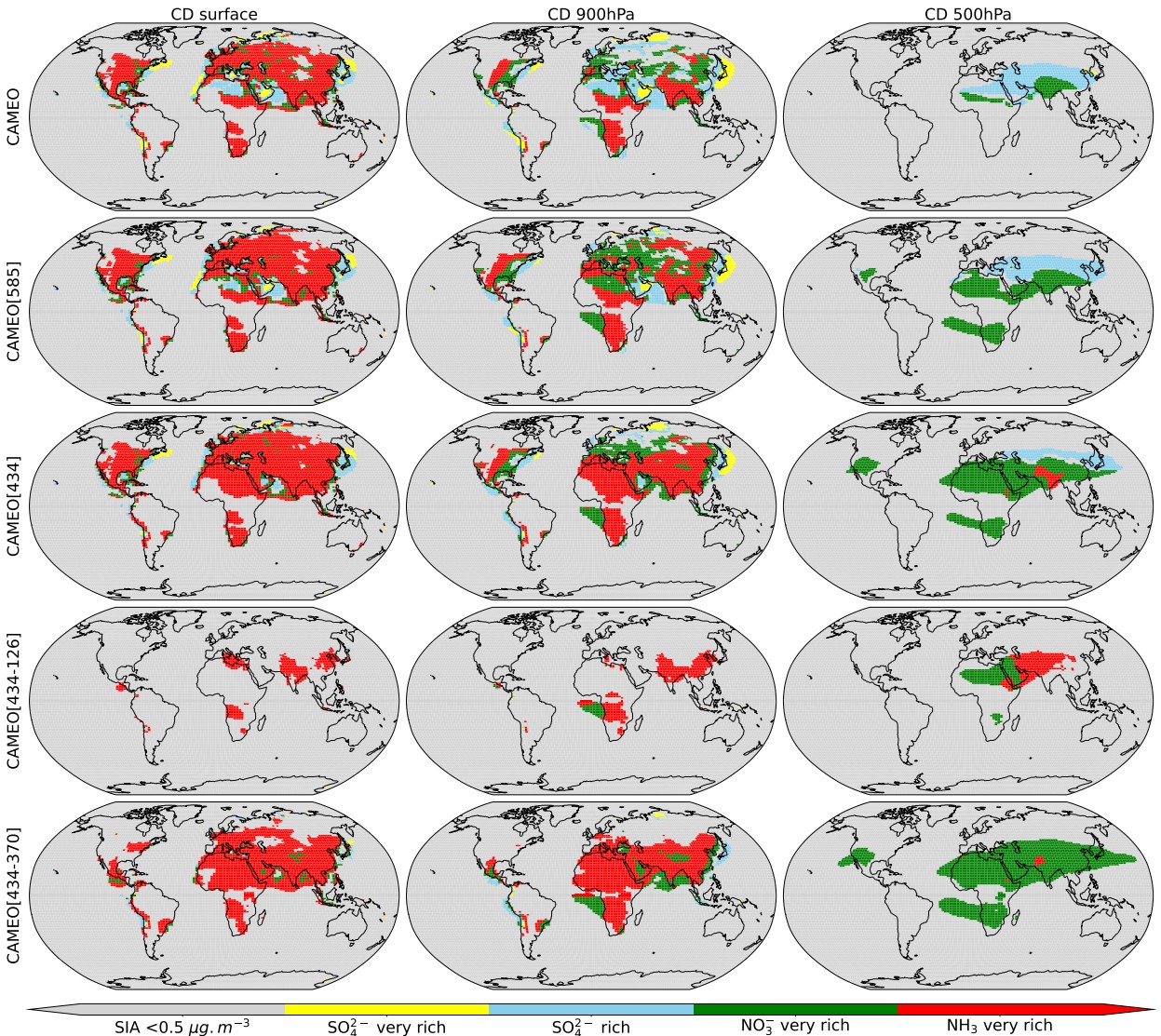

**Figure 8.** The state of ammonia neutralization of sulfuric and nitric acids for areas where secondary inorganic aerosol concentration in the
fine particle fraction (PM2.5) is $> 0.5$ $\mu$g.m$^{-3}$ calculated from the different simulations (averages done over 2004-2014 for CAMEO and
over 2090-2100 for CAMEO[SSPi]) at the surface, 900 hPa and 500 hPa (first, second and third columns). The four chemical domains are
defined as : sulfate very rich ($T_A$ / $T_S$ < 1, yellow area), sulfate rich (1 < $T_A$ / $T_S$ < 2, blue areas), nitrate rich (0 < $T_{A-free}$/$T_N$ < 1, green
areas) and ammonia rich ($T_{A-free}$/$T_N$ > 1, red areas).

## 5.2 Impact on nitrogen surface deposition

The impact of the future emissions on the $NH_x$ and $NO_y$ surface deposition is depicted in Figure 9. Independently of the $NO_x$ and $SO_2$ scenario, $NH_x$ deposition increases significantly. Total $NH_x$ deposition is estimated to increase from 65 $TgNyr^{-1}$ to 98-105 $TgNyr^{-1}$ with the lowest and highest value reached in respectively, CAMEO[434] and CAMEO[434-370] (Table 5).

Regionally, increases in $NH_x$ deposition can reach 2000 $mgN.m^{-2}.yr^{-1}$ (Figure 9, subplots C to E) and are mostly located in areas where $NH_3$ emissions are enhanced under SSP4-3.4 (northern Africa, India and China). This large increase is mostly due to enhanced total $NH_3$ deposition while $NH_4^+$ deposition either increases slightly (around + 4 $TgNyr^{-1}$) or even decreases, for example, in CAMEO[434-126] (-7 $TgNyr^{-1}$). In this latter case, $NH_4^+$ deposition decreases as a result of a shift in the chemical regime where most of $NH_3$ does not neutralize sulfuric and nitric acids and remains in its gaseous phase due to lower

$[NO_x]$ and $[SO_2]$. Therefore, in parallel to less $NH_4^+$ deposition in CAMEO[434-126], more $NH_3$ deposition occurs. Regarding the future $NO_y$ deposition, the results are more contrasted between the different simulations. In CAMEO[585], CAMEO[434] and CAMEO[434-370] simulations, total $NO_y$ deposition keeps a constant value close to the present-day simulation (~100 $TgNyr^{-1}$) because of a similar decrease in $HNO_3$ deposition and increase in $NO_3^-$ deposition (2-4 $TgNyr^{-1}$). Compared to CAMEO, the total $NO_y$ deposition is reduced by more than half in CAMEO[434-126] (-58 $TgNyr^{-1}$) as a result of a decrease

in both $NO_3^-$ and $HNO_3$ depositions.

There are minimal spatial differences (<5%) in the deposition of $NO_y$ between CAMEO and CAMEO[434] (and CAMEO[585]), as constant $NO_x$ emissions lead to a balancing effect, resulting in decreased $HNO_3$ deposition and increased $NO_3^-$ deposition, especially in China (see Figure 9, subplots G and H). Under the low $NO_x$ scenario (CAMEO[434-126]), $NO_y$ deposition decreases all over the world, and the highest anomalies are located in China (< -800 $mgN.m^{-2}.yr^{-1}$). When future SSP3-7.0

emissions of $NO_x$ are prescribed, the impact on $NO_y$ deposition follows a similar pattern as $NO_x$ emissions. Compared to CAMEO, the deposition of $NO_y$ in CAMEO [434-370] is significantly increased in India (> 800 $mgN.m^{-2}.yr^{-1}$) and to a lesser extent in Africa and the Arabian Peninsula (~300 $mgN.m^{-2}.yr^{-1}$). Over the most developed countries, $NO_y$ deposition depicts negative anomalies of around 300 $mgN.m^{-2}.yr^{-1}$ (Figure 9, subplot J).

## 5.3 Associated radiative forcing

The impact of the different future emissions on the total nitrate, and sulfate AOD at 550 nm is presented in Figure 10 and Table 6. The global increase in the nitrate AOD due to future $NH_3$ emissions from CAMEO ranges from 50% to 100% for CAMEO[434-370]. As mentioned in the previous section, considering the future SSP4-3.4 emissions from CAMEO while keeping other emissions at their present-day levels (CAMEO[434]) positively impacts nitrate aerosol formation. This results in an increase in the total Aerosol Optical Depth (AOD) ranging from 0.01 for most land and ocean areas to 0.05 over China.

While sulfate AOD contributed the most to the total AOD with present-day level emissions, nitrate AOD becomes very much important in CAMEO[434]. When considering strict regulations in the $NO_x$ and $SO_2$ emissions as in CAMEO[434-126], the impact on the AOD is significant for the sulfate aerosol depth where the decrease can reach -0.15 over China, for instance, compared to the CAMEO simulation. The positive impact on the nitrate AOD in this simulation is of the same range as the

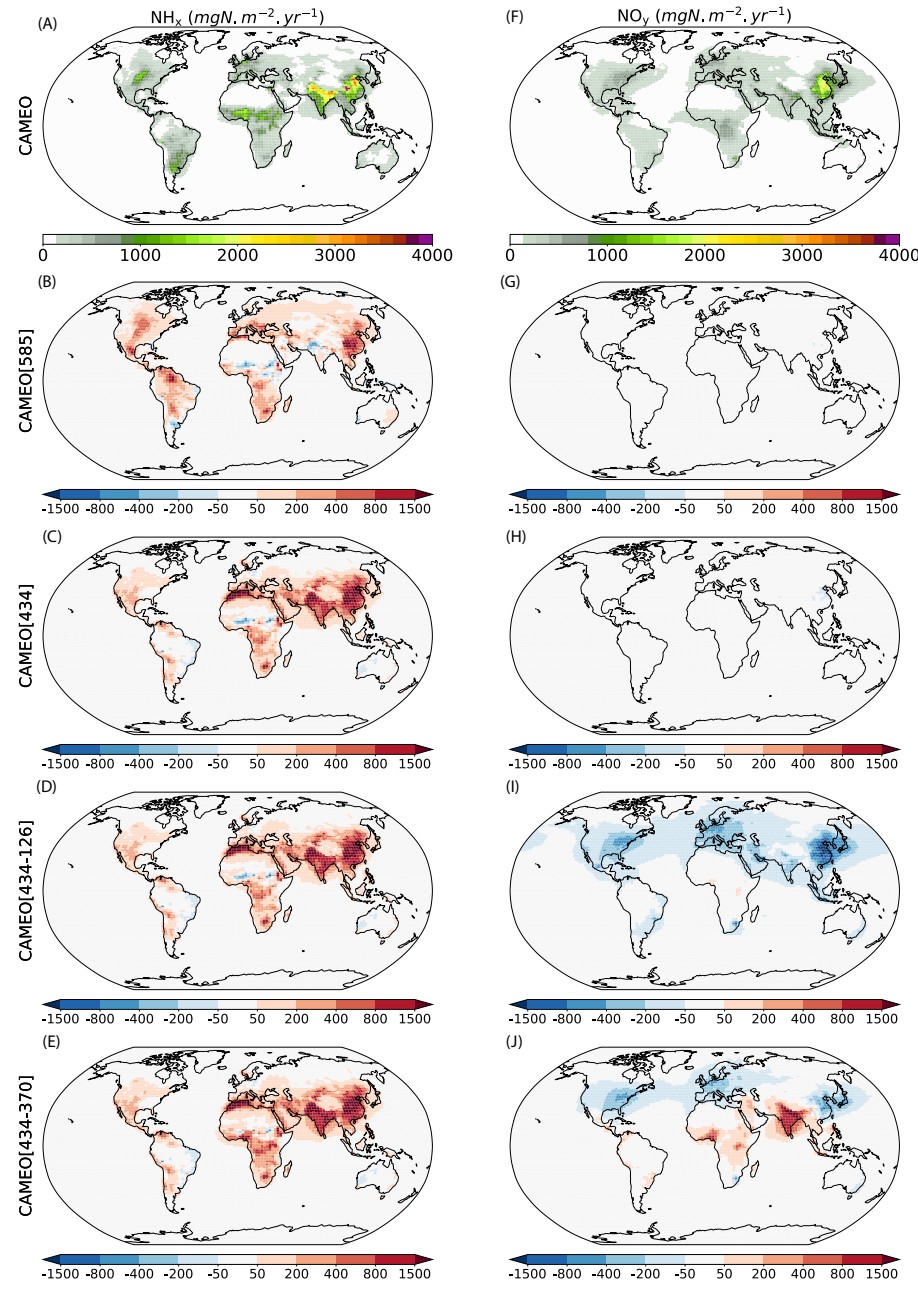

**Figure 9.** Mean annual surface depositions of $NH_x$ and $NO_y$ simulated in the CAMEO simulation (1st row; over 2004-2014) and the anomalies between the CAMEO[SSPi] and CAMEO simulations ([SSPi]:585, 434, 434-126 and 434-370 in rows 2-5; over 2090-2100) ($mgN.m^{-2}.yr^{-1}$).

one in CAMEO[434] except in China where the decrease in $NO_x$ emissions leads to a decrease in the AOD of around 0.03.

It is interesting to note that, in CAMEO[434-126], the decrease of $SO_2$ emissions largely counterbalances the $NO_x$ emission reductions, as $NH_3$ is reacting with the sulfate in priority to form ammonium sulfate aerosols. Except in tropical Africa where there is almost no impact of future emissions on the sulfate AOD, most of the land regions depict negative anomalies in the total AOD. Finally, the impact of future $NO_x$ and $SO_2$ emissions from SSP3-7.0 combined with $NH_3$ emissions from SSP4-3.4 leads to a strong increase in the total AOD over Africa and India (of around 0.10 and 0.15, respectively) and a slight decrease over western US and Europe (around 0.03). The highest increases in the total AOD are explained by large positive anomalies in the nitrate and sulfate AOD (the impact on nitrate is around three times higher than on sulfate) while the negative patterns are mostly the result of negative anomalies in the sulfate AOD and slight changes in nitrate AOD. The different impacts on the total AOD inform about the importance of not only considering ammonia behavior alone but accounting for $NO_x$ and $SO_2$, especially in the context of emission mitigation policies. It is important to note that global present-day nitrate AOD in CAMEO is twice higher (0.016; Table 6) than the 6 models-average quantified in the intercomparison from AeroCom Phase III but close to the GISS-OMA model estimate (0.015; Bian et al., 2017). However, global sulfate AOD in CAMEO (0.042) is in the recent model range (0.047) presented by Bian et al. (2017).

The all-sky direct radiative forcings at the top of the atmosphere (RF TOA) are presented in Table 6 and are calculated as the difference between the future considered CAMEO radiative fluxes and the historical CAMEO fluxes. Only replacing historical $NH_3$ emissions with those from SSP585 and SSP434 results in a net cooling of -114 $mW.m^{-2}$ and -160 $mW.m^{-2}$ induced by nitrate aerosol radiative forcing and a slight positive warming from the sulfate forcing ($\simeq 3$ $mW.m^{-2}$). The nitrate aerosol effects of the other experiments (CAMEO[434-126] and CAMEO[434-370]) are much more important (-164 and -243 $mW.m^{-2}$) than the highest anthropogenic radiative forcing calculated by Hauglustaine et al. (2014) which compares the scenario RCP8.5 for 2100 with pre-industrial conditions (-115 $mW.m^{-2}$). The sulfate aerosol radiative effect is 7 times more important in CAMEO[434-126] (343 $mW.m^{-2}$) than in CAMEO[434-370] where both $NO_x$ and $SO_2$ emissions from SSP126 are highly slown down in 2100.

## 5.4 Impact on $N_2O$ production

The oxidation of ammonia with the hydroxyl (OH) radical into $N_2O$ is an additional atmospheric pathway that can represent an important climate factor in the future. Multiple studies investigated the importance of the production of $N_2O$ from $NH_3$ which can range from 0.60 to 1.8 $Tg(N_2O)yr^{-1}$ (Dentener and Crutzen, 1994; Kohlmann and Poppe, 1999; Hauglustaine et al., 2014; Pai et al., 2021). Our present-day production matches well with this range (1.6 $Tg(N_2O)yr^{-1}$, Table 5) and represent 15 % of the present-day total anthropogenic $N_2O$ emissions used for CMIP6 (Gidden et al., 2018). However, considering that natural soil emission contribution (10 $Tg(N_2O)yr^{-1}$) is as important as the total anthropogenic source as estimated by Tian et al. (2024), our present-day production would in fact, represent 8% of the total $N_2O$ emissions. When considering our highest future $NO_x$ scenario (SSP3-7.0) combined with $NH_3$ emissions from SSP4-3.4, $N_2O$ production accounts for 18% (3.7 $Tg(N_2O)yr^{-1}$) of the future $N_2O$ anthropogenic emissions (under SSP3.70, Gidden et al., 2018). This result is close to the 21% quantified by Pai et al. (2021) using RCP trajectories for 2100.

**Table 6.** All-sky direct radiative forcing at the top of the atmosphere (RF TOA; $mW.m^{-2}$) and aerosol optical depth (AOD) of the nitrate and sulfate aerosols since the present-day and future evolution under the different scenarios considered in this study. Note that for AOD, future evolution is given as $\Delta AOD$ as the difference between the future and present-day AODs.

| | | $NO_3^-$ | $SO_4^{2-}$ |
|---|---|---|---|
| Present-day (2004-2014) | | | |
| CAMEO | AOD | 0.016 | 0.042 |
| Future (2090-2100) | | | |
| CAMEO[585] | $\Delta AOD$ | 0.008 | -0.0002 |
| | RF (TOA) | -114 | 1.9 |
| CAMEO[434] | $\Delta AOD$ | 0.011 | -0.0002 |
| | RF (TOA) | -160 | 4 |
| CAMEO[434-126] | $\Delta AOD$ | 0.012 | -0.026 |
| | RF (TOA) | -164 | 343 |
| CAMEO[434-370] | $\Delta AOD$ | 0.016 | -0.003 |
| | RF (TOA) | -243 | 46 |

## 6   Summary and conclusions

Because $NH_3$ impacts on the nitrate aerosol and nitrous oxide levels in the atmosphere, change in agricultural $NH_3$ emissions have important implications for climate and air quality. Regulating the agricultural sector is a challenge due to its importance in feeding the population and thus, understanding the impact of future agricultural $NH_3$ emissions on the atmospheric chemistry, is of high interest to design accurate mitigation emission scenarios. In this paper, the LMDZ-INCA global model is exploited to evaluate the impact of a new agricultural $NH_3$ emission dataset recently developed based on the ORCHIDEE Land Surface Model. This new dataset investigates the role played by $NH_3$ emissions in the atmosphere considering the dynamical environmental conditions and accounting for natural soil sources. The model results have been compared to $NH_3$ columns observed by the IASI instrument but also to surface concentrations measured by various observational networks. In addition, in LMDZ-INCA, tropospheric aerosols are also included through a representation of the sulfate nitrate–ammonium cycle and heterogeneous reactions between gas-phase chemistry and aerosols. With this model, we investigate the impact of present-day and future (2090-2100) $NH_3$ emissions on atmospheric composition, N deposition fluxes and climate forcing.

The key results of this paper are summarized as follows :

1. $NH_3$ emissions provided by CAMEO show good accuracy in the simulated $NH_3$ columns when evaluated against the IASI observations. Large reductions in the spatial model biases are noticeable compared to the reference version where the CEDS inventory is prescribed. More specifically, the biases decreased by at least 50 % in Africa, Latin America, and the US. CAMEO emissions not only improved the spatial representation of the columns, but also their seasonal cycle, especially in India, Equatorial Africa, China, and South America, where the skill functions calculated for the

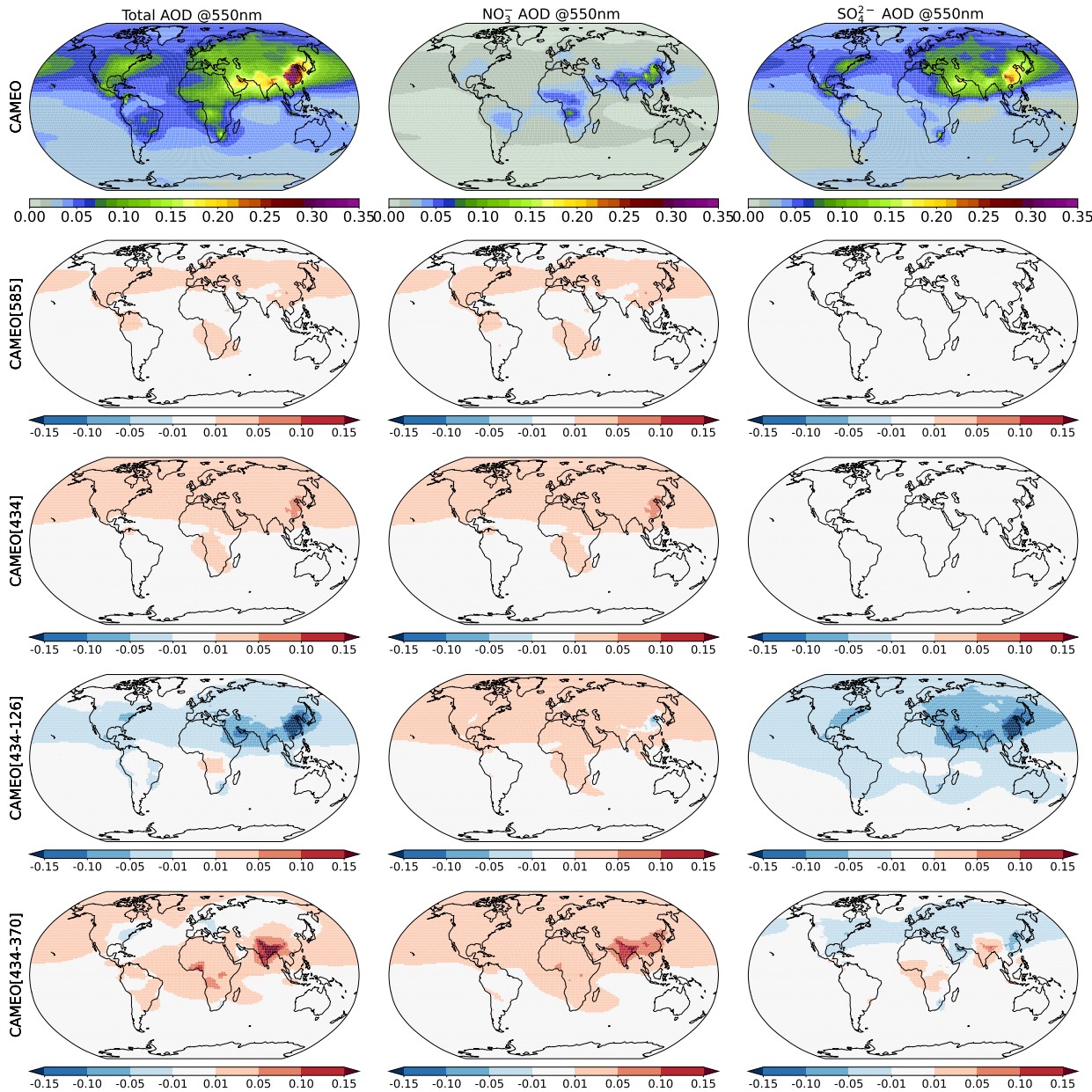

**Figure 10.** Mean annual total anthropogenic aerosol (i.e: nitrate + sulfate AOD; first column), nitrate aerosol (second column), and sulfate aerosol (third column) optical depths at 550nm simulated in the CAMEO simulation (1st row; over 2004-2014) and the anomalies between the CAMEO[SSPi] and CAMEO simulations ([SSPi]:585, 434, 434-126 and 434-370 in rows 2-5; over 2090-2100).

temporal variability gained between 1 to 3 points compared to the CEDS simulation. Comparisons of the simulated surface observations with ground-based observations indicate that using CAMEO emissions improved the representation

of both annual $NH_3$ and $NO_3^-$ concentrations at the surface in 2015 in China, the US, and Canada. In Europe, the reduction of the $NH_3$ bias however does not lead to improvement in the aerosol representation compared to CEDS.

2. The impact of CAMEO $NH_3$ emissions on $NH_x$ and $NO_y$ deposition fluxes has been investigated. The global budget of $NH_x$ is around 65 $TgNyr^{-1}$ which is 20 % higher than the average calculated from other model-based estimates (CCMI, EMEP MSC-W and CEDS). The difference is mainly explained by enhanced deposition in Africa which is twice higher than the deposition budget of the three alternative estimates. Due to relatively low nitrate levels and much higher $NH_3$ emissions in equatorial Africa, more $NH_3$ is removed through deposition processes, especially during the precipitation

season when wet scavenging occurs more frequently. Despite differences with the EMEP and CCMI modelling results, a seasonal comparison at a specific measurement station from the INDAAF network in western Africa shows good agreement in the $NH_4^+$ wet deposition when CAMEO emissions are used in the LMDZ-INCA model.

   3. Our analysis of the $NH_x$ deposition seasonal cycle highlights some discrepancies in the simulated fluxes from CCMI where seasonal variation is absent. The CCMI deposition dataset is a crucial forcing file for ESM more specifically

for Land Surface Models. Even though the agricultural sector is the major driver for $NH_3$ emission seasonality, $NH_x$ deposition can also play a role in more remote regions characterized by intensive precipitation seasons. Bi-directional flux of $NH_3$ can significantly impact $NH_3$ deposition, emission, reemission, and atmospheric lifetime Sutton et al. (2007). The interactive calculation of the different fluxes between the surface and the atmosphere has already been implemented in modelling approaches and shows significant improvements in the $[NH_3]$, $[NH_4^+]$, $[NO_3^-]$ and $NH_4^+$ wet depositions

at regional and global scale (Pleim et al., 2019; Vira et al., 2019, 2022). This aspect motivates the implementation of a coupling based on a compensation point for $NH_3$ between LMDZ-INCA and ORCHIDEE, which is already under development. The ongoing coupling seems promising to address the overestimation from CAMEO emissions and the resulting $NH_3$ columns over the US and Europe in July.

   4. Even though we are aware of some uncertainties and potential room for improvement, the model evaluation provides

some confidence for using CAMEO emissions to investigate the impact of future $NH_3$ emissions on atmospheric chemistry and climate. We have constructed four future scenarios for 2090-2100 in which the impact of CAMEO emissions for SSP5-8.5 and SSP4-3.4 under different $NO_x$ and $SO_2$ emission conditions has been studied. It is worth noticing that as far as we know, no future gridded livestock and interactive soil emissions have been used to investigate future $NH_3$ emission perturbations on the atmospheric chemistry at the global scale.

5. Future CAMEO emissions lead to an overall increase of the global $NH_3$ burden ranging from 59% to 235% while $NO_3^-$ burden increases by 57% - 114% depending on the scenario. By analyzing the behavior of CAMEO[434] and CAMEO[585], we investigated the isolated impact of future divergent $NH_3$ emissions. Our results highlight small changes in the nitrate formation mainly over eastern Asia, more specifically China (+ $2\mu g.m^{-3}$) where nitric acid concentrations are high ($HNO_3 > 6 \ \mu g.m^{-3}$) and thus ammonium neutralization is possible. It leads to an increase of around

0.05 in the total nitrate and sulfate AOD in China and a global increase of 19%. In CAMEO[434-126], in which $NO_x$

and $SO_2$ emissions are highly decreasing compared to present-day, we observed important decreases in surface nitrate and sulfate aerosol concentrations, especially over China (-4 $\mu g.m^{-3}$). In this scenario, even though $NH_3$ emissions increase the global nitrate AOD (+0.016), the negative impact of sulfate aerosol AOD is more important (-0.026), which results in a total AOD reduction of 23%. In CAMEO[434-126], the increase in the total nitrate burden and AOD indicates that despite less nitrate being formed at the surface, more nitrate is vertically uplifted in the upper troposphere. When combined with increased $NO_x$ and $SO_2$ emissions, higher $NH_3$ emissions lead to an enhanced formation of aerosol (+5 $\mu g.m^{-3}$ of $NO_3^-$) at the surface compared to present-day levels as is the case over India in CAMEO[434-370]. Despite the decrease of $NO_x$ and $SO_2$ emissions over China, the US, Europe, and Saudi Arabia, the total nitrate and ammonium burden is doubled due to the contribution of India as one of the highest hotspots in terms of aerosol ammonium nitrate precursors in this scenario. In addition, India and Africa are the regions experiencing the highest change in the total nitrate and sulfate AOD (+80 to +100 %) due to a higher contribution of the nitrate AOD.

6. In addition to the impact on the air quality and climate, future $NH_3$ emissions have a positive impact on the total $NH_x$ deposition fluxes over land and oceans (+35%). As already mentioned, the coupling between LMDZ-INCA and ORCHIDEE would improve the representation of the N exchanges. In addition to the direct impact of climate change on the emissions and deposition fluxes, one could also expect a change coming from the land-use shift due to its influence on the deposition velocity for instance.

7. Radiative forcings associated with the aerosol formation in the different scenarios have been presented. The impact of future CAMEO emissions alone results in a net cooling from nitrate aerosols which ranges from -114 $mW.m^{-2}$ to -160 $mW.m^{-2}$. By varying the future sulfate and nitrate emissions, the nitrate radiative effect can either overshoot (net total impact of -200 $mW.m^{-2}$) or be offset by the sulfate effect (net total impact of +180 $mW.m^{-2}$). As a comparison, Hauglustaine et al. (2014) estimated a negative radiative forcing from nitrate under RCP8.5 of around -115 $mW.m^{-2}$ (as pre-industrial emissions state as the baseline). These results from CAMEO[434-126] and CAMEO[434-370] suggest a significant impact of the future evolution of the $NH_3$ emissions on the climate depending on the mitigation measures that would be undertaken for $NO_x$ and $SO_2$ emissions.

8. In addition to the aerosol radiative effect, the $N_2O$ production from the oxidation of $NH_3$ has been estimated to be non-negligible in the present-day (1.6 $Tg(N_2O)yr^{-1}$) and could represent up to 18% (3.7 $Tg(N_2O)yr^{-1}$) of the future $N_2O$ anthropogenic emissions under our highest future $NO_x$ scenario (SSP3-7.0). Even though agricultural production is one of the most significant sector which impacts the N cycle, the potential use of ammonia for low-carbon energy production is rising the attention. The emerging ammonia economy, linked to hydrogen fuel has been estimated to produce an additional $N_2O$ atmospheric source of 1 $Tg(N_2O)yr^{-1}$ when considering a high estimate of reactive N emissions from the ammonia use in the energy sector (Bertagni et al., 2023). Knowing that a 1% atmospheric conversion of nitrogen in ammonia into $N_2O$ was used in the latter study, and that our estimate ranges between 1.5% and 2.25% depending on the scenario, we can expect a greater impact from the new global-scale ammonia economy.

# 7 Future directions : towards N interactions in ESM

In this study, the simulations are designed to isolate the impact of emission changes by keeping meteorological conditions fixed at present-day levels during 2090-2100. Climate change is expected to influence atmospheric chemistry through multiple interrelated factors, such as altered mean and extreme precipitation patterns that affect deposition, warming that could shift key chemical reactions, and wind variations that can affect aerosol transport. In a subsequent study, additional simulations will explore the combined impact of both emissions and climate change by incorporating changing meteorological conditions for atmospheric chemistry.

Incorporating the nitrogen cycle into Earth System Models is a recent advancement, as highlighted by Davies-Barnard et al. (2022). Developing interactions of nitrogen compounds is complex due to the intricate processes involved, necessitating readiness in coupling atmospheric chemistry and land components. The studies by Pleim et al. (2019); Vira et al. (2019, 2022) provide a foundational step toward bidirectional ammonia handling, though not yet fully integrated into existing ESMs. Vira et al. (2022) notes that FANv2 does not currently feed back nitrogen losses to the nitrogen cycle in the Community Land Model, leaving fertilizer nitrogen availability to crops unaffected. Our present approach does include feedback from nitrogen loss affecting available soil nitrogen for vegetation, even without a bidirectional scheme yet exploited. Additionally, in the CAMEO framework, we incorporated nitrogen biomass removal for livestock needs, ensuring nitrogen and carbon budget accuracy. Current efforts are focusing on developing nitrogen species exchanges at the atmosphere-surface interface in the IPSL-ESM, aiming to assess chemical and climate impacts through interactive coupling.

*Code availability.*  The LMDZ-INCA global model is part of the Institut Pierre Simon Laplace (IPSL) Climate Modelling Center Coupled Model. The documentation on the code and the code itself can be found at https://cmc.ipsl.fr/ipslclimate-models/ipsl-cm6/ (IPSL, 2024). The Python scipts used for analysing the data and plotting the analysed data are available from the corresponding author upon reasonable request. The ammonia columns measured from the IASI instrument onto the LMDZ grid are also available from the corresponding author upon request.

*Data availability.*  To access datasets of LMDZ-INCA results, please contact the corresponding author. Present-day and future simulated emissions from CAMEO can be respectively found at the following Zenodo repositories: https://zenodo.org/records/6818373, (Beaudor et al., 2022) and https://doi.org/10.5281/zenodo.10100435, (Beaudor et al., 2023b)

*Author contributions.*  NV, DH, JL, and MB designed the study. DH and MB prepared the emission sets and the model configuration. MB performed the simulation experiments, analyzed the output and prepared the manuscript with contributions from NV, DH, and JL. MVD and LC provided the IASI satellite product and performed the regridding of the data. All of the authors contributed to writing the manuscript.

*Competing interests.* The authors declare that they have no conflict of interest.

*Acknowledgements.* This study was partly funded by the European Union Horizon 2020 research and innovation programme under the ESM2025 project (grant agreement no. 101003536) and by the the Research Council of Norway under project No. 336227 "AMMONIA: Climate and environmental impacts of green ammonia (NH3)". The simulations were performed using HPC resources from GENCI (Grand Equipement National de Calcul Intensif) under project gen2201 and gen6328. Lieven Clarisse is a research associate supported by the Belgian F.R.S.-FNRS. We also thank Yao Ge for the open-access Python scripts exploited for evaluating the model to the ground-based observations. We are grateful for the technical support received from Anne Cozic and the fruitful discussions with Fabien Paulot. We thank the three reviewers for giving constructive suggestions that improved the manuscript.

*Financial support.* This study was partly funded by the European Union Horizon 2020 research and innovation programme under the ESM2025 project (grant agreement no. 101003536) and by the the Research Council of Norway under project No. 336227 "AMMONIA: Climate and environmental impacts of green ammonia (NH3)".

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
