# Peer review of "Evaluating present-day and future impacts of agricultural ammonia emissions on atmospheric chemistry and climate"

_EGUsphere, 2024_

## Referee Comment (RC1)

The manuscript "Evaluating present-day and future impacts of agricultural ammonia emissions on atmospheric chemistry and climate" by Beaudor et al. examines the impacts of ammonia ($NH_3$) emissions on atmospheric chemistry and climate. The authors used the chemistry-climate model (CCM) LMDZ-INCA to simulate present-day and future atmospheric processing of $NH_3$ (atmospheric concentration, aerosol formation and deposition), while also evaluating the climatic consequences (changes in radiative forcing RF and aerosol optical depth AOD) under different shared socio-economic pathways (SSPs). The novelty of the study lies in the incorporation of a new $NH_3$ emission input from a global land surface model, ORCHIDEE-CAMEO to the CCM. The model results were compared with satellite observations and ground-based measurements, demonstrating good agreement. This study contributes to adding knowledge of future projections of atmospheric chemistry and climate, with an emphasis on the role of $NH_3$ emissions. The manuscript aligns well with the scope of Atmospheric Chemistry and Physics, and I recommend publication after the authors address a few points.

General comments:

1. A key improvement the authors could make is to explain why the CCM was used to simulate the period from 2004 to 2014, but the model results were compared with ground-based measurements from 2015. It seems less convincing to compare modelled outputs averaged over 11 years to a different year that is not included in the simulation period. I understand the same-year comparisons might not always be possible because of insufficient data, but this does not appear to be an issue for this study. The emission model (Beaudor et al. 2023 GMD) provides $NH_3$ emissions from 2005 to 2015, and meteorological input from the ERA-interim reanalysis for 2015 should be accessible. Measured datasets of annual surface $NH_3$ concentrations in year 2010 are available as shown in Fig. 4 in Ge et al. (2021 GMD). This also raises the question of why $NH_3$ concentration comparisons for 2010 were not included in the analysis.

   I think it is important for the authors to either:
   A) Extend the CCM simulation by an additional year, incorporating the corresponding emission input, so that year 2015 can be included and directly compared with the observations from the same year.
   B) Provide a clear justification for comparing model outputs with measurements from different years, along with a discussion of the uncertainties and

implications associated with this approach, particularly given that there is inter-annual variability in NH$_3$ emissions.

Regardless of how the authors choose to address this question, I strongly encourage to include an evaluation for the year 2010, i.e., by adding annual NH$_3$ comparisons for different regions to Fig. 4 to 6.

2. It would be helpful to provide more details in the Method section. I find it unclear on the description of how RF and AOD are calculated.

3. It would be interesting to see a full NH$_3$ budget. E.g., I would like to see the authors show how much NH$_3$ contributes to the formation of N aerosols, in addition to what has been reported in Table 5.

Specific comments:

Line 87: The authors can provide a timeseries of the new NH$_3$ emissions that were used for the modelling. This figure can be put to the Supplementary materials, and readers can learn the inter-annual variability in the emissions.

Line 100: Why estimated agricultural NH$_3$ emissions were reported to be 44 Tg N per year from 2005 to 2015 in the emission paper (Beaudor et al. 2023 GMD), but it turns to 35 Tg N per year for 2004 to 2014 in this study? Why is there such a big difference?

Line 125: Is the emission resampled to fit the resolution of the CCM?

Line 154-161: Which meteorological variables were used for the modelling? What is the spatial and temporal resolution of the meteorological inputs? Another question is since LMDZ-INCA is a CCM, what is the reason for not using the weather fields generated by itself?

P9: Regarding Fig. 1, please consider showing the percentage difference for map (d) CAMEO – IASI.

Line 224-225: By what evidence can the authors claim that IASI observations does not reveal a "unique peak" which is a modelled feature?

Line 235-236: I see CAMEO shows the peak in the same months as CEDS for India (Fig. 3). Can you check?

P11: Figure 3 shows that CEDS performs better in EU than CAMEO.

Line 245: Why not compare annual NH$_3$ concentrations in 2010?

Line326: Why LMDZ-INCA uses a low constant? What is the implication?

P20: As suggested, it would be interesting to show the aerosol formation in Table 5.

Line 359-360: I think there is a problem with the calculations. E.g., for CAMEO[585], the increase is (0.27-0.17)/0.17 = 0.59, 59 % rather 37 %. Please do check the numbers in the following text and in the abstract.

Line 388: Why $H_2SO_4$ is not included for $T_S$ in Equation 1?

P23: I think it is helpful to explain why look at different pressure levels.

Line 404-405: Please restructure the sentence "It is explained by the reduced amount of $NH_3$ ..." to improve the clarity.

Line 416-418: Why is there a shift from nitrate-rich to ammonia-rich at 900 hPa, while the chemical domains at the surface does not change much?

Line 425-426: The sentence is not very clear to me. The effect of what?

Line 426-428: TBH, I barely see the difference between 434-126 and 434 over India, Europe and the US...

Line 440: What does "levels" mean here? Emissions?

P25: I feel there is a need for better referring to figures in this whole section. Sometimes it is difficult to follow the text without looking at the figures, but there is no clear referring.

Line 448: Delete "Finally,".

Line 458: CAMEO[434-370] or CAMEO[585]? Please check Fig. 13.

Line 412-413: Why attribute the same $NO_y$ deposition between CAMEO and CAMEO[434] to identical NOx emissions? Figure 10 shows that there are higher $[NO_3^-]$ over EA.

Line 472-474: This is a long sentence. I suggest split into short sentences to improve readability.

Line 478-479: Why nitrate AOD is mostly increase in CAMEO[434-126]?

Line 503: How is the value 1.6 $Tg(N_2O)yr^{-1}$ calculated? Same question for the value of 2.9 in line 508.

P1 in SM: I think the authors should consider using maps that show the differences between emissions from future and present-day.

---

## Referee Comment (RC2)

In "Evaluating present-day and future impacts of agricultural ammonia emissions on atmospheric chemistry and climate", the authors present simulations of present day and future ammonia pollution at a global scale with the LMDZ-INCA model with new and improved NH3 emissions from ORCHIDEE, then evaluate the impact of future ammonia burden on air quality and climate. Comparison of present-day simulated NH3 with satellite and ground-based measurements shows improvement from other inventories. Future projections are informed by SSP scenarios, and consider both changes in aerosol burden and composition, as well as changes in N2O production from NH3 oxidation. Overall, this manuscript presents an interesting outlook on future air quality and climate conditions from the perspective of ammonia emissions that is within the scope of ACP. I recommend publication following some clarification and revision of methods and results.

General Notes:

When comparing against ground-based data, no consideration appears to have been made for differences in monitor spatial or temporal coverage, which may hinder direct comparisons. For example, it appears ground-based measurements are only reported for 2015, which is mis-matched temporally with the model simulation. What averaging methods are employed to gather a single value for different species that are each measured with different methods and temporal resolution? Beyond data completion of data record, is any further QA performed with this data to ensure measurement validity? The authors report excellent agreement between measured NH3 and modeled results in the Mid-US in Figure 6; however, there are only 2-3 surface sites shown within the "hotspot" region of the Mid-US to make this comparison, and at least one site appears to have the maximum difference threshold between model and measurements. Surface monitor locations are notably sparse in agricultural regions of the US where emissions tend to be high, and similar biases may exist in other regional networks. Please clarify the methods and findings in this section.

Further analysis may be warranted regarding the findings of section 5.1, particularly for changes projected in China. This topic is important, but to my knowledge there is no substantial evidence that SO2 and NO2 controls increase ambient ammonia burden, and the authors only reference one publication in this discussion of these results, though others exist for China and other regions. In Warner et al., 2017 GRL, for example, the authors postulate that ammonia increases in China are due to a combination of sulfur controls, increased fertilizer use, and increasing local temperatures. It seems that these variables could be further investigated within the model outputs here.

I find the expansion of results towards radiative forcing to be a particularly interesting portion of this work; however, the methodologies here are unclear, especially regarding the N2O estimates from ammonia oxidation. These N2O estimates are also highlighted as a major conclusion of this work in the abstract, but the analysis presented in the actual manuscript seems overly brief for this to be a main conclusion. Similarly, the abstract references a range of N2O (0.43 to 2.10 Tg/yr), but only single figures are presented in the results and discussion. This section also refers to figure 5, but N2O is not a component listed within that table. Should changes in the N2O budget as a result of increasing ammonia be a main conclusion of this work, more in-depth analysis should be considered, perhaps with a figure or table dedicated to this section. The conclusion section also presents somewhat different estimates than what is represented in the abstract and main text. It feels startling to see mention and estimates of ammonia/hydrogen economy in this section of the

conclusions when it has not been previously mentioned in the manuscript. Clarification of numerical estimates and methodologies in these sections is necessary.

Specific Comments:

-Line 32: the meteorological variables described are unclear. Is this referring to air temperature and humidity, or soil temperature and moisture?

-Line 42: a citation should be used regarding the claim that livestock activities and synthetic fertilizer use are projected to increase.

-Please ensure that citation formatting is consistent throughout the manuscript. For example, on page 2 line 43-48, the authors reference Hauglustaine 2014 on the first sentence but not the second. Later in the same paragraph, the authors reference Beaudor 2024 on both sentences referring to that work.

-Line 67: clarify the gas-phase and particulate species examined in this work.

-Line 68: It may be beneficial to include greater detail of the SSP scenarios used in this work in the introductory section of the manuscript. As mentioned here, it is unclear what is meant exactly by "most and least significant increase".

Line 72-74: delete "more precisely" and "respectively".

-Line 75: what is meant by different levels? This type of descriptor is used several times throughout the manuscript but is poorly defined. Delete "structure of the".

-Page 4: why were the present day years 2004-2014 chosen for this analysis? This time period is somewhat mismatched both with satellite record (2011-2014) and surface-based measurements (2015).

-Lines 97-99: Animal density considerations are an interesting and somewhat underexamined variable towards NH3 emissions, and are described here as a critical input, yet the text does not describe how changing densities are represented. While this is described by a prior publication, that publication is not referenced at this point of the manuscript, and it may be worthwhile to add further detail to the current manuscript.

-Line 107: change "it represents" to "each represent"

-Please ensure that all acronyms/abbreviations are described before they are used in the manuscript. For example, LMDZ-INCA is not defined until line 120 but is used several times before this.

-Line 151: Sander 2015 is an outdated reference and should be updated to Sander 2023.

-Line 154: Is there a reason the ERAInterim reanalysis product was used instead of the ERA5 product?

-Line 160: What meteorological variables are employed?

-Line 161: delete "however"

-Line 171-172: I'm not fully certain what is meant by low levels and contrasting conditions in these descriptions.

-It would be helpful to have greater reference towards what each CAMEO-SSP simulation represents, such as an additional column in Table 2 describing textually the purpose of the different conditions.

-Line 178: Why are only these years of the IASI record employed for comparison?

-Line 203: change to "CAMEO emission also enhance".

-Lines 209-214: I find this analysis of compensating errors to be unclear. Where is the 47% bias estimate coming from for the US? From table 3, I see a 46% difference between Mean Obs and MBE CEDS. What is the threshold for significance of these differences?

-Line 220: Why is the seasonal cycle omitted? Could this be added to the supporting information?

-Page 9: An additional figure showing the difference between CAMEO and CEDS would be helpful, possibly for the Supporting information. In this figure and others (e.g., Figure 12), differences between outputs are reported only by absolute differences—would it be more clear to represent these as relative (%) differences between outputs?

-Lines 239-241: It is unclear why dust estimates are being brought up here.

-Line 252 and Table: The AMoN network is run by the US National Atmospheric Deposition Program (NADP), not the US EPA. Please ensure the text and tables reflect this.

-Line 261: Beijing

-Line 264: high concentrations of what?

-Related to the comment above, use of panel lettering on plots such as Figures 4-6 and more detailed in-text references would aid in understanding these comparisons. I question if all of these figures are necessary for the main body of the manuscript instead of the Supporting Information, as they are not heavily mentioned or discussed within the main text.

-Page 16: figure caption 6 describes European stations but displays information pertaining to North America. Ensure all figure captions are representative.

-Line 300: delete "also".

-Paragraph starting on Line 315: "depositions" should be referred to as just "deposition".

-Line 322: Change "except in" to "aside from".

-Line 359: Change "relatively" to "relative".

-Page 20: The calculations presented in the text are not correct. Ex: The difference between CAMEO and CAMEO[585] should be 59%, not 37%. Please ensure all percent changes are calculated correctly throughout the manuscript.

-Page 23: Figure 11 is referenced in text before Fig 10. Please ensure all figures are referenced in the appropriate order.

-Line 488-489: Why is AeroCom Phase III and/or GISS brought up here when it has not been mentioned before this? Citations should be provided if the authors are comparing their findings to other literature.

-Line 497: Why is the sulfate radiative impact not shown for both CAMEO simulations?

-Line 499: change "slow down" to "slowed down".

-Page 33. The last numbered paragraph (9) of the summary and conclusions represent future directions, and therefore should not be included as a numbered conclusion item. I believe it would be stronger to end the manuscript with a paragraph or two that bring together the full conclusions of the manuscript, incorporating the future directions noted in the (9) paragraph.

---

## Author Comment (AC1)

**Response to reviewers and description on the revised manuscript**

First, we thank the reviewers and editor for taking the time to review our work. We appreciate the constructive comments made to improve the manuscript.
The manuscript has been thoroughly and carefully revised in line with the evaluations received.

Our point-by-point responses (in magenta, unformatted text) following the referee's comments (in black) can be found below.
The previous text version is in blue and corrections applied to the manuscript appear in **bold magenta**.
Please note that the lines mentioned refer to the newly submitted version.

**Reviewer 1:**

The manuscript "Evaluating present-day and future impacts of agricultural ammonia emissions on atmospheric chemistry and climate" by Beaudor et al. examines the impacts of ammonia (NH3) emissions on atmospheric chemistry and climate. The authors used the chemistry-climate model (CCM) LMDZ-INCA to simulate present-day and future atmospheric processing of NH3 (atmospheric concentration, aerosol formation and deposition), while also evaluating the climatic consequences (changes in radiative forcing RF and aerosol optical depth AOD) under different shared socio-economic pathways (SSPs). The novelty of the study lies in the incorporation of a new NH3 emission input from a global land surface model, ORCHIDEE-CAMEO to the CCM. The model results were compared with satellite observations and ground-based measurements, demonstrating good agreement. This study contributes to adding knowledge of future projections of atmospheric chemistry and climate, with an emphasis on the role of NH3 emissions. The manuscript aligns well with the scope of Atmospheric Chemistry and Physics, and I recommend publication after the authors address a few points.

We are thankful for the fruitful comments given by Reviewer 1. Our point-by-point responses can be found below.

General comments:
    1. A key improvement the authors could make is to explain why the CCM was used to simulate the period from 2004 to 2014, but the model results were compared with ground-based measurements from 2015. It seems less convincing to compare modelled outputs averaged over 11 years to a different year that is not included in the simulation period. I understand the same-year comparisons might not always be possible because of insufficient data, but this does not appear to be an issue for this study. The emission model (Beaudor et al. 2023 GMD) provides NH3 emissions from 2005 to 2015, and meteorological input from the ERA-interim reanalysis for 2015 should be accessible. Measured datasets of annual surface NH3 concentrations in year 2010 are available as shown in Fig. 4 in Ge et al. (2021 GMD).

This also raises the question of why NH3 concentration comparisons for 2010 were not included in the analysis.
I think it is important for the authors to either:

A) Extend the CCM simulation by an additional year, incorporating the corresponding emission input, so that year 2015 can be included and directly compared with the observations from the same year.
B) Provide a clear justification for comparing model outputs with measurements from different years, along with a discussion of the uncertainties and implications associated with this approach, particularly given that there is inter-annual variability in $NH_3$ emissions.
Regardless of how the authors choose to address this question, I strongly encourage to include an evaluation for the year 2010, i.e., by adding annual NH3 comparisons for different regions to Fig. 4 to 6.

We fully agree with the reviewer that the mismatch in the years of the model and the observations was not optimal for a consistent evaluation.
To address this important point we extended the CAMEO simulation until 2015 and compared the results against the same year of observations. We replaced Figures 4 to 6 in the revised manuscript and adjusted Table 4 and text accordingly.
The comparison for 2010 has been added to the Supplementary Material and not in the main text because the results appear less robust than in 2015 due to much fewer observation numbers as highlighted in the following table with the example for $NH_3$ observations:

| # obs | EMEP cc | UK Networks | NNDM | EANET | US EPA | NAPS |
|---|---|---|---|---|---|---|
| 2010 | 26 | 23 | 10 | 25 | 11 | 7 |
| 2015 | **38** | 22 | **25** | 27 | **31** | 7 |

This new sentence has been incorporated:
**An evaluation for 2010 has also been conducted to enhance the robustness of our findings and similar regional signals are found as for 2015.**
**Owing to the fewer observations available globally in 2010 compared to 2015, these results are presented in the Supplementary Material (Fig. S7, S8, and S9).**

2. It would be helpful to provide more details in the Method section. I find it unclear on the description of how RF and AOD are calculated.
This description has been added in the Methods section, l.215:
**Multiple radiative forcings (RFs) and aerosol optical depths (AODs) related to changes in atmospheric composition due to agricultural emissions are calculated online during the LMDZ-INCA simulations.**

**As also mentioned by Terrenoire et al., 2022, the radiative calculations in the general circulation model (GCM) utilize an enhanced version of the ECMWF scheme established by Fouquart et al., 1980 for the solar spectrum and by Morcrette et al., 1991 for the thermal infrared spectrum.**

**The short-wave spectrum is segmented into two ranges: 0.25–0.68 and 0.68–4.00 μm. The model incorporates the diurnal variation of solar radiation and permits fractional cloud cover within a grid cell. These RFs are computed as instantaneous, clear-sky, and all-sky forcings at both the surface and the top of the atmosphere. To evaluate the future effects of ammonia emissions on aerosol concentration and climate, the all-sky direct radiative forcings are determined by subtracting the historical CAMEO radiative fluxes from the future simulation being analyzed.**

**In Section 5.3, the all-sky forcings at the top of the atmosphere and AOD will be discussed for aerosols, similar to what was done by Hauglustaine et al., 2014.**

3. It would be interesting to see a full NH3 budget. E.g., I would like to see the authors show how much NH3 contributes to the formation of N aerosols, in addition to what has been reported in Table 5.

We agree with the Reviewer, this information was missing so we added the $NH_3$ loss from the $NH_4$ formation in Table 5. Please note that we revised the entire table and numbers might have changed because of an inconsistent total area used for the calculation of the different terms. We also added biomass burning to the total $NH_3$ emissions which were missing.

Specific comments:

Line 87: The authors can provide a timeseries of the new NH3 emissions that were used for the modelling. This figure can be put to the Supplementary materials, and readers can learn the inter-annual variability in the emissions.

A figure has been added to the SI representing the evolution of the global agricultural $NH_3$ emissions for SSP2-4.5, SSP4-3.4 and SSP5-8.5 from CAMEO under future climate (SSP2-4.5 is shown but not exploited in the present study).

Line 100: Why estimated agricultural NH3 emissions were reported to be 44 Tg N per year from 2005 to 2015 in the emission paper (Beaudor et al. 2023 GMD), but it turns to 35 Tg N per year for 2004 to 2014 in this study? Why is there such a big difference?

Thanks for raising this question.

The reference paper (Beaudor et al. 2023, GMD) presents CAMEO simulations from a different setup from the CMIP6 framework used in this study and the latest submitted study to JAMES (https://essopenarchive.org/doi/full/10.22541/essoar.170542263.35872590/v1).

In the framework of phase 6 of The Coupled Model Intercomparison Project, a common set of experiments is designed and aims to provide the boundary condition and forcing dataset needed for CMIP6.

Regarding the boundary conditions, for self-consistency, CAMEO for the CMIP6 framework exploits the 3-hourly near-surface meteorological fields simulated by the Institut Pierre Simon Laplace (IPSL) Earth System Model :IPSL-CM6A-LR ESM (Boucher et al., 2020), in the context

of CMIP6 for near-surface air temperature, specific humidity, wind speed, pressure, short- and longwave incoming radiation, rainfall, and snowfall.

The reference paper is based on the Climatic Research Unit (CRU) and Japanese reanalysis (JRA) dataset (CRU-JRA V2.1) (Harris et al., 2014) (preprocessed and adapted by Vladislav Bastrikov, LSCE, July 2020), provided at 6 h time steps.

On another hand, "forcing files" (such as N deposition, N fertilizers etc..) can be less realistic than those used in our reference version, even though they have been extensively used in Land Surface Models and evaluated.

For instance, data for synthetic fertilizer in CMIP6 does not assume that managed grasslands are fertilized.

For this reason, the reference simulation presented in the "emission" paper does not exploit the latest data from CMIP6 as forcing files.

For the present study, to be consistent with the forcing files from CMIP6 for the different SSPs, another historical CAMEO simulation with CMIP6 data was necessary.

As mentioned above, the fertilization budget is much lower in the "CMIP6" simulation (97 TgN/yr against 118  TgN/yr) while the manure produced is 5 TgN/yr less.

The lower total N input in the "CMIP6" simulation explains its lower soil emission budget.

The region where the fertilization use budget is the most reduced in the "CMIP6" simulation is Asia (44% less in India and around 30% in China and Asia Tropical South).

However, synthetic fertilizer use in Africa and Latin America is higher in the "CMIP6" simulation than in the reference simulation (of about 48% and 30%).

Even though fertilizer use is more important in these regions, the resulting regional soil emissions also decrease due to its weak contribution to the total N input and a small reduction in N manure application (-13% and -2%).

The following paragraph has been added:
**Please note that due to a different set of input/boundary conditions data, the agricultural ammonia emissions from the present study are 9 TgN/yr lower than the one reported in the reference study (Beaudor et al., 2023, GMD). This difference is mainly explained by the non-consideration of managed grassland in the CMIP6 synthetic fertilizer forcing which led to a total fertilization input of 97 TgN/yr against 118 TgN/yr in the reference study.**
**On another hand, the different climatic forcings may also impact the emissions.**
**For self-consistency, CAMEO for the CMIP6 framework exploits the 3-hourly near-surface meteorological fields simulated by the Institut Pierre Simon Laplace (IPSL) Earth System Model :IPSL-CM6A-LR ESM (Boucher et al., 2020), in the context of CMIP6 for near-surface air temperature, specific humidity, wind speed, pressure, short- and longwave incoming radiation, rainfall, and snowfall.**
**The reference paper is based on the Climatic Research Unit (CRU) and Japanese reanalysis (JRA) dataset (CRU-JRA V2.1) (Harris et al., 2014) (preprocessed and adapted by Vladislav Bastrikov, LSCE, July 2020), provided at 6 h time steps.**

Line 125: Is the emission resampled to fit the resolution of the CCM?

Yes, indeed, the emissions are regridded at the LMDz-INCA spatial resolution and read at a monthly time-step.

The following sentence has been added:

**The CAMEO emissions are, first, carefully regridded onto the model grid through a preprocessor program and provided at a monthly time resolution to the chemistry-transport model.**

Line 154-161: Which meteorological variables were used for the modelling? What is the spatial and temporal resolution of the meteorological inputs? Another question is since LMDZ-INCA is a CCM, what is the reason for not using the weather fields generated by itself?

In this study, meteorological data from the European Centre for Medium-Range Weather Forecasts (ECMWF) ERA5 reanalysis were used. The relaxation of the GCM winds towards ECMWF meteorology is performed by applying a correction term at each time step to the GCM u and v wind components with a relaxation time of 2.5 h (Hourdin and Issartel, 2000; Hauglustaine et al., 2004). The ECMWF fields are provided every 6 h and interpolated onto the LMDZ grid.

It has been clarified line 238.

P9: Regarding Fig. 1, please consider showing the percentage difference for map (d) CAMEO – IASI.

As indicated in the following figure, the percentage difference map leads to extremely high values in the non source regions.

In these zones, there is little $NH_3$ or its presence is not easily detected by IASI.

Therefore this representation of the differences is not optimal.

We rather mentionned percentages in the manuscript for the emissions zones, for instance:

**In contrast, in the tropical Sub-Saharan zone, these emissions lead to an underestimation of column values by -0.4 molecules $10^{16}$.cm$^{-2}$(- 45%)**

[Figure]

Percentage Difference: (CAMEO - IASI) / IASI

Mean: -33.97%
Max: 71336.96%

Percentage Difference (%)

Line 224-225: By what evidence can the authors claim that IASI observations does not reveal a "unique peak" which is a modelled feature?
This sentence has been reformulated as follows:
**In the US and Europe, the CAMEO columns show a unique peak (0.7 molecules x10$^{16}$cm$^{-2}$) during summer, while the IASI observations inform about a lower maximum value (0.5 and 0.4 molecules x10$^{16}$cm$^{-2}$, respectively) reached over March-September.**

Line 235-236: I see CAMEO shows the peak in the same months as CEDS for India (Fig. 3). Can you check?
Thanks for noticing this inaccuracy. We corrected it as follows:
**In India, both CAMEO and CEDS simulate a peak value occurring 2 months earlier than that measured by IASI, but the value is 1.5 times higher with CEDS than with CAMEO.**

P11: Figure 3 shows that CEDS performs better in EU than CAMEO.
A more explicit statement is added regarding the better performance of CEDS:
**In Europe, CEDS surpasses CAMEO when it comes to the magnitude of seasonal variations.**

Line 245: Why not compare annual NH3 concentrations in 2010?
This aspect has been addressed in the general comment responses.

Line326: Why LMDZ-INCA uses a low constant? What is the implication?
We apologize but it is a confusion from our side, the Henry's law constant has been updated according to Bian et al., 2017.
We corrected this aspect in the manuscript.

P20: As suggested, it would be interesting to show the aerosol formation in Table 5.
This aspect has been addressed above.

Line 359-360: I think there is a problem with the calculations. E.g., for CAMEO[585], the increase is (0.27-0.17)/0.17 = 0.59, 59 % rather 37 %. Please do check the numbers in the following text and in the abstract.

Thank you very much for your careful reading, we corrected the numbers.

Line 388: Why H2SO4 is not included for TS in Equation 1?

According to Seinfeld et al., 1997, Metzger et al., 2012 and Hauglustaine et al., 2014 $SO_4^{2-}$ represents several forms that are neutralized by ammonia (more precisely it refers to all of $H_2SO_4$, $NH_4HSO_4$, $(NH_4)_3H(SO_4)_2$, and $(NH_4)2SO_4$).

It has been clarified in the manuscript l.487:

**(TS, including all forms of $SO_4^{2-}$ as $H_2SO_4$, $NH_4HSO_4$, $(NH_4)_3H(SO_4)_2$, and $(NH_4)2SO_4$)**

P23: I think it is helpful to explain why look at different pressure levels.

The different pressure levels help to understand the persistence of ammonia in the atmosphere even though it is known as a relatively short-lived species. In addition at the surface we can not distinguish between the different regimes properly because of the excess of NH3.

To be clearer this sentence has been added:

**To gain a better understanding of the behavior of ammonia and its persistence in the atmosphere under future scenarios, we have selected different pressure levels, including surface level, 900 hPa, and 500 hPa.**

Line 404-405: Please restructure the sentence "It is explained by the reduced amount of NH3 …" to improve the clarity.

We restructured the sentence as:

**The decrease in $NH_3$ can be attributed to its rapid transformation into $NH_4$ at pressures of 900 hPa and 500 hPa**.

Line 416-418: Why is there a shift from nitrate-rich to ammonia-rich at 900 hPa, while the chemical domains at the surface does not change much?

Indeed this was unclear and was in fact a comparison to the historical CAMEO simulation. We clarified as follows:

**In this region, compared to the CAMEO simulation, there is a noticeable expansion of the nitrate-rich and ammonia-rich domains at 900 hPa which is explained by relatively higher $NH_3$ concentration and a stronger limitation by $HNO_3$ availability.**

Line 425-426: The sentence is not very clear to me. The effect of what?

The impact of both NOx and $SO_2$ emissions decreases on ammonia abundance. It has been clarified.

Line 426-428: TBH, I barely see the difference between 434-126 and 434 over India, Europe and the US…

Indeed by plotting the absolute difference between the two simulations, it appears that the increase in the ammonia concentration is the highest over China but non negligible over multiple regions (India, the US, Middle East, North Africa and Europe).
The sentence has been modified:
**This is also confirmed by comparing NH3 from CAMEO[434-126] and CAMEO[434] where NH3 emissions are identical but a slightly stronger impact on the concentrations is highlighted for instance in China and India.**

[Figure]

Line 440: What does "levels" mean here? Emissions?
Yes, it has been clarified.

P25: I feel there is a need for better referring to figures in this whole section. Sometimes it is difficult to follow the text without looking at the figures, but there is no clear referring.
We added more references to the figures when needed throughout this specific section.

Line 448: Delete "Finally,".
It has been done.

Line 458: CAMEO[434-370] or CAMEO[585]? Please check Fig. 13.
It is actually for the 3 simulations a similar pattern. We corrected it accordingly.

Line 462-463: Why attribute the same NOy deposition between CAMEO and CAMEO[434] to identical NOx emissions? Figure 10 shows that there are higher [NO3 - ] over EA.
Indeed, we agree that this is an uncomplete interpretation of our result.
We analyzed it differently:
**There are minimal spatial differences in the deposition of NOy between CAMEO and CAMEO[434] (and CAMEO[585]), as constant NOx emissions lead to a balancing effect, resulting in decreased HNO3 deposition and increased NO3- deposition, especially in China.**

Line 472-474: This is a long sentence. I suggest split into short sentences to improve readability.

We adopted this recommendation.

Line 478-479: Why nitrate AOD is mostly increase in CAMEO[434-126]?
Emissions of $SO_2$ are much lower in that scenario than in the other ones.
As $NH_3$ is reacting with the sulfate in priority to form ammonium sulfate aerosols, there is much more available $NH_3$ to form ammonium nitrate aerosols. This counterbalances largely the $NO_x$ emission reductions.
This explanation has been incorporated in the main text.

Line 503: How is the value 1.6 Tg(N2O)yr-1 calculated? Same question for the value of 2.9 in line 508.
The calculation of $N_2O$ production has been added to the Model description section (3) as:
**Ammonia losses occur as a result of both wet and dry deposition, ammonium formation, and the oxidation processes in the gas phase, although the latter contributes only a small amount to its overall loss.**
**However, the loss through this oxidation pathway generates non-negligible source of nitrous oxide ($N_2O$).**
**The production of $N_2O$ results from the following reaction:**
**$NH_2+NO_2\rightarrow N_2O+H_2O$**

**The overall production rate is calculated as:**
$R_{nh2->N2o} = A \times exp^{-Ea/RT} \times [NH_2] \times [NO_2]$ **with A = 2.1e$^{-12}$ and Ea/R = -650**

The factor used to convert Tg(N2O)/yr into TgN/yr is 1.58.

We apologize for this mistake, the 2.9 Tg(N2O)/yr value is not the correct one for SSP434-370 (this one corresponds to the production under SSP434), we corrected the value in the text to 3.7 Tg(N2O)/yr (the percentage given was correct).

P1 in SM: I think the authors should consider using maps that show the differences between emissions from future and present-day.
We agree on that suggestion and changed Figure S1 for anomalies instead.

---

## Author Comment (AC2)

**Response to reviewers and description on the revised manuscript**

First, we thank the reviewers and editor for taking the time to review our work. We appreciate the constructive comments made to improve the manuscript.
The manuscript has been thoroughly and carefully revised in line with the evaluations received.

Our point-by-point responses (in magenta, unformatted text) following the referee's comments (in black) can be found below.
The previous text version is in blue and corrections applied to the manuscript appear in **bold magenta**.
Please note that the lines mentioned refer to the newly submitted version.

**Reviewer 2:**

In "Evaluating present-day and future impacts of agricultural ammonia emissions on atmospheric chemistry and climate", the authors present simulations of present day and future ammonia pollution at a global scale with the LMDZ-INCA model with new and improved NH3 emissions from ORCHIDEE, then evaluate the impact of future ammonia burden on air quality and climate. Comparison of present-day simulated NH3 with satellite and ground-based measurements shows improvement from other inventories. Future projections are informed by SSP scenarios, and consider both changes in aerosol burden and composition, as well as changes in N2O production from NH3 oxidation. Overall, this manuscript presents an interesting outlook on future air quality and climate conditions from the perspective of ammonia emissions that is within the scope of ACP. I recommend publication following some clarification and revision of methods and results.

We are thankful for the constructive comments given by Reviewer 2. Our point-by-point responses can be found below.

General Notes:

When comparing against ground-based data, no consideration appears to have been made for differences in monitor spatial or temporal coverage, which may hinder direct comparisons. For example, it appears ground-based measurements are only reported for 2015, which is mismatched temporally with the model simulation.
We fully agree with the reviewer that the mismatch in the years of the model and the observations was not optimal for a consistent evaluation.
To address this important point we extended the CAMEO simulation until 2015 and compared the results against the same year of observations. We replaced the Figures 4 to 6 in the revised manuscript and adjusted Table 4 and text accordingly.
In addition, as also requested by Reviewer 1, the comparison to 2010 has been added to the Supplementary Material. The results for 2010 appear less robust than in 2015 due to much fewer observation numbers as highlighted in the following table with the example for $NH_3$ observations:

| # obs | EMEP cc | UK Networks | NNDM | EANET | US EPA | NAPS |
|-------|---------|-------------|------|-------|--------|------|
| 2010 | 26 | 23 | 10 | 25 | 11 | 7 |
| 2015 | **38** | 22 | **25** | 27 | **31** | 7 |

This new sentence has been incorporated:

**An evaluation for 2010 has also been conducted to enhance the robustness of our findings and similar regional signals are found as for 2015.**
**Owing to the fewer observations available globally compared to 2015, these results are presented in the Supplementary Material (Fig. S7, S8, and S9).**

What averaging methods are employed to gather a single value for different species that are each measured with different methods and temporal resolution? Beyond data completion of data record, is any further QA performed with this data to ensure measurement validity?

The spatial matching between model-observation has been performed with a simple data 'mining' corresponding to the extraction of the closest pixel for each network site. The spatial averages of each network are then performed to gather a single value for different species.
Regarding the temporal coverage, we ensured that the yearly period was covered at 75% in the observation dataset for each site and performed yearly averages in the model data.
For the present study, no further QA has been performed on the observation data but we acknowledge that the quality of the measurement is a crucial aspect that can bring uncertainties to the evaluation. By using both satellite and ground-based measurements, we hope to have demonstrated that the model produces relatively reasonable results.
Precision has been incorporated into the manuscript:

**As recommended in Ge et al., 2021, we only consider measurements where 75% of the year was captured to avoid bias in our analysis and we perform yearly averages on the model data.**

The authors report excellent agreement between measured NH3 and modeled results in the Mid-US in Figure 6; however, there are only 2-3 surface sites shown within the "hotspot" region of the Mid-US to make this comparison, and at least one site appears to have the maximum difference threshold between model and measurements. Surface monitor locations are notably sparse in agricultural regions of the US where emissions tend to be high, and similar biases may exist in other regional networks. Please clarify the methods and findings in this section.

We completely agree with the Reviewer on this aspect, however, the original text does not report "excellent agreement" or similar terms:

"In North America, CAMEO reflects a realistic spatial pattern against measurements with high concentrations located in the Mid-US (> 4 $\mu g.m^3$) and rather low concentrations on the Mid-Atlantic side.
An underestimation of CAMEO is still observable in the Mid-West region (<2 $\mu g.m^3$)."
To emphasize the scarcity of the data in the intensive agricultural region of Mid-US, we added the following precision:

**Even though the spatial gradient is fairly represented in the model, it is crucial to note that only a few observations are available, especially in the Mid-US region, an intensive agricultural area that would benefit from further observation data for more accurate evaluation.**

Further analysis may be warranted regarding the findings of section 5.1, particularly for changes projected in China. This topic is important, but to my knowledge there is no substantial evidence that SO2 and NO2 controls increase ambient ammonia burden, and the authors only reference one publication in this discussion of these results, though others exist for China and other regions. In Warner et al., 2017 GRL, for example, the authors postulate that ammonia increases in China are due to a combination of sulfur controls, increased fertilizer use, and increasing local temperatures. It seems that these variables could be further investigated within the model outputs here.

We thank the reviewer for providing an insightful view of the future changes in ammonia burden in intensive agricultural areas such as China.

We think that investigating other variables would be beneficial, however, we want to emphasize that in the framework of our study, the meteorological variables (including local temperatures) are kept at the present-day level. This choice is justified by the fact that we want to isolate the agricultural and industrial drivers that impact atmospheric chemistry.

As highlighted in the simulation experiments by the following Figure, CAMEO434 and CAMEO434-126, we can clearly state that both the decrease of $SO_2$ and NOx are important factors of the increased ammonia burden (2-3 µg.m$^{-3}$).

[Figure]

The impact of increased ammonia emissions alone is responsible for at least 5 µg.m$^{-3}$ increase in the region (see following figure, already presented in the manuscript), mainly explained by the strong increase in synthetic fertilizer use in the SSP4-3.4 (+30 TgN/yr compared to historical application, see Fig. 4 of https://essopenarchive.org/doi/full/10.22541/essoar.170542263.35872590/v1).

[Figure]

[NH$_3$] ($\mu g.m^{-3}$)
CAMEO run

Δ[NH$_3$] ($\mu g.m^{-3}$)
434 - CAMEO

0.0  0.1  0.1  0.8  2.0  4.0  6.0  8.0

-5.0 -4.0 -2.0 -1.0 -0.5 -0.1 -0.0 0.0 0.1 0.5 1.0 2.0 4.0 5.0

To consider your interesting suggestion we reformulated the paragraph:

**In line with the later study, the effect of the simultaneous reductions in Nox and SO$_2$ emissions in CAMEO[434-126] is even stronger on [NH$_3$] due to the combined increase in NH$_3$ emissions mainly explained by the significant increase in the use of synthetic fertilizers in China (+30 TgN/yr compared to historical application).**

**This is also confirmed by comparing [NH$_3$] from CAMEO[434-126] and CAMEO[434] where NH$_3$ emissions are identical but a slightly stronger impact on the concentrations is highlighted for instance in China and India (Figure 9).**

**It is notable that other combined factors have been shown to significantly contribute to the increased NH$_3$ levels in China.**

**For instance, in Warner et al., 2017, the authors suggest that the rise in ammonia levels in China between 2003 and 2015 can be attributed to sulfur controls, greater fertilizer application, and rising local temperatures.**

**The present study does not explore the impact of meteorological factors, as it focuses on the isolated impact of human-related ammonia emissions.**

I find the expansion of results towards radiative forcing to be a particularly interesting portion of this work; however, the methodologies here are unclear, especially regarding the N2O estimates from ammonia oxidation. These N2O estimates are also highlighted as a major conclusion of this work in the abstract, but the analysis presented in the actual manuscript seems overly brief for this to be a main conclusion. Similarly, the abstract references a range of N2O (0.43 to 2.10 Tg/yr), but only single figures are presented in the results and discussion. This section also refers to figure 5, but N2O is not a component listed within that table. Should changes in the N2O budget as a result of increasing ammonia be a main conclusion of this work, more in-depth analysis should be considered, perhaps with a figure or table dedicated to this section. The conclusion section also presents somewhat different estimates than what is represented in the abstract and main text. It feels startling to see mention and estimates of ammonia/hydrogen economy in this section of the conclusions when it has not been previously mentioned in the

manuscript. Clarification of numerical estimates and methodologies in these sections is necessary.

We agree with the reviewer that the results around the $N_2O$ production were lacking context. To do so, the introduction has been completed with a motivation for studying $N_2O$ production:
**The ammonia oxidation pathway mentioned is a direct contributor to nitrous oxide ($N_2O$) emissions in the atmosphere, which is a potent greenhouse gas.**
**Future losses of nitrous oxide could increase significantly due to intensified agricultural emissions and the emerging hydrogen fuel economy, which heavily relies on ammonia as an energy carrier (Hauglustaine et al., 2014, Bertagni et al., 2023).**

The calculation of $N_2O$ production has been added to the Model description section (3) as:
**Ammonia losses occur as a result of both wet and dry deposition, ammonium formation, and the oxidation processes in the gas phase, although the latter contributes only a small amount to its overall loss.**
**However, the loss through this oxidation pathway generates a non-negligible amount of nitrous oxide ($N_2O$).**
**The production of $N_2O$ results from the following reaction:**
**$NH_2 + NO_2 \rightarrow N_2O + H_2O$**

**The overall production rate is calculated as:**
$R_{nh2->N2o} = A \times exp^{-Ea/RT} \times [NH_2] \times [NO_2]$ **with A = 2.1e$^{-12}$ and Ea/R = -650**

This production term is part of Table 5 under the new label "$N_2O$ prod." and is described in the legend as "$N_2O$ production through $NH_3$ gas phase loss (TgN/yr)".

We double-checked the values and corrected them to have consistency between the main text, the conclusions and the abstract.

Specific Comments:
-Line 32: the meteorological variables described are unclear. Is this referring to air temperature and humidity, or soil temperature and moisture?
Indeed, there is a lack of precision. We are primarily referring to soil temperature and moisture. This info has been added in the revised version.

-Line 42: a citation should be used regarding the claim that livestock activities and synthetic fertilizer use are projected to increase.
References have been added.

-Please ensure that citation formatting is consistent throughout the manuscript. For example, on page 2 line 43-48, the authors reference Hauglustaine 2014 on the first sentence but not the second. Later in the same paragraph, the authors reference Beaudor 2024 on both sentences referring to that work.
Thanks, we double-checked that.

-Line 67: clarify the gas-phase and particulate species examined in this work.
We clarified as follows: **trace gases: $NH_3$, $NO_2$ and ionic species: $NH_4^+$ , $NO_3^-$ , $SO_2^{-4}$.**

-Line 68: It may be beneficial to include greater detail of the SSP scenarios used in this work in the introductory section of the manuscript. As mentioned here, it is unclear what is meant exactly by "most and least significant increase".
We agree that more explanation is needed. Therefore additional descriptions have been included in the introduction:
**SSP4-3.4 and SSP5-8.5 describing respectively "A world of deepening inequality", and "Fossil-fueled Development – Taking the Highway" (Calvin et al., 2017, Kriegler et al., 2017), reflect divergent agricultural drivers.**

**In the first place, SSP4-3.4 represents the scenario with the weakest evolution of livestock, while SSP5-8.5 shows the most significant increase among all Shared Socioeconomic Pathways (SSPs) according to Riahi et al., 2017.**
**In line with these trends, the fossil fuel-intensive scenario SSP5-8.5 also experiences the highest demand and production of food and feed crops among the three scenarios considered, as noted by Beaudor et al., 2024.**
**This increase occurs despite low population growth and is driven by the prevalence of diets high in animal products (Fricko et al., 2017, Kriegler et al., 2017).**
**Despite the peak in food and feed crop production, projected fertilizer applications in SSP5-8.5 rise only slightly.**
**This is attributed to the minimal production of biofuel crops, a result of the lack of climate mitigation policies and rapid advancements in agricultural productivity.**
**In contrast, SSP4-3.4 exhibits the highest use of fertilizer and reveals significant regional differences, with high consumption lifestyles among elite socioeconomic categories and low consumption levels for the rest of the population (Calvin et al., 2017).**

Line 72-74: delete "more precisely" and "respectively".
Done.

-Line 75: what is meant by different levels? This type of descriptor is used several times throughout the manuscript but is poorly defined. Delete "structure of the".
The following sentence have been modified in the manuscript for better comprehension:
**In this paper, we present six simulations from LMDZ-INCA, including two present-day simulations, with CEDS and CAMEO inventories for $NH_3$ emissions and four future simulations over 2090-2100 with future $NH_3$ emissions from CAMEO and other sources at different future levels (i.e. globally decreased and regionally-contrasted level of emissions).**

-Page 4: why were the present day years 2004-2014 chosen for this analysis? This time period is somewhat mismatched both with satellite record (2011-2014) and surface-based measurements (2015).

In order to have a more robust signal from the model results, we considered an 11-year period (2004-2014), which has also been selected for future simulations. Even though the CEDS inventory is extended until 2020 (McDuffie et al., 2020), the forcings for the CAMEO model (and ORCHIDEE) are considered a "future period" after 2014. This is why the CAMEO emission coverage influenced our simulation period.
The IASI instrument period selected is justified by the overlap of the two satellites (Metop A and B), which did not offer the best accuracy before 2011.
Regarding the mismatched years with the model and the observations, we extended our historical ammonia emissions with CAMEO to 2015 but the forcings (N input, livestock densities, Land Use maps and meteorology fields) are not considered present-day and it is part of a particular SSP for 2015. We assumed that similar emissions results are expected for any SSP since 2015 is the starting year of all SSPs.

-Lines 97-99: Animal density considerations are an interesting and somewhat underexamined variable towards NH3 emissions, and are described here as a critical input, yet the text does not describe how changing densities are represented. While this is described by a prior publication, that publication is not referenced at this point of the manuscript, and it may be worthwhile to add further detail to the current manuscript.
We agree that more details regarding the livestock density would benefit the manuscript.
We incorporated a description of the methodology of this driver and highlighted the advantages of the use of the CAMEO emission datasets:
**These emission datasets have been recently constructed from a newly gridded livestock product and the use of the global process-based CAMEO before being evaluated against CMIP6 emissions developed by the Integrated Assessment Models (IAMs) in Beaudor et al., 2024.**
**The future livestock distribution has been estimated until 2100, originally, for three divergent SSPs (SSP2-4.5, SSP4-3.4 and SSP5-8.5) through a downscaling method based on regional livestock trends and future grassland areas (the detailed methodology can be found in Beaudor et al., 2024).**
**The simulated agricultural ammonia emissions show a good alignment with the global IAMs estimates of 50 to 66 TgN/yr from CMIP6 under SSP4-3.4 and SSP5-8.5.**
**Although a global agreement is shown between the IAMs and CAMEO future emissions, we identified three interesting advantages in favor of the use of CAMEO emissions:**
- **The consistent consideration of the key ammonia emissions drivers (i.e. N input, meteorology, livestock, and land use) among all future SSPs which is the result of the use of a single process-based model.**
- **The spatial heterogeneity is driven by environmental conditions and not kept constant over time within predefined regions using the information from the historical period.**
- **Incorporating CAMEO into the land component of the IPSL ESM ensures better consistency throughout the various components, including LMDZ-INCA, paving the way for advancements in our understanding.**

Ref: Beaudor, M., Vuichard, N., Lathiere, J., and Hauglustaine, D.: Future trends of agricultural ammonia global emissions in a changing climate,https://doi.org/10.22541/essoar.170542263.35872590/v1, 2024.

-Line 107: change "it represents" to "each represent"
Done.

-Please ensure that all acronyms/abbreviations are described before they are used in the manuscript. For example, LMDZ-INCA is not defined until line 120 but is used several times before this.
Thank you for pointing it out. We checked and defined properly the abbreviation used in the manuscript.

-Line 151: Sander 2015 is an outdated reference and should be updated to Sander 2023.
Thank you for pointing this outdated reference. We corrected it.

-Line 154 & Line 160 : Is there a reason the ERAInterim reanalysis product was used instead of the ERA5 product? What meteorological variables are employed?
In this study, meteorological data from the European Centre for Medium-Range Weather Forecasts (ECMWF) ERA5 reanalysis were used. The relaxation of the GCM winds towards ECMWF meteorology is performed by applying a correction term at each time step to the GCM u and v wind components with a relaxation time of 2.5 h (Hourdin and Issartel, 2000; Hauglustaine et al., 2004). The ECMWF fields are provided every 6 h and interpolated onto the LMDZ grid.
It has been clarified and corrected line 238.

-Line 161: delete "however".
Done.

-Line 171-172: I'm not fully certain what is meant by low levels and contrasting conditions in these descriptions. It would be helpful to have greater reference towards what each CAMEO-SSP simulation represents, such as an additional column in Table 2 describing textually the purpose of the different conditions.
We thank the reviewer for this suggestion and added a column to Table 2 to explicit the scope of each simulation.

-Line 178: Why are only these years of the IASI record employed for comparison?
The period selected for the IASI comparison has been recommended by the IASI team because of the quality of the retrievals, mainly due to the overlap of Metop B and A satellites.

-Line 203: change to "CAMEO emission also enhance".
Done.

-Lines 209-214: I find this analysis of compensating errors to be unclear. Where is the 47% bias estimate coming from for the US? From table 3, I see a 46% difference between Mean Obs and MBE CEDS. What is the threshold for significance of these differences?

We agree with the reviewer that this calculation was confusing since the MBE for CAMEO in that particular case is closed to 0.

This sentence has been corrected:

**Using CAMEO also significantly reduced the modeled bias over the US, with an MBE close to 0 (Table3).**

The sentence about the compensating error has been clarified also as follows:

**However, it is important to note potential compensating errors within the regions, particularly in the selected African region (shown in the black box in Figure 1).**
**For instance, in the Saharan area, CAMEO emissions cause an overestimation of column values by 0.3 molecules $10^{16}$ cm$^{-2}$.**
**In contrast, in the tropical Sub-Saharan zone, these emissions lead to an underestimation of column values by -0.4 molecules $10^{16}$ cm$^{-2}$ (-45%).**

-Line 220: Why is the seasonal cycle omitted? Could this be added to the supporting information?

We replaced Figure 3 by the following figure in which the seasonal cycle of the emissions is also represented.

[Figure]

-Page 9: An additional figure showing the difference between CAMEO and CEDS would be helpful, possibly for the Supporting information.

This has been added to Figure S1.

In this figure and others (e.g., Figure 12), differences between outputs are reported only by absolute differences—would it be more clear to represent these as relative (%) differences between outputs?

We considered this suggestion and mentioned some percentage difference values in the text into parenthesis when appropriate. For instance, l.528:

**On another hand, CAMEO[434-126] depicts important negative anomalies of $NH_4^+$, $NO_3^-$, and $SO_4^{2-}$ especially in China (> 4 µg.m−3, equivalent to 60-80% Figure 10, subplots D, I and N).**

-Lines 239-241: It is unclear why dust estimates are being brought up here.

We thank the reviewer for detecting this mistake. This sentence was part of a previous version which should have been also removed.

-Line 252 and Table: The AMoN network is run by the US National Atmospheric Deposition Program (NADP), not the US EPA. Please ensure the text and tables reflect this.

It is right that this is a misunderstanding from our side, AMoN network belong to the NADP which we use for [$NH_{3(g)}$] only. Annual $NO_2$, $NH_4^+$, $NO_3^-$, and $SO_4^{2-}$ are provided by the EPA. It has been clarified in the text and tables.

-Line 261: Beijing

We corrected it.

-Line 264: high concentrations of what?

We clarified it.

-Related to the comment above, use of panel lettering on plots such as Figures 4-6 and more detailed in-text references would aid in understanding these comparisons. I question if all of these figures are necessary for the main body of the manuscript instead of the Supporting Information, as they are not heavily mentioned or discussed within the main text.

Thank you for the recommendation, we decided to move to the SI, the scatterplot figures presenting the evaluation of surface concentrations using ground-based measurements.
We also applied a letter labelling on the relevant Figures and added more references in the text.

-Page 16: figure caption 6 describes European stations but displays information pertaining to North America. Ensure all figure captions are representative.

We corrected it.

-Line 300: delete "also". Done.
-Paragraph starting on Line 315: "depositions" should be referred to as just "deposition". Done.
-Line 322: Change "except in" to "aside from". Done.
-Line 359: Change "relatively" to "relative". Done.

-Page 20: The calculations presented in the text are not correct. Ex: The difference between CAMEO and CAMEO[585] should be 59%, not 37%. Please ensure all percent changes are calculated correctly throughout the manuscript. Done.

-Page 23: Figure 11 is referenced in text before Fig 10. Please ensure all figures are referenced in the appropriate order. We doubled check.

-Line 488-489: Why is AeroCom Phase III and/or GISS brought up here when it has not been mentioned before this? Citations should be provided if the authors are comparing their findings to other literature.

Indeed, that was missing. We introduced more carefully the AeroCom initiative at the *Model set up* level and cited the authors when comparing to their results:

**Extensive evaluations of the aerosol component of the LMDZ-INCA model have been carried out during the various phases of Aerosol Comparisons between Observations and Models (i.e. AeroCom (Gliss et al., 2021,Bian et al., 2017).**

-Line 497: Why is the sulfate radiative impact not shown for both CAMEO simulations?

The radiative forcing impact of sulfate is shown in table 6 for all "future" simulations (CAMEO[SSPi]) at the row "RF (TOA)". The value is not given for the historical simulation because we defined this simulation as the baseline for RF and AOD.

This has been newly clarified line 550:

**The all-sky direct radiative forcings at the top of the atmosphere (RF TOA) are presented in Table 6 and are calculated as the difference between the future considered CAMEO radiative fluxes and the historical CAMEO fluxes.**

But also line 188:

**In addition to the concentrations of ammonia-related aerosols and gases exploited in this study, the all-sky direct radiative fluxes at the top of the atmosphere and the Aerosol Optical Depth (AOD) of the various aerosol components are calculated online by the atmospheric circulation model. More details on the radiative fluxes computation can be found in Hauglustaine et al. (2014).**

-Line 499: change "slow down" to "slowed down". Done.

-Page 33. The last numbered paragraph (9) of the summary and conclusions represent future directions, and therefore should not be included as a numbered conclusion item. I believe it would be stronger to end the manuscript with a paragraph or two that bring together the full conclusions of the manuscript, incorporating the future directions noted in the (9) paragraph.

A new section named "Future directions : towards N interactions in ESM" have been added and the old paragraph (9) of the previous section has been moved to this new section.

**In this study, the simulations are designed to isolate the impact of emission changes by keeping meteorological conditions fixed at present-day levels during 2090-2100. Climate change is expected to influence atmospheric chemistry through multiple interrelated factors, such as altered mean and extreme precipitation patterns that affect**

deposition, warming that could shift key chemical reactions, and wind variations that can affect aerosol transport.

In a subsequent study, additional simulations will explore the combined impact of both emissions and climate change by incorporating changing meteorological conditions for atmospheric chemistry.

Incorporating the nitrogen cycle into Earth System Models is a recent advancement, as highlighted by Davies-Barnard et al., (2022).

Developing interactions of nitrogen compounds is complex due to the intricate processes involved, necessitating readiness in coupling atmospheric chemistry and land components.

The studies by Pleim et al., 2019, Vira et al., 2019 and Vira et al., (2022) provide a foundational step toward bidirectional ammonia handling, though not yet fully integrated into existing ESMs.

Vira et al., (2022) notes that FANv2 does not currently feed back nitrogen losses to the nitrogen cycle in the Community Land Model, leaving fertilizer nitrogen availability to crops unaffected.

Our present approach does include feedback from nitrogen loss affecting available soil nitrogen for vegetation, even without a bidirectional scheme yet exploited.

Additionally, in the CAMEO framework, we incorporated nitrogen biomass removal for livestock needs, ensuring nitrogen and carbon budget accuracy.

Current efforts are focusing on developing nitrogen species exchanges at the atmosphere-surface interface in the IPSL-ESM, aiming to assess chemical and climate impacts through interactive coupling.

---

## Author Comment (AC3)

**Response to reviewers and description on the revised manuscript**

First, we thank the reviewers and editor for taking the time to review our work. We appreciate the constructive comments made to improve the manuscript.
The manuscript has been thoroughly and carefully revised in line with the evaluations received.

Our point-by-point responses (in magenta, unformatted text) following the referee's comments (in black) can be found below.
The previous text version is in blue and corrections applied to the manuscript appear in **bold magenta**.
Please note that the lines mentioned refer to the newly submitted version.

**Reviewer 3:**

This is an interesting paper on present-day and future impacts of NH3 emissions. After careful reading I find the study also somewhat limited. The big sales-argument of the study is the new CAMEO agricultural emission module- and the authors spend a lot space to demonstrate that the overall model performance using CAMEO derived emissions compared to CEDS emissions is better. The model is consequently used to explore some future SSP-like scenarios- unfortunately without contrasting to the impacts of the NH3 emissions from the more well-known existing SSP marker scenarios that have e.g. been assessed in Chapter 6 of the AR6 WG1 report. On a first glance- at least for global totals- the IPCC WG future SSP emission ranges (e.g. Figure 6.18) look quite similar to the CAMEO emission changes reported in Table 1- and a valid but unanswered question is to what extent the exploration of the future impacts (aerosol burden, deposition, N2O production) would have looked very different if the scenarios used by the CMIP community would have been used. Is this study new, or confirming existing results?

We thank Reviewer 3 for taking the time to carefully read our manuscript and for the insightful comments.

It is important to note that, this is the first time that future agricultural $NH_3$ emissions, influenced by climate change, livestock management, and nitrogen fertilizer use, are used to explore their impact on atmospheric chemistry and climate.
It is not straightforward to compare our results with previous studies (from Aerchmip for instance) because of the unicity of the present experiments in which future agricultural emissions for ammonia have been isolated from other future changes. It is likely that the differences arising from a comparison with other versions of the model or previous studies would not inform us on the $NH_3$ emission impacts purely.

We suggest, however, to include, the relevance of land surface modelling of future $NH_3$ emissions against CMIP6 inventory (the emissions presented in Chapter 6 of the AR6 WG1 report) for atmospheric chemistry impact analysis as a part of Section 2.2 "Future emission scenarios".

Beaudor et al. (2024) demonstrate a global agreement between agricultural ammonia emissions developed by the IAMs and simulated with CAMEO. The global estimates from the IAMs inventories are, respectively, 50 and 66 TgN.yr−1 under SSP5-8.5 and SSP4-3.4, compared to 50 TgN.yr−1 and 68 TgN.yr−1 for CAMEO. In this previous work, three interesting advantages are highlighted in favor of the use of CAMEO emissions:

– The consideration of environmental conditions and therefore climate change (i.e. soil temperature and humidity, $CO_2$ increase, vegetation changes).

– The consistent consideration of the key ammonia emissions drivers (i.e. N input, meteorology, livestock, and land use) among all future SSPs which is the result of the use of a single process-based model.

– The spatial heterogeneity is driven by environmental conditions and not kept constant over time within predefined regions using the information from the historical period.

– Incorporating CAMEO into the land component of the IPSL ESM ensures better consistency throughout the various components, including LMDZ-INCA, paving the way for advancements in our understanding.

Considering the constraints of IAMs in precisely reflecting the primary factors influencing ammonia emissions, exploring their effects on atmospheric chemistry and climate beyond a global level appears unconvincing. We propose a hypothetical comparison based on the regional differences observed in the IPCC emissions and the CAMEO emissions projected for 2100.

Figure S3 (Supplementary Material) highlights the major regional differences between CMIP6 and CAMEO emissions in 2100 for the two considered SSPs (SSP4-3.4 and SSP5-8.5). The most distinguishable region is Africa, specifically North Africa's savanna combined with Equatorial Africa, where the CMIP6 emissions for both SSPs are more than twice as high as those for CAMEO (>15 TgN.yr−1). The primary explanation for this pattern lies in the simplified downscaling strategy adopted by the IAM method for projection. The approach applies a constant factor across the entire African continent over time, based on historical emissions, neglecting to account for regional influences such as livestock raising expansion and changes in fertilizer application. Specifically, the northern Maghreb region is expected to play a significant role in the future, particularly under SSP4-3.4, as projections indicate an expansion in cultivated lands and fertilizer application, likely driven by the cultivation of bioenergy crops. As a consequence, one of the most expected differences between CMIP6 and CAMEO emissions impact would be a more enhanced production of aerosol formation and NOy and NHx deposition under [434-370] where NOx and SO2 emissions are projected to increase compared to the present-day in Africa. In contrast, in China, the smaller emission fluxes predicted by the IAMs under both SSPs compared to CAMEO indicate that we can expect a limitation / decrease in the formation of ammonium-related aerosols and therefore the resulting deposition, which would be stronger under [434-126].

[Figure]

The second limitation, acknowledged by the authors, is the use of present-day climate conditions to explore future NH3 emissions (and impacts). To my opinion, this aspect is of particular importance (along with the inclusion of compensation point approaches), where the use of CAMEO could represent a step forward. The authors promise to develop a separate study on this aspect- I understand the material could be too much for a single publication- but it does undermine  undermine the relevance of the 'future' evaluation in this paper.

The first objective of this paper is to investigate the impact of present-day and future CAMEO emissions on atmospheric chemistry and climate.
It is critical to note, that the future ammonia emissions from CAMEO do include the impact fo climate change and has been explored more in details in the following study:
(https://essopenarchive.org/doi/full/10.22541/essoar.170542263.35872590/v1)
 We acknowledge and apologize that this aspect was not clearly stated.
We hope to have addressed this more clearly by mentioning it in the introduction as:
**For the first time, we propose to investigate how future agricultural NH$_3$ emissions, influenced by climate change, livestock management, and nitrogen fertilizer use, will impact atmospheric chemistry and climate (kept at present-day conditions).**

Climate is kept at present-day climate conditions, only for the impact atmospheric chemistry and climate (i.e., aerosol formation, deposition and nitrous oxide formation), in our study. This allows us to disentangle the different complex drivers at play and focus on the direct impact of the additional NH$_3$ produced by the agricultural sector under different regional levels of aerosol precursor emissions.
We are aware that climate change constitutes a critical aspect of future atmospheric chemistry and we suggest a new extended section dedicated to future perspectives:

**In this study, the simulations are designed to isolate the impact of emission changes by keeping meteorological conditions fixed at present-day levels during 2090-2100. Climate change is anticipated to influence atmospheric chemistry through multiple interrelated factors, such as altered mean and extreme precipitation patterns impacting deposition, warming that could shift some aerosol precursor reactions, and wind variations that may affect aerosol transport. In a subsequent study, additional simulations will explore the combined impact of both emissions and climate change by incorporating future meteorological conditions.**

Ass the paper stands it relies qstrongly on the evaluation of the model system with satellite and in-situ data. Obviously the authors have done a substantial and commendable effort, and I am not always convinced how relevant and constraining the comparisons are. In addition the addition the manuscript is very lengthy, and the length of the model evaluation section is contributing to this. My suggestion is to move a lot of detailed evaluation material to supplementary material and instead making an effort to better summarize and discuss the signficance of these evaluation findings in the main manuscript.  An example of where better discussion is warranted is the discussion of the match of seasonal cycles (vs annual average) - where it is not made very clear why the effort is done, and what we can learn from this.

Thank you for the recommendation, as also proposed by Reviewer #2, we decided to move to the SI, the scatterplot figures presenting the evaluation of surface concentrations using ground-based measurements.
We also applied a letter labelling on the relevant Figures and added more references in the text. Regarding the suggested discussion, we improved this aspect by adding this paragraph at the end of the evaluation section:

**The main takeaway from the evaluation of $NH_3$ columns and surface concentrations is that using CAMEO emissions results in a significant improvement in the spatial and temporal patterns, particularly in the seasonal cycle, compared to CEDS, except in the US and Europe. It is still important to note that, CAMEO improves the ground spatial variability of  $NH_3$  in the US as highlighted by measurement comparison.  The skill functions shown in the Taylor plots indicate that CAMEO emissions can more accurately capture the temporal variability of emissions in hotspot regions when compared to IASI observations.**

**It is important to focus on matching seasonal cycles rather than only comparing annual averages for multiple reasons. Seasonal cycles provide insights into the variations in emissions and atmospheric pathways throughout the year, which can be linked to meteorological conditions (air temperature and precipitation), seasonal activities (like fertilizer application or manure handling) and specific events (like biomass burning). Understanding these patterns allows for more accurate predictions of air pollution and climate impacts. The effort to improve emission estimates, particularly in regions where discrepancies exist, such as Europe and the US, highlights the importance of utilizing process-based approaches that lets room for considering the bi-directionality property of ammonia.**

Bringing it back to my earlier comment- what is the difference of this study with earlier efforts: The relevance of the better performance for future climate impact could then focus on showing that the changes (e.g. in Africa and South America) make a sizeable difference for the overall global results.

We thank the reviewer for sharing this interesting point.
We addressed this point conjointly with the first Reviewer's comment as a new section about the relevance of land surface modelling of future $NH_3$ emissions against CMIP6 inventory for atmospheric chemistry impact analysis.

Lastly, I recommend proofreading by a native speaker, in particular I noticed space for improvement in the abstract- the entry point for most readers. I made some suggestions for abstract and introduction, but the manuscript would benefit throughout from a proper proofreading.

We are thankful for the careful reading and the constructive suggestions for improving the manuscript understanding.
The manuscript has been proofread by a native speaker.

Below I provide detailed comments- I have spent less effort to discuss details of the model evaluation section.

L1: are responsible for a major source=>English. Are a major source or responsible for a major fraction of emissions. It has been corrected.

L2. Intensification is usually used I the context of agricultural production methods. The drivers are growing population and increasing food demand leading a.o. to intensification. It has been corrected.

L4 Surface deposition feedback is not clear. Feedbacks of the carbon cycle to increased N-deposition? We clarified.

L6 ammonia and ammonium pathways. Reduced nitrogen pathways? We corrected it.

L9 explain what is the CAMEO module about. Emissions,deposition, bidirectional? Note that the journal may require first-use explanation of acronyms. We defined it.

L10 And what about the climate- was also for 2100 conditions, or remained present day?
The climate for the emissions was also taken for 2100 conditions.

L11 What is meant with ammonia representation? Comparison to observations (from satellite, in-situ?). We detailed the sentence as follows:

**We demonstrate that this novel emission set enhances the spatial and temporal variability of atmospheric ammonia in regions such as Africa, Latin America, and the United States in comparison to the static reference inventory (Community Emissions Data System; CEDS) when assessed against satellite and surface network observations.**

L12 Higherammonia emissions in Africa, as simulated by CAMEO compared to other studies, reflect enhanced present-day reduced nitrogen (NHx) deposition flux.This sentence is not clear: I suspect that the authors indicate that the emissions are also reflected in higher deposition fluxes, which is logical, and even more logical if these are confirmed by observational evidence.

We clarified the sentence:

**The CAMEO simulation indicates higher ammonia emissions in Africa relative to other studies, which is corroborated by increased current levels of reduced nitrogen deposition NHx, a finding that aligns with observations in West Africa.**

L14 At this place a sentence introducing the scenario framework of this study would be needed; as there are probably more implementations. Also explain that apart from the magnitude of emissions changes, an important parameter is the ratio of NOx/NH3 emissions, and also SO2 emissions.
We incorporated these aspects in the abstract as recommended.

L 19 In climate sciences the word Overshoot is used in a very specific climate scenario context, related to emission pathways. Suggest: Overcompensate? We corrected it.

L20 could be useful to include here how much this is as a fraction of the current best estimate of the overall N2O budget. We incorporated the fraction of future anthropogenic emissions in the abstract as highlighted in the main text.

L24 the issues wrg nitrogen deposition are mostly biodiversity loss (and climate)- maybe for abstract to mention these rather than nitrogen deposition. It has been newly mentionned.

l 29 surface deposition processes. Wet deposition is not a surface deposition process, but still important. Indeed, we corrected it.

l32 account for 85 % of anthropogenic atmospheric NH3 emissions. I would doubt that this statement holds to NH3 abundance in general. The reviewer is right, we corrected it.

l39 very good agreement (can you add one sentence what you mean with this? The following sentence has been completed:
**CAMEO-based seasonal variation of $NH_3$ emissions which depend on both meteorological and agricultural practices highlights very satisfying correlation scores with satellite-based emissions as demonstrated in Beaudor et al., 2023 and Beaudor et al., 2024.**

l45/48 clarfiy whether is this still referening to Hauglustaine 2014? We added the reference for this part.

L49 not sure what is meant with 'removal treatments' ? Oxidation of NH3?
We reformulated as follows:
**RCP scenarios have also been exploited to study the importance of future atmospheric NH$_3$ on chemistry and climate with a special focus on atmospheric NH$_3$ losses including oxidation processes**

L55 Can you clarify shortly (and in later section somewhat more extenstively how the livestock distribution differ, and whether this study is using Beaudor, SSP or both?

This new paragraph has been added to the introduction:
**In the first place, SSP4-3.4 represents the scenario with the weakest evolution of livestock, while SSP5-8.5 shows the most significant increase among all Shared Socioeconomic Pathways (SSPs) according to Riahi et al., 2017.**

In addition, the "Future emission scenarios" section has been extended:

**In this study, future emissions for different SSPs are used for the 2090-2100 period. CAMEO emissions for SSP5-8.5 and SSP4-3.4 have been exploited for future agricultural and natural NH$_3$ emissions in the CAMEO[SSPi] (SSPi: 585, 434, 434-126, 434-370) simulations where agricultural sources account for 50 and 68 TgN yr$^{-1}$ (respectively for SSP5-8.5 and SSP4-3.4).**
**SSP5-8.5 and SSP4-3.4 have been chosen primarily as they represent, respectively, the least and most important increases of NH$_3$ emissions estimated over 2090-2100 Beaudor et al., 2024.**
**These datasets have been recently constructed from a newly gridded livestock product and the use of the global process-based CAMEO before being evaluated against CMIP6 emissions developed by the Integrated Assessment Models (IAMs) in Beaudor et al., 2024.**

**The future livestock distribution has been estimated until 2100, originally, for three divergent SSPs (SSP2-4.5, SSP4-3.4 and SSP5-8.5) through a downscaling method based on regional livestock trends and future grassland areas (the detailed methodology can be found in Beaudor et al., 2024).**

L62 could improve the correspondence of modelled concentrations. …. with …
We corrected it.

L70 importance for .. We corrected it.

L87 Two other reference databases that come to mind are EDGAR and IIASA/GAINS. One sentence quoting the numbers for these alternative would help understanding whether the quoted 'improvements' apply in comparison to all available databases.
This sentence has been added:

**As comparison the EDGARv8.1 inventory (https://edgar.jrc.ec.europa.eu/index.php/dataset_ap81) quantifies for all anthropogenic sectors a total of NH3 emissions of 42 TgNyr$^{-1}$ in 2010 (including 36 TgNyr$^{-1}$ for the agricultural sector).**

l99 I guess not only indoor, but important also to understand the manure management aspects.I remember also a rather large contribution from fire emissions in CEDS- can you comment.
It is right, the general term would be "manure management", and we changed "indoor" for this term.

CAMEO does not include a representation of biomass burning from agricultural practices, the total fire emissions including small fires from cultivated land come from the Global Fire Emissions Database GFED4s inventory (Van der Werf et al., 2017).
This sentence has been added at line 110:

**Emissions from biomass burning, including small fires from agricultural waste burning come from the Global Fire Emissions Database (GFEDs) inventory (Van der Werf et al., 2017). NH$_3$ emissions from fire account for 4.2 TgN/yr for the historical period.**

L109. Summarize what you found from this comparison, and why that is important.
This section has been extended as detailed in our answer to the first point raised by the reviewer.

L113 stringent emission regulations, but clarify that this is not necessarily the case for NH3 which is much less regulated.
Indeed, this sentence applies specifically to NO$_x$ and SO$_2$ emissions due to the sectors which are projected to regulated :

**These two SSPs were selected because they represent divergent scenarios for global NOx and SO$_2$ emissions. SSP1-2.6 represents a "low" scenario with stringent emission regulations, implemented almost worldwide, on various economic sectors such as energy generation, industrial processes and transportation.**

L130 22 tracers representing aerosol.There is an extensive discussion of the microphysics, but relevant for this paper, it is not clear to me how the completion for nitrate between coarse and fine fraction aerosol is modelled.

This aspect is brought line 196:

A modal approach for the size distribution is used to track the number and mass of aerosols which is described by a superposition of five log-normal modes (Schulz, 2007). The particle modes are represented for three ranges: sub-micronic (diameter <1 μm) corresponding to the accumulation mode, micronic (diameter between 1 and 10 μm) corresponding to coarse particles, and super-micronic or super coarse particles (diameter >10 μm).

L161 this an important limitation that should be mentioned upfront (i.e. not evaluating climate change influence on the emissions).

This sentence is a mistake. The future simulated emissions by CAMEO do include climate change as assessed in Beaudor et al., 2024. This sentence was inherited from a first draft version of the paper and has been removed.

We apologize for this confusion and understand why the reviewer was not convinced at first by the relevance of our work considering this sentence.
The following sentence has been added instead:

**The combined impact of climate change and future agricultural emissions $NH_3$ on atmospheric chemistry and climate is an interesting topic to further investigate in the future.**

L164 The ocean emission estimate is probably an upper limit; e.g. Paulot et al. 2016 give twice lower estimates.

We agree and we acknowledge this difference as follows:

**.. which is higher than the estimate from Paulot et al., 2015 (2-5 TgN/yr).**

L171 I would recommend to include a set of simulations that also uses the SSP1. As eluded to previously, the lack of comparison of the community SSP scenarios to the ones from CAMEO, leaves an open question on the novelty of the results.

Adding the CTM simulations for SSP1 is challenging since we did not simulate the agricultural $NH_3$ emissions with CAMEO for this specific scenario.
For tackling the lack of comparison, we added a hypothetical analysis based on the regional differences observed in the IPCC emissions and the CAMEO emissions projected for 2100 (this aspect is detailed in the first comment).

L200-214 It will be useful to also provide the relative changes in percent to the absolute numbers.

We mentioned relative changes in percent into parentheses when relevant. For instance see, the following sentence:

**When the CEDS inventory is replaced by CAMEO in LMDZ-INCA, the global simulated columns are 50% higher (of around 0.04 molecules x $10^{16}$ cm$^{-2}$) but closer to the IASI-measured global average (0.15 molecules x $10^{16}$ cm$^{-2}$).**

L212 Biomass burning inventory of NH3?
This sentence has been added at line 110:

**Emissions from biomass burning, including small fires from agricultural waste burning come from the Global Fire Emissions Database (GFEDs) inventory (Van der Werf et al., 2017). $NH_3$ emissions from fire account for 4.2 TgN/yr for the historical period.**

L217 Do I understand correctly that CEDS simulation was run without natural emissions? Isn't that comparing apples and pears?

The CEDS simulation did not run with natural soil emissions while CAMEO dataset does include them. Most of the CTM that investigated $NH_3$ emissions and aerosol formation up to now were not run with natural soil emissions since this dataset is not easily available. CEDS inventory does not provide natural soil emissions since it is an anthropogenic sources inventory.
Our objective was to analyze the benefit of using process-based emissions for $NH_3$ (i.e from CAMEO), by adding natural emissions to CEDS, we would not have been able to assess as clearly this aspect.

L225 the S and T markers in the Taylor plots are not terribly well explained- is it discussed somewhere what is evaluated with this?

Explanation has been added L.279

**The Taylor plots in Figure 2 represent statistical metrics for both temporal and spatial analyses. The temporal analysis is shown for monthly time steps, using triangle markers with T labels, and involves averaging over the corresponding regions. On the other hand, the spatial analysis is derived by averaging over the monthly time-series from 2011-2014, indicated by plain circle markers with S labels. These plots include metrics such as normalized standard deviation (plotted on the x-y axis, where the observation is normalized to 1), Pearson's R correlation, and a skill function, represented by grey isolines.**

L235 the monthly column comparison show indeed improvement of column levels over Africa and S. America, but not really or even contradicting elsewhere. What can we still learn about CAMEO vs GCM modelvs CEDS?

By looking at the Taylor plots, we can see that CAMEO does improve the seasonal and spatial variabilities of the $NH_3$ columns worldwide except in Europe and the US.
An additional sentence is incorporated to highlight what can be learnt about these results:

**While the CAMEO emission prescription appears promising for improving the seasonal cycle of the columns, there is still potential for refinement in the process-based approach, particularly in Europe and the US, where summer emissions appear excessively high. Future advancements in bi-directional flux, accounting for deposition and the compensation point, could address this issue.**

**Moreover, in Africa, biomass burning emissions significantly impact temporal representation, which is presently derived from an external inventory (Van der Werf et al., 2017).**

249 I would say that the 'gold' standard for quality controlled deposition observations is by the WMO GAW program. Vet et al.However, I think that not all data needed for this study where available. It would be relevant to mention this 'lack' of evaluated data in discussion (if considered important).

Thank you for raising this aspect. Deposition observations will be used in the next step of the work, to evaluate the performance of the bi-directional NH$_3$ fluxes scheme.

Figure 4,5,6 It is hard to get a general picture from the surface concentrations comparison, but overall in particular the measured particulate concentrations of NH4/NO3 seem to be up to a factor of 10 higher than the modelled ones for all networks. What are the possible consequences for this work? Have you considered mismatch of SO4 as one of the root causes for discrepeancies?

A combinaison of factors explain the low simulated nitrate concentrations at the surface.
This version of the model has always shown a strong vertical transport combined with low scavenging in the upper troposphere (Bian et al., 2017);
To some extend, this strong transport of nitrates to the upper troposphere is a robust signal and has been observed in the Asian Tropopause Aerosol Layer region during the monsoon season (Höpfner et al., 2019; Yu et al., 2022);
However, the CAMEO NH$_3$ emissions are significantly increased compared to CEDS during this period (JJA) over India; more nitrates are produced and subsequently transported to the upper-troposphere in that region and then spread all over the globe due to the high residence time of aerosols in the UT.
This feature of the scavenging is currently investigated in a newer version (79 levels, CMIP6 physics) of the model (PhD N. Février).

This aspect has been incorporated into the manuscript at the end of the model-observation comparison.

L512 Do you mean 'reducing agricultural' emissions- which can be done by e.g. reducing livestock numbers, but also practices (e.g. feed or manure management).
We clarified.

L534 twice higher than the deposition budget of the three alternative estimates. Corrected.

L535 higher NH3 emissions in equatorial Africa (clarify) We clarified.

L536. It is not well explained how wet deposition of NH3 is considered- where NH3 perse has a low Henry's coefficient.

We apologize but it is a confusion from our side, the Henry's law constant has been updated according to Bian et al., 2017.
We corrected this aspect in the manuscript.

L539 what is meant with 'good correlation'; and how are EMEP and CCMI modelling results entering the story? We corrected it for "good agreement".

L542 is deficient? Do you mean absent (i.e. they provide annual numbers)? The CCMI deposition dataset is a crucial… We corrected.

L543 I would agree with this statement, but it raises the question why it was not included (or maybe it is, but not clearly described).In general it should be considered that it is probably to be considered that in the end we are talking about ecosystem emissions, which would included interactions between soil, vegetation and atmosphere.

We included this section at line 700:

**Incorporating the nitrogen cycle into Earth System Models (ESM) is a recent advancement, as highlighted by Davies-Barnard (2022). Developing interactions of nitrogen compounds is complex due to the intricate processes involved, necessitating readiness in coupling atmospheric chemistry and land components. The studies by Pleim (2019) and Vira (2019, 2022) provide a foundational step toward bidirectional ammonia handling, though not yet fully integrated into existing ESMs. Vira (2022) notes that FANv2 does not currently feed back nitrogen losses to the nitrogen cycle in the Community Land Model, leaving fertilizer nitrogen availability to crops unaffected. Our approach does include feedback from nitrogen loss affecting available nitrogen for vegetation, even without a bidirectional scheme, yet exploited. Additionally, we uniquely incorporated nitrogen biomass removal from livestock needs, ensuring nitrogen and carbon budget accuracy. Efforts are ongoing to develop nitrogen species exchanges at the atmosphere-surface interface in the IPSL-ESM, aiming to assess chemical and climate impacts through interactive coupling.**

L550 clearly state that this is future perspective. We clarified.

L555 for sure future livestock is at the basis of many future emission estimates. It is the combination with 'interactive' soils that is probably not explored.
As mentionned, we are not aware of any future product of *gridded* livestock exploited for future ammonia emission projections. It is worth noting that the worldwide IIASA database provide only regional trends of livestock.

L560 (and throughout paper when talking about nitrate do you mean HNO3, NO3- or the sum of the two?
Thanks for bringing it to our attention.
For this specific line, we referred as nitric acid and corrected it to be more precise.
"Nitrate" throughout the text is considered as $NO_3^-$ only.

L595 It is not so clear to me what the Bertagni study is calculating. Still the N2O from atmospheric processes, or e.g. the additional N2O emission resulting from enhance NH3 deposition? Clarify.Agree that this is an important issue in particular if emissions from NH3 as an energy carrier are not well controlled (which it should as it is a quite dangerous and toxic component).
It is an interesting aspect.

Bertagni et al., estimate the same $N_2O$ atmospheric source as we do from the ammonia oxidation. See this quote from their paper referring to the factor they used in their approach: *1% of the nitrogen in ammonia can be converted into $N_2O$ following ammonia reaction with the atmospheric OH radical.*
We made it clearer in the text.

L597-602 I encourage the authors to persue this work, as it is probably going to be quite important.—
We appreciate the positive encouragements from the reviewer and are excited to continue this investigation.